# AdaSCALE: Adaptive Scaling for OOD Detection

## Abstract

The ability of the deep learning model to recognize when a sample falls outside its learned distribution is critical for safe and reliable deployment. Recent state-of-the-art out-of-distribution (OOD) detection methods leverage activation shaping to improve the separation between in-distribution (ID) and OOD inputs. These approaches resort to sample-specific scaling but apply a static percentile threshold across all samples regardless of their nature, resulting in suboptimal ID-OOD separability. In this work, we propose **AdaSCALE**, an adaptive scaling procedure that dynamically adjusts the percentile threshold based on a sample's estimated OODness. This estimation leverages our key observation: OOD samples exhibit significantly more pronounced activation shifts at high-magnitude activations under minor perturbation compared to ID samples. AdaSCALE enables stronger scaling for likely ID samples and weaker scaling for likely OOD samples, yielding highly separable energy scores. Our approach achieves state-of-the-art OOD detection performance, outperforming the latest rival OptFS by **14.94%** in near-OOD and **21.67%** in far-OOD datasets in average FPR@95 metric on the ImageNet-1k benchmark across eight diverse architectures.

## 1 Introduction

The reliable deployment of deep learning models hinges on their ability to handle unknown inputs, a task commonly known as OOD detection. A critical application is in medical diagnosis, where a model trained on common diseases should be able to flag inputs representing unknown conditions as potential outliers, requiring further review by clinicians. OOD detection primarily involves identifying semantic shifts, with robustness to covariate shifts being a highly desirable characteristic (Yang et al., 2023; Baek et al., 2024). As modern deep learning models scale in both data and parameter counts, effective OOD detection within large-scale settings is critical. Given the difficulties of iterating on large models, *post-hoc* approaches that preserve ID accuracy are generally preferred.

A variety of post-hoc approaches have emerged, broadly categorized by where they operate. One class of methods focuses on computing OOD scores directly in the output space (Hendrycks & Gimpel, 2017; Liang et al., 2018; Liu et al., 2020; Hendrycks et al., 2022a; Djurisic et al., 2023; Zhao et al., 2024), while another operates in the activation space (Lee et al., 2018; Sun et al., 2022; Ren et al., 2021; Rajasekaran et al., 2024). Finally, a more recent line of research also explores a hybrid approach (Wang et al., 2022; Kim et al., 2024), combining information from both spaces. The efficacy of many high-performing methods relies on either accurate computation of ID statistics (Sun et al., 2021; Kong & Li, 2022; Xu et al., 2023; Sun & Li, 2022; Krumpl et al., 2024) or retention of training data statistics (Sun et al., 2022; Rajasekaran et al., 2024). However, as retaining full access to training data becomes increasingly impractical in large-scale settings, methods that operate effectively with minimal ID samples are particularly valuable for practical applications.

Alleviating the dependence on ID training data/statistics, recent state-of-the-art post-hoc approaches center around the concept of "fixed scaling." ASH (Djurisic et al., 2023) prunes and scales activations on a per-sample basis. SCALE (Xu et al., 2024), the direct successor of ASH, critiques pruning and focuses purely on scaling, which improves OOD detection performance without accuracy degradation. LTS (Djurisic et al., 2024) extends this concept by directly scaling logits instead, using post-ReLU activations. These methods leverage a key insight: scaling based on relative strength of a sample's *top-k* activations (with respect to entire activations) yields highly separable ID-OOD energy scores.

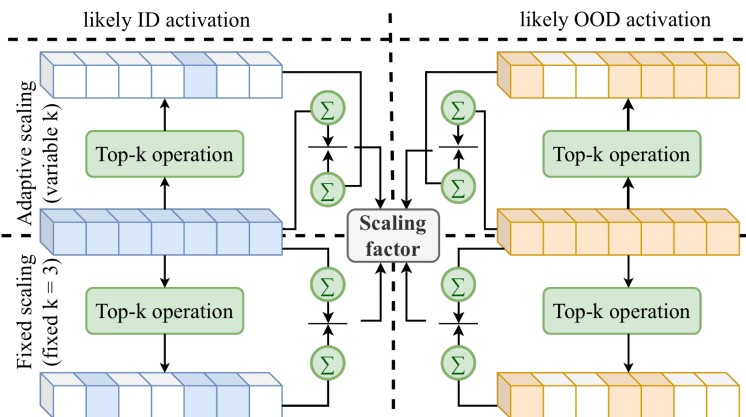

Figure 1: **Adaptive scaling (AdaSCALE) vs. fixed scaling (ASH (Djurisic et al., 2023), SCALE (Xu et al., 2024), LTS (Djurisic et al., 2024)).** While fixed scaling approaches uses a constant percentile threshold $p$ and hence constant $k$ (e.g., $k = 3$) across all samples, AdaSCALE adjusts $k$ based on estimated OODness. AdaSCALE assigns larger $k$ values (e.g., $k = 5$) to OOD-likely samples, yielding smaller scaling factors, and smaller $k$ values (e.g., $k = 1$) to ID-likely samples, yielding larger scaling factors. This adaptive mechanism enhances ID-OOD separability. (See Figure 4 for complete working mechanism.)

However, although such approaches provide sample-specific scaling factors, the scaling mechanism remains uniform across all samples as the percentile threshold $p$ and thereby $k$ is fixed, as shown in Figure 1. This static approach is inherently limiting for optimal ID-OOD separation while also failing to leverage even minimal ID data, which could be reasonably practical in most deployment scenarios.

We hypothesize that designing an adaptive scaling procedure based on each sample's estimated OODness offers greater control for enhancing ID-OOD separability. Specifically, this mechanism should assign smaller scaling factors for samples with high OODness to yield lower-magnitude energy scores and larger scaling factors for probable ID samples to yield higher-magnitude energy scores. To achieve this, we propose a heuristic for estimating OODness based on a key observation in activation space: minor perturbations applied to OOD samples induce significantly more pronounced shifts in their top-k activations compared to ID samples. Consequently, samples exhibiting substantial activation shifts are assigned lower scaling factors, while those with minimal shifts receive higher scaling factors. This adaptive scaling mechanism can be applied in either logit or activation space. Our method, **AdaSCALE**, achieves state-of-the-art performance, delivering significant improvements in OOD detection albeit with higher computational cost while requiring only minimal ID samples.

We conduct an extensive evaluation across 8 architectures on ImageNet-1k and 2 architectures on CIFAR benchmarks, demonstrating the substantial effectiveness of AdaSCALE. For instance, AdaSCALE surpasses the average performance of the *best-generalizing* method, OptFS (Zhao et al., 2024), by **14.94%**/**8.96%** for near-OOD detection and **21.48%**/**3.39%** for far-OOD detection in terms of FPR@95 / AUROC, on the ImageNet-1k benchmark across eight architectures. Furthermore, AdaSCALE outperforms the *best-performing* method, SCALE (Xu et al., 2024), when evaluated on the ResNet-50 architecture, achieving performance gains of **12.95%**/**6.44%** for near-OOD and **16.79%**/**0.79%** for far-OOD detection. Additionally, AdaSCALE consistently demonstrates superiority in full-spectrum OOD (FSOOD) detection (Yang et al., 2023). Our key contributions are:

- We reveal that OOD inputs exhibit more pronounced shifts in top-k activations under minor perturbations compared to ID inputs. Leveraging this, we propose a novel post-hoc OOD detection method using adaptive scaling that attains state-of-the-art OOD detection.

- We demonstrate state-of-the-art generalization of AdaSCALE via extensive evaluations across 10 architectures and 3 datasets by tuning mere one hyperparameter for a given setup.

## 2 RELATED WORKS

**Post-hoc methods.** Early research on OOD detection primarily focused on designing scoring functions based on logit information (Hendrycks & Gimpel, 2017; Liang et al., 2018; Liu et al., 2020; Hendrycks et al., 2022a; Liu et al., 2023). While these methods leveraged logit-based scores, alternative approaches have explored gradient-based information, such as GradNorm (Huang et al., 2021), GradOrth (Behpour et al., 2023), GAIA (Chen et al., 2023), and Greg-OOD (Sharifi et al., 2025). Given the limited dimensionality of the logit space, which may not encapsulate sufficient information for OOD detection, subsequent studies have investigated activation-space-based methods. These approaches exploit the high-dimensional activations, leading to both parametric techniques such as MDS (Lee et al., 2018), MDS Ensemble (Lee et al., 2018), and RMDS (Ren et al., 2021), as well as non-parametric methods such as KNN-based OOD detection (Sun et al., 2022; Park et al., 2023). Recent advancements have proposed hybrid methodologies that integrate parametric and non-parametric techniques to improve robustness. For instance, ComboOOD (Rajasekaran et al., 2024) combines these paradigms to enhance near-OOD detection performance. Similarly, VIM (Wang et al., 2022) employs a combination of logit-based and distance-based metrics. However, reliance of such approaches on ID statistics (Sun & Li, 2022; Olber et al., 2023; Zhang et al., 2023c) can become a constraint, hindering scalability and practical deployment in real-world applications. To mitigate computational challenges for real-world deployment, recent methods, such as FDBD (Liu & Qin, 2024) and NCI (Liu & Qin, 2023), have focused on enhancing efficiency. Recent advances, such as NECO (Ammar et al., 2024) examines connections to neural collapse phenomena, while WeiPer (Granz et al., 2024), explore class-direction perturbations. Unlike WeiPer, our work deals with perturbation in the input image similar to ODIN (Liang et al., 2018).

**Activation-shaping post-hoc methods.** A seminal work in OOD detection, ReAct (Sun et al., 2021), identified abnormally high activation patterns in OOD samples and proposed clipping extreme activations. LINe (Ahn et al., 2023) integrates activation clipping with Shapley-value-based pruning, selectively masking irrelevant neurons to mitigate noise. Activation clipping has been further generalized by BFAct (Kong & Li, 2022) and VRA (Xu et al., 2023) for enhanced effectiveness. Additionally, BATS (Zhu et al., 2022) refines activation distributions by aligning them with their respective typical sets, while LAPS (He et al., 2024) enhances this strategy by incorporating channel-aware typical sets. Inspired by activation clipping, another line of research explores activation "scaling" as a means to improve OOD detection. ASH (Djurisic et al., 2023) introduces a method to compute a scaling factor as a function of the activation itself, pruning and rescaling activations to enhance the separation of energy scores between ID and OOD samples. However, this approach results in a slight degradation in ID classification accuracy. In response, SCALE (Xu et al., 2024) observes that pruning adversely affects performance and thus eliminates it, leading to improved OOD detection while preserving ID accuracy. SCALE currently represents the state-of-the-art method for ResNet-50-based OOD detection. Despite their efficacy, these activation-based methods exhibit limited generalization across diverse architectures. To address this issue, LTS (Djurisic et al., 2024) extends SCALE by computing scaling factors using post-ReLU activations and applying them directly to logits rather than activations. Our work builds on this line of work, introducing the adaptive scaling mechanism. ATS (Krumpl et al., 2024) argues that relying solely on final-layer activations may result in the loss of critical information beneficial for OOD detection and proposes to leverage intermediate-layer activations too. However, its efficacy is contingent upon the availability of a large number of training samples, whereas our approach attains state-of-the-art performance while utilizing a minimal ID samples. A newly proposed method OptFS (Zhao et al., 2024) introduces a piecewise constant shaping function with the goal of generalization across diverse architectures in large-scale settings, while our work exhibits superior generalization extending to small-scale settings too.

**Training methods.** The training methods incorporate adjustments during training to enhance the ID-OOD differentiating characteristics. They either make architectural adjustments (DeVries & Taylor, 2018; Hendrycks et al., 2019b; Hsu et al., 2020), apply enhanced data augmentations (Xiong et al., 2024; Hendrycks* et al., 2020; Hendrycks et al., 2022b), or make simple training modifications Wei et al. (2022); Regmi et al. (2024a); Zhang et al. (2024). More recent methods have adopted contrastive learning in the context of OOD detection (Ming et al., 2023; Regmi et al., 2024b; Lu et al., 2024; Zou et al., 2025). Moreover, some approaches also either utilize external real outliers (Hendrycks et al., 2019a; Zhang et al., 2023a; Du et al., 2024; Zhu et al., 2023) or synthesize virtual outliers either in image space (Du et al., 2023; Regmi, 2024; Wang et al., 2023; Gao et al., 2023; Zhang et al., 2025; Li et al., 2024; Bai et al., 2024; Nie et al., 2024) or in feature space (Du et al., 2022; Tao et al., 2023;

Gao et al., 2024; Li & Zhang, 2025). However, training methods can be costlier and less effective than post-hoc approaches in some large-scale setups (Yang et al., 2022).

## 3 PRELIMINARIES

Let $\mathcal{X}$ denote the input space and $\mathcal{Y} = \{1, 2, ..., C\}$ denote the label space, where $C$ is the number of classes. We consider a multi-class classification setting where a classifier $h$ is trained on ID data drawn from an underlying joint distribution $\mathcal{P}_{\text{ID}}(x, y)$, where $x \in \mathcal{X}$ and $y \in \mathcal{Y}$. The ID training dataset is denoted as $\mathcal{D}_{\text{ID}} = \{(x_i, y_i)\}_{i=1}^N$, where $N$ is the number of training samples and $(x_i, y_i) \sim \mathcal{P}_{\text{ID}}(x, y)$. The classifier $h$ is composed of a feature extractor $f_\theta : \mathcal{X} \to \mathcal{A} \in \mathbb{R}^D$, and a classifier $g_{\mathcal{W}} : \mathcal{A} \to \mathcal{Z} \in \mathbb{R}^C$. The feature extractor maps an input $x$ to a feature vector $\mathbf{a} \in \mathcal{A}$, where $\mathbf{a} = f_\theta(x)$ and the classifier then maps this feature vector to a logit vector $\mathbf{z} = g_{\mathcal{W}}(\mathbf{a}) \in \mathbb{R}^C$. We refer to individual dimensions of the feature vector $\mathbf{a}$ as activations, denoted by $a_j$ for the $j$-th dimension. The classifier $h$ is trained on $\mathcal{D}_{\text{ID}}$ to minimize the empirical risk: $\min_{\theta, \mathcal{W}} \frac{1}{N} \sum_{i=1}^N \mathcal{L}(g_W(f_\theta(x_i)), y_i)$ where $\mathcal{L}$ is a loss function, such as cross-entropy loss. During inference, the model may encounter data points drawn from a different distribution, denoted as $\mathcal{P}_{\text{OOD}}(x)$, which is referred to as OOD data. The OOD detection problem aims to identify whether a given input $x$ is drawn from marginal distribution $\mathcal{P}_{\text{ID}}(x)$ or from $\mathcal{P}_{\text{OOD}}(x)$. Hence, the goal is to design a scoring function $S(x) : \mathcal{X} \to \mathbb{R}$ that assigns a scalar score to each input $x$, reflecting its likelihood of being an OOD sample. A higher score typically indicates a higher probability of the input being OOD. A threshold $\tau$ is used to classify an input as either ID or OOD: $\text{OOD}(x) = \begin{cases} \text{True}, & \text{if } S(x) > \tau \\ \text{False}, & \text{if } S(x) \leq \tau \end{cases}$.

## 4 METHOD

In this section, we introduce AdaSCALE, a novel post-processing approach that dynamically adapts the scaling mechanism based on each sample's estimated OODness. We first revisit and analyze the core principle underlying recent scaling-based static state-of-the-art approaches. Then, we present our key empirical observations regarding activation behavior under minor perturbations, building upon insights from ReAct (Sun et al., 2021). Finally, we detail our proposed adaptive scaling mechanism that leverages these observations to achieve superior OOD detection performance.

### 4.1 REVISITING STATIC SCALING MECHANISM

Scaling baselines (Djurisic et al., 2023; Xu et al., 2024; Djurisic et al., 2024) use energy score $-\log \sum_{i=1}^C e^{(\mathbf{z}_i)}$ on (directly or indirectly) scaled logits, with higher (magnitude) values indicating higher IDness. They operate by scaling activations / logits with scaling factor $r$ computed as:

$$r = \left( \frac{\sum_j \mathbf{a}_j}{\sum_{\mathbf{a}_j > P_p(\mathbf{a})} \mathbf{a}_j} \right) \tag{1}$$

where $P_p(\mathbf{a})$ denotes the $p^{th}$ percentile of all elements in activation $\mathbf{a}$. While this approach yields sample-specific scaling factors, it imposes a critical constraint: the $p^{th}$ percentile threshold is static and identical across all test samples, regardless of the nature of samples. We argue that this static nature limits the effectiveness of the scaling procedure and prevents optimal ID-OOD separability. We hypothesize that a proxy for estimating the OODness can be useful for computing dynamic percentiles giving rise to dynamic scaling factors.

### 4.2 OBSERVATIONS IN ACTIVATION SPACE

A seminal work ReAct (Sun et al., 2021) demonstrated that OOD samples often induce abnormally high activations within neural networks. We extend this finding with an important observation: *the positions of such high activations in OOD samples are relatively unstable under minor perturbations compared to ID samples*. This instability provides a valuable signal for distinguishing OOD samples from ID samples. Below, we formalize this observation.

### 4.2.1 PERTURBATION MECHANISM

Let $x \in \mathbb{R}^{C_{\text{in}} \times H \times W}$ be an input image with $C_{\text{in}}$ input channels, $H$ height, and $W$ width. We denote channel value at position $(c, h, w)$ as $x[c, h, w]$. To identify channel values for perturbation, we employ pixel attribution that quantifies each input element's influence on the model's prediction. An attribution function, $AT(x, c, h, w)$, assigns a score to each channel value, with *lower* absolute scores indicating *less* influence. We select $o\%$ of channel value indices with *lowest* absolute attribution scores, forming the set $R$. We use a gradient-based attribution $AT(x, c, h, w) = \frac{\partial(g_W(f_\theta(x)))_{y_{pred}}}{\partial x[c,h,w]}$, where $y_{pred}$ is predicted class index. To create a perturbed input, we select a subset $R$ containing $o\%$ of channel values to perturb. If $\varepsilon$ is perturbation magnitude, the perturbed image $x^\varepsilon$ is obtained as:

$$x^\varepsilon[c, h, w] = \begin{cases} x[c, h, w] + \varepsilon \cdot \text{sign}(AT(x, c, h, w)), & \text{if } (c, h, w) \in R \\ x[c, h, w], & \text{if } (c, h, w) \notin R \end{cases} \tag{2}$$

> **Remark: Is gradient attribution necessary for perturbation?**
>
> While we employ gradient-based attribution for principled pixel selection for perturbation, as we show later in Section E.5, it is important to note that even random selection empirically performs similarly, whereas selecting salient pixels degrades performance.

### 4.2.2 ACTIVATION SHIFT AS OOD INDICATOR

After obtaining perturbed input $x^\varepsilon$, we compute its activation $\mathbf{a}^\varepsilon = f_\theta(x^\varepsilon)$. We define *activation shift* as the absolute element-wise difference between original activation and the perturbed one:

$$\mathbf{a}^{\text{shift}} = |\mathbf{a}^\varepsilon - \mathbf{a}| \tag{3}$$

Figure 2 illustrates the key insight of our approach: activation shift at extreme (high-magnitude) activations is consistently more pronounced in OOD samples compared to ID samples. This behavior can be understood intuitively: ID samples activate network features in a stable, predictable manner reflecting learned patterns, while OOD samples trigger less stable, more arbitrary high activations that shift significantly under perturbation. Based on this observation, we propose using activation shift at the top-$k_1$ highest activations as a metric to estimate OODness of a sample:

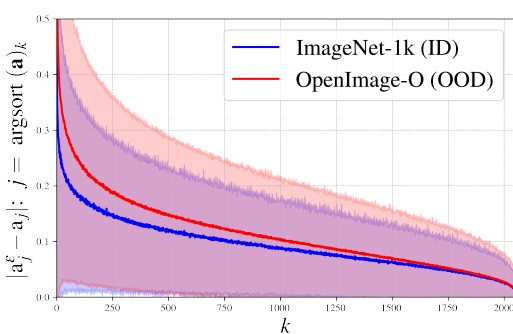

$$Q = \sum_{j \in \text{argsort}(\mathbf{a}, \text{ desc}=\text{True})[:k_1]} (|a_j^\varepsilon - a_j|) \tag{4}$$

where $\text{argsort}(\mathbf{a}, \text{ desc} = \text{True})[: k_1]$ returns the indices of the $k_1$ highest values in $\mathbf{a}$. As evidenced by $Q_{\text{OOD}}/Q_{\text{ID}}$ ratio ($> 1$) shown in Figure 3, the $Q$ statistic generally assigns higher values to OOD samples than ID ones. However, the high variance of $Q$ metric (Figure 2) sug-

Figure 2: Activation shift comparison (with the mean denoted by a solid line and the standard deviation by a shaded region) between ID and OOD in the ResNet-50 model. The activation shift is significantly more pronounced in OOD samples compared to ID samples at high-magnitude activations (left side of the x-axis), providing a discriminative signal for OOD detection.

gests the possibility of overoptimistic estimations. To address this issue, we introduce a correction term $C_o$ that exhibits an opposing behavior: it tends to be higher for ID samples than for OOD samples. Figure 5 in Section C shows that the perturbed activations of ID samples tend to be higher than those of OOD ones, especially in high-activation regions. We leverage this complementary signal by defining $C_o = \sum_{j \in \text{argsort}(\mathbf{a}, \text{ desc}=\text{True})[:k_2]} \text{ReLU}(a_j^\varepsilon)$, where $k_2$ is a hyperparameter denoting the number of considered activations. We refine our OOD quantification by combining both metrics, weighted by a hyperparameter $\lambda$:

$$Q' = \lambda \cdot Q + C_o \tag{5}$$

> The motivation behind $Q'$ formulation instead of $Q$ alone is to prevent the overconfident scaling factor from dominating the logit's contribution in the final energy score. (See Section D.1)

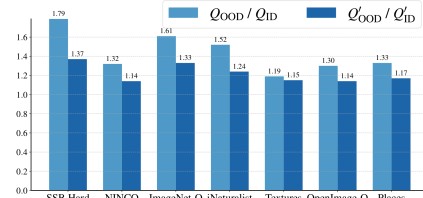

Figure 4: **Schematic diagram of AdaSCALE's working mechanism.** AdaSCALE computes activation shifts between an original image and its slightly perturbed counterpart to estimate OODness. It, in-turn, determines an adaptive percentile threshold ($p$ and thereby $k$), which controls the scaling factor $r$. Since $r$ is defined as the ratio of total activation sum to the sum of activations above the percentile threshold, samples with higher OODness receive lower scaling factors. This adaptive approach yields highly separable energy scores that enable effective OOD detection.

Indeed, Figure 3 illustrates that $Q'_{\text{OOD}}/Q'_{\text{ID}} < Q_{\text{OOD}}/Q_{\text{ID}}$, suggesting that the correction term $C_o$ helps mitigate overconfident estimations. If $\bar{Q}_s = \{\bar{Q}'_1, \bar{Q}'_2, ..., \bar{Q}'_{n_{\text{val}}}\}$ be the set of $Q'$ values on $n_{\text{val}}$ ID validation samples, we could transform any $Q'$ into a normalized probability scale by constructing empirical cumulative distribution function (eCDF) derived from $\bar{Q}_s$. The eCDF, denoted as $F_{Q'}(Q')$, can be defined as: $F_{Q'}(Q') = \frac{1}{n_{\text{val}}} \sum_{i=1}^{n_{\text{val}}} \mathbb{1}(\bar{Q}'_i \leq Q')$, where $\mathbb{1}(\cdot)$ is the indicator function. A higher value of $F_{Q'}(Q')$ indicates a higher likelihood of the sample being OOD. Importantly, our experiments suggest that as few as 10 ID validation samples are sufficient to construct an effective eCDF for this purpose (See Table 9).

Figure 3: $Q_{\text{OOD}}/Q_{\text{ID}}$ vs $Q'_{\text{OOD}}/Q'_{\text{ID}}$ in various OOD datasets with ResNet-50 on ImageNet-1k. $Q'_{\text{OOD}}/Q'_{\text{ID}} < Q_{\text{OOD}}/Q_{\text{ID}}$ suggests $C_o$ helps mitigate overconfident estimations.

> **Remark: ODIN vs. AdaSCALE in terms of perturbation**
>
> ODIN (Liang et al., 2018) perturbs entire image, inducing stronger confidence in ID inputs than OOD ones. In contrast, we apply trivial perturbations, perturbing only small number of trivial/random pixels to primarily compute shifts in top-k activations.

### 4.3 PROPOSED APPROACH: ADAPTIVE SCALING

Building on our observations, we propose AdaSCALE (Adaptive SCALE), a novel approach that introduces dynamic, sample-specific adjustments to the scaling procedure. The key insight is that $p^{th}$ percentile threshold should be a function of each test sample's estimated OODness rather than a fixed value. The scaling factor $r$ increases as the $p^{th}$ percentile threshold rises (i.e., when more activations are excluded from the denominator in Equation 1). For optimal ID-OOD separation, we must scale ID samples more strongly than OOD samples, requiring a higher $p^{th}$ percentile for ID samples. We define an adaptive percentile threshold as: $p = p_{\min} + (1 - F_{Q'}(Q')) \cdot (p_{\max} - p_{\min})$, where $p_{\min}$ and $p_{\max}$ are hyperparameters that define the minimum and maximum limits of percentile threshold. It ensures samples with lower estimated OODness receive higher percentile thresholds, resulting in stronger scaling. (See Algorithm 1). We implement two variants: **AdaSCALE-A** scales activations as $\mathbf{a}_{\text{scaled}} = \mathbf{a} \cdot \exp(r)$ (Djurisic et al., 2023; Xu et al., 2024). **AdaSCALE-L** scales logits as $\mathbf{z}_{\text{scaled}} = \mathbf{z} \cdot r^2$ (Djurisic et al., 2024). This approach is also outlined in Figure 4.

## 5 EXPERIMENTS

We use pre-trained models provided by PyTorch for ImageNet-1k experiments. For CIFAR experiments, we train three models per network using the standard cross-entropy loss and report the mean results across these three independent trials. The evaluation setup is provided in Table 1. The non-percentile hyperparameters ($\lambda = 10, k_1 = 1\%, k_2 = 5\%, o = 5\%, \epsilon = 0.5$) were determined only once using ResNet-50 model via OpenOOD's (Yang et al., 2022; Zhang et al., 2023b) automatic parameter search. Although we tune ($p_{\min}, p_{\max}$) for optimal results in each case (see Section G),

Table 2: OOD detection results (FPR@95 ↓ / AUROC ↑) on ImageNet-1k benchmark.

| Method | ResNet-50 | ResNet-101 | RegNet-Y-16 | ResNeXt-50 | DenseNet-201 | EfficientNetV2-L | ViT-B-16 | Swin-B | Average |
|---|---|---|---|---|---|---|---|---|---|
| **near-OOD** | | | | | | | | | |
| MSP | 74.23 / 60.21 | 71.96 / 67.25 | 62.22 / 80.74 | 73.25 / 67.86 | 73.44 / 67.29 | 72.51 / 80.76 | 86.72 / 68.62 | 87.11 / 69.82 | 75.18 / 70.32 |
| MLS | 74.87 / 64.55 | 72.05 / 71.51 | 62.94 / 84.66 | 74.11 / 71.62 | 75.51 / 68.91 | 81.44 / 79.22 | 93.78 / 63.64 | 94.80 / 64.68 | 78.69 / 71.10 |
| EBO | 75.32 / 64.52 | 72.32 / 71.54 | 62.80 / 84.76 | 74.21 / 71.61 | 75.85 / 68.68 | 82.86 / 77.15 | 94.37 / 59.19 | 95.34 / 59.79 | 79.13 / 69.66 |
| ReAct | 72.61 / 68.81 | 68.07 / 75.00 | 70.73 / 75.37 | 70.96 / 74.13 | 69.97 / 73.65 | 72.36 / 71.39 | 86.63 / 68.35 | 82.64 / 73.26 | 74.25 / 72.50 |
| ASH | 69.47 / 71.33 | 65.24 / 76.61 | 82.51 / 67.81 | 70.98 / 75.25 | 92.83 / 52.30 | 94.85 / 44.78 | 94.45 / 53.20 | 96.37 / 47.58 | 83.34 / 61.11 |
| SCALE | 67.76 / 74.20 | 63.87 / 78.60 | 67.09 / 82.90 | 70.59 / 76.20 | 71.56 / 73.72 | 89.70 / 60.12 | 94.48 / 56.18 | 88.62 / 61.47 | 76.71 / 70.42 |
| BFAct | 72.35 / 68.88 | 67.96 / 75.16 | 78.72 / 66.09 | 70.96 / 74.14 | 71.20 / 72.61 | 75.53 / 62.46 | 82.09 / 70.66 | **71.81 / 75.28** | 73.83 / 70.66 |
| LTS | 68.01 / 73.37 | 63.91 / 78.27 | 69.82 / 80.75 | 70.27 / 76.20 | 71.29 / 74.56 | 87.30 / 73.63 | 88.83 / 67.43 | 86.61 / 67.22 | 75.76 / 73.93 |
| OptFS | 69.66 / 70.97 | 65.46 / 75.83 | 73.53 / 75.21 | 69.27 / 74.84 | 71.74 / 72.10 | 72.29 / 75.29 | 76.55 / 72.73 | 76.81 / 74.06 | 71.91 / 73.88 |
| **AdaSCALE-A** | **58.98 / 78.98** | 57.96 / 81.68 | **47.91 / 89.18** | 64.14 / 79.96 | **61.28** / 79.66 | **53.78 / 86.94** | **71.87** / 73.14 | 73.41 / 74.48 | **61.17 / 80.50** |
| **AdaSCALE-L** | 59.84 / 78.62 | **56.41 / 81.86** | 56.13 / 87.11 | **62.08 / 80.18** | 61.75 / **80.06** | 54.95 / 85.77 | 71.99 / **73.23** | 72.89 / 74.58 | 62.00 / 80.18 |
| **far-OOD** | | | | | | | | | |
| MSP | 53.15 / 84.06 | 53.87 / 83.81 | 40.41 / 90.08 | 53.07 / 84.21 | 53.60 / 84.43 | 54.74 / 87.92 | 56.41 / 84.62 | 73.39 / 82.02 | 54.83 / 85.14 |
| MLS | 42.57 / 88.19 | 43.89 / 88.30 | 32.92 / 93.70 | 44.91 / 87.97 | 48.43 / 87.44 | 68.64 / 84.80 | 81.89 / 81.42 | 95.16 / 73.37 | 57.30 / 85.65 |
| EBO | 42.72 / 88.09 | 44.30 / 88.23 | 32.47 / 93.82 | 45.12 / 87.86 | 48.95 / 87.15 | 74.48 / 81.13 | 86.95 / 76.34 | 96.08 / 63.99 | 58.88 / 83.33 |
| ReAct | 30.14 / 92.98 | 29.89 / 93.10 | 45.20 / 86.17 | 30.06 / 92.65 | 30.72 / 92.65 | 60.05 / 75.33 | 59.31 / 83.65 | 58.86 / 84.77 | 43.03 / 87.67 |
| ASH | 24.69 / 94.43 | 26.18 / 94.06 | 59.65 / 83.94 | 29.17 / 93.47 | 33.50 / 92.17 | 96.56 / 41.57 | 95.98 / 52.16 | 98.23 / 43.20 | 57.99 / 74.38 |
| SCALE | 21.44 / 95.39 | 22.54 / 95.05 | 32.16 / 94.16 | 30.62 / 93.54 | 33.17 / 92.70 | 89.63 / 62.58 | 88.36 / 72.32 | 86.59 / 66.77 | 50.56 / 84.06 |
| BFAct | 29.46 / 93.01 | 29.43 / 93.04 | 58.69 / 77.22 | 29.71 / 92.67 | 32.45 / 92.29 | 66.72 / 65.70 | 51.58 / 85.77 | **38.99 / 88.47** | 42.13 / 86.02 |
| LTS | 22.20 / 95.24 | 23.07 / 94.94 | 34.99 / 93.57 | 30.37 / 93.49 | 30.92 / **93.29** | 86.85 / 76.30 | 64.37 / 84.43 | 85.84 / 44.80 | 47.33 / 84.51 |
| OptFS | 25.66 / 93.87 | 26.97 / 93.55 | 47.37 / 86.73 | 27.54 / 93.40 | 34.42 / 91.04 | 53.62 / 83.62 | **46.11** / 87.35 | 44.27 / 87.79 | 38.25 / 89.67 |
| **AdaSCALE-A** | **17.84 / 96.14** | 18.51 / **95.95** | 21.37 / 95.84 | **22.08 / 95.24** | 28.01 / **93.23** | **37.61 / 91.48** | 47.63 / 86.83 | 47.81 / 87.14 | **30.11 / 92.73** |
| **AdaSCALE-L** | 17.92 / 96.12 | 19.15 / 95.76 | **20.10 / 96.19** | 22.16 / 95.01 | **28.00** / 93.18 | 38.81 / 90.51 | 47.28 / 86.97 | 46.24 / 87.97 | 29.96 / 92.71 |

near-optimal performance can be achieved by tuning only $p_{\max}$ while fixing $p_{\min}$ to 60 across all 22 cases (See Table 5). Best results are **bold**, and second-best results are underlined.

**Metrics.** We use two commonly used OOD Detection metrics: Area Under Receiver-Operator Characteristics (AU-ROC) and False Positive Rate at 95% True Positive Rate (FPR@95), where a higher AUROC and lower FPR@95 indicates better OOD detection performance.

**Baselines.** MSP (Hendrycks & Gimpel, 2017), EBO (Liu et al., 2020), ReAct (Sun et al., 2021), MLS (Hendrycks et al., 2022a), ASH (Djurisic et al., 2023), SCALE (Xu et al., 2024), BFAct (Kong & Li, 2022), LTS (Djurisic et al., 2024), OptFS (Zhao et al., 2024). See Section D.2 for additional comparisons. Currently, SCALE is the *best-performing* method (with ResNet-50), while OptFS is the *best-generalizing* method.

Table 1: Experimental evaluation setup for OOD detection.

| Conventional OOD detection | | | |
|---|---|---|---|
| **ID datasets** | **Near-OOD** | **Far-OOD** | **Network** |
| CIFAR-10/100 | CIFAR-100/10 TIN | MNIST, SVHN, Textures, Places365 | WRN-28-10, DenseNet-101 |
| ImageNet-1k | SSB-Hard NINCO ImageNet-O | iNaturalist, OpenImage-O Textures Places | EfficientNetV2-L, ResNet-101 DenseNet-201, ViT-B-16 ResNet-50, ResNeXt-50 RegNet-Y-16, Swin-B |
| **Covariate shifted datasets for full spectrum OOD detection** | | | |
| ImageNet-1k | ImageNet-C, ImageNet-R, ImageNet-V2, ImageNet-ES | | |

## 5.1 Empirical Results

**CIFAR benchmark.** We compare AdaSCALE with post-hoc baselines on CIFAR benchmarks using WRN-28-10 and DenseNet-101 networks, reporting the averaged performance in Table 3. AdaSCALE outperforms all methods in average AUROC metric across CIFAR benchmarks in both near- and far-OOD detection. For far-OOD detection on CIFAR-10 benchmark, AdaSCALE-A achieves the best FPR@95 score of **33.11**, outperforming the MSP baseline by approximately 1.4 points. Similarly, AdaSCALE-A attains the best FPR@95 / AUROC of **43.07 / 90.31** in near-OOD detection, though MSP remains competitive. In near-OOD detection on CIFAR-100 benchmark,

Table 3: OOD detection results (FPR@95↓ / AUROC↑) averaged over WRN-28-10 and DenseNet-101 on CIFAR benchmarks across 3 trials. (See Section H for complete results.)

| Method | CIFAR-10 | | CIFAR-100 | |
|---|---|---|---|---|
| | **Near-OOD** | **Far-OOD** | **Near-OOD** | **Far-OOD** |
| MSP | 43.18 / 89.07 | 34.49 / 90.88 | **55.64** / 80.23 | 61.73 / 76.82 |
| MLS | 51.54 / 89.33 | 39.62 / 91.68 | 57.24 / 81.25 | 60.19 / 78.92 |
| EBO | 51.54 / 89.37 | 39.58 / 91.75 | 57.45 / 81.10 | 60.12 / 78.96 |
| ReAct | 49.71 / 88.59 | 37.32 / 92.00 | 63.20 / 79.58 | 54.78 / 80.46 |
| ASH | 78.11 / 77.97 | 63.12 / 83.35 | 80.97 / 70.09 | 69.38 / 79.06 |
| SCALE | 53.00 / 89.20 | 39.27 / 91.93 | 58.38 / 81.00 | 57.19 / 80.56 |
| BFAct | 54.90 / 88.56 | 43.05 / 90.66 | 72.26 / 74.70 | 57.44 / 77.63 |
| LTS | 55.71 / 88.77 | 41.06 / 91.74 | 59.98 / 80.60 | 50.48 / 81.79 |
| OptFS | 64.82 / 85.72 | 47.67 / 89.99 | 76.80 / 73.02 | 60.23 / 77.76 |
| **AdaSCALE-A** | **43.07 / 90.31** | **33.11** / 92.66 | 57.33 / **81.35** | 54.53 / 81.14 |
| **AdaSCALE-L** | 44.71 / 90.14 | 33.43 / **92.69** | 58.70 / 81.07 | **52.49 / 82.21** |

AdaSCALE-A achieves the highest AUROC of **81.35**, while in far-OOD detection, AdaSCALE-L reaches the best performance with FPR@95 / AUROC of **52.49 / 82.21**. While activation-shaping methods perform well in ImageNet-1k, they seem to underperform in CIFAR. In contrast, AdaSCALE achieves consistently superior performance across all setups.

**ImageNet-1k benchmark.** We compare our proposed method, AdaSCALE, with recent state-of-the-art approaches across eight architectures on the ImageNet-1k benchmark, as presented in Table 2. AdaSCALE demonstrates consistently strong performance across all architectures compared to existing methods. Specifically, it surpasses the *best-generalizing* method, OptFS, by **14.94%/8.96%** in the FPR@95/AUROC metric for near-OOD detection across all architectures. Additionally, it

outperforms the *best-performing method*, SCALE (on ResNet-50), by **12.96%**/**6.44%** in the same metric. A closer observation reveals that while OptFS excels in architectures such as EfficientNet, ViT-B-16, and Swin-B, scaling baselines perform comparably or even better in architectures like ResNet-50, ResNet-101, RegNet-Y-16, and DenseNet-201. In contrast, AdaSCALE-A achieves the best performance in near-OOD detection across all architectures, except for Swin-B, where BFAct performs optimally. Furthermore, effectiveness of AdaSCALE extends beyond near-OOD detection to far-OOD detection, demonstrating an average gain of **21.67%** over OptFS in the FPR@95 metric.

**FSOOD Detection.** FSOOD detection extends conventional OOD detection by incorporating model's ability to generalize on covariate-shifted ID inputs. We present FSOOD detection results in Table 4. We can observe that this is a highly challenging task, as covariate-shifted ID datasets cause a significant performance drop for all methods compared to conventional case. Despite this, AdaSCALE outperforms OptFS by **4.49** and **4.13** points on average in the FPR@95 metric for FSOOD detection across both near- and far-OOD datasets.

Table 4: FSOOD detection results on ImageNet-1k averaged over 8 architectures.

| Method | Near-OOD | | Far-OOD | |
|---|---|---|---|---|
| | FPR@95 ↓ | AUROC ↑ | FPR@95 ↓ | AUROC ↑ |
| ReAct | 87.22 | 51.38 | 67.23 | 69.53 |
| ASH | 87.01 | 52.02 | 72.36 | 65.70 |
| SCALE | 86.75 | 52.27 | 69.36 | 68.97 |
| BFAct | 87.12 | 51.14 | 66.13 | 69.69 |
| LTS | 86.46 | 53.29 | 66.76 | 71.63 |
| OptFS | 85.83 | 52.17 | 63.32 | 71.44 |
| AdaSCALE-A | **81.34** | 55.03 | 58.87 | 72.41 |
| AdaSCALE-L | 81.62 | **55.14** | 59.19 | 72.85 |

**Accuracy:** Like SCALE and LTS, AdaSCALE applies linear transformations to scale activations or logits, preserving accuracy, unlike post-hoc rectification methods (ASH, ReAct).

## 5.2 Ablation / Hyperparameter Studies / Discussion

**Sensitivity study.** We study the sensitivity of AdaSCALE-A hyperparameters with ResNet-50 on ImageNet-1k (Table 7). The optimal perturbation magnitude is $\varepsilon = 0.5$, close to RGB means and sufficient to disturb peak activations. Setting $\lambda = 10$ ensures activation shift $Q$ dominates the regularizer $C_o$. Optimal performance at $k_1 = 1\%$ confirms that shifts in peak activations are the most reliable OOD signal, while $k_2 = 5\%$ empirically performs best. $o = 5\%$ yielding optimal result supports our hypothesis that perturbation should be trivial enough to only disturb peak activations. (See Section E for complete sensitivity study.)

**Constrained hyperparameter tuning.** The hyperparameters $(\lambda, k_1, k_2, o, \epsilon) = (10, 1\%, 5\%, 5\%, 0.5)$ determined with ResNet-50 model in ImageNet-1k benchmark generalize across architectures and datasets. To demonstrate this, we conduct experiments only allowing $p_{\max}$ to be tuned across 8 and 2 diverse architectures in ImageNet-1k and CIFAR-100 datasets, respectively and present the results in

Table 5: Tuning just one hyperparameter $p_{\max}$ is enough for superior generalization of AdaSCALE over OptFS.

| Method | ImageNet-1k | | CIFAR-100 | |
|---|---|---|---|---|
| | Near-OOD | Far-OOD | Near-OOD | Far-OOD |
| OptFS | 71.91 / 73.88 | 38.25 / 89.67 | 76.81 / 73.02 | 60.23 / 77.76 |
| AdaSCALE-A | **62.29 / 79.72** | **32.72 / 91.82** | **58.00 / 81.14** | **56.47 / 80.99** |

Table 5. Inspired by early works that tune the percentile hyperparameter in $[60, 99]$, we set $p_{\min}$ to the lower limit 60. Even under this highly constrained setting, AdaSCALE significantly outperforms the previous state-of-the-art (OptFS) by an average of **13**% (FPR95) / **8**% (AUROC) on near-OOD and **14**% (FPR@95) / **2**% (AUROC) on far-OOD datasets on ImageNet-1k benchmark. The superiority of AdaSCALE over OptFS is also evident in a small-scale setting (CIFAR-100).

**Estimated OODness $Q'$.** Adaptive scaling depends on estimated OODness to determine the extent of scaling. We study the effect of various estimated OODness functions using ResNet-50 network (ImageNet-1k) in Table 6. It clearly shows Q component of $Q'$ being most critical while $\sum_{k=1}^{k_2} \mathbf{a}^{\varepsilon}_{\mathrm{argsort}(\mathbf{a})_k}$ as correction term being a relatively superior choice. However, estimated OODness alone – without adaptive scaling – does not result in strong performance.

Table 6: Ablation studies of $Q'$ in FPR@95 ↓ / AUROC ↑ format.

| $Q'$ | Near-OOD | Far-OOD |
|---|---|---|
| $Q$ without scaling | 79.81 / 72.32 | 84.00 / 68.13 |
| $Q$ | 59.43 / 78.14 | 19.70 / 95.73 |
| $\sum_{k=1}^{k_2} \mathbf{a}^{\varepsilon}_{\mathrm{argsort}(\mathbf{a})_k}$ | 70.39 / 74.00 | 21.40 / 95.31 |
| $\lambda \cdot Q + \sum_{k=1}^{k_2} \mathbf{a}^{\varepsilon}_{\mathrm{argsort}(\mathbf{a})_k}$ | 58.97 / **78.98** | **17.84 / 96.14** |
| $\sum_{k=1}^{k_2} \max_k (\mathbf{a})$ | 65.11 / 76.23 | 19.76 / 95.67 |
| $\lambda \cdot Q + \sum_{k=1}^{k_2} \max_k (\mathbf{a})$ | **58.91** / 78.74 | 18.02 / 96.08 |

Table 7: Sensitivity study (FPR@95↓ / AUROC↑) with ResNet-50 model on ImageNet-1k benchmark.

| $\varepsilon$ | Near-OOD | Far-OOD | $\lambda$ | Near-OOD | Far-OOD | $k_1$ | Near-OOD | Far-OOD | $k_2$ | Near-OOD | Far-OOD | $o$ | Near-OOD | Far-OOD |
|---|---|---|---|---|---|---|---|---|---|---|---|---|---|---|
| 0.1 | 63.76 / 77.50 | 19.26 / 95.85 | 1 | 67.49 / 75.45 | 20.41 / 95.54 | 1% | **58.97 / 78.98** | **17.84 / 96.14** | 1% | 59.08 / 78.56 | 18.35 / 96.02 | 1% | 61.77 / 78.29 | 19.28 / 95.77 |
| 0.5 | **58.97 / 78.98** | **17.84 / 96.14** | 10 | **58.97 / 78.98** | **17.84 / 96.14** | 10% | 60.40 / 77.89 | 19.72 / 95.63 | 5% | **58.97 / 78.98** | **17.84 / 96.14** | 5% | **58.97 / 78.98** | **17.84 / 96.14** |
| 1.0 | 61.60 / 76.96 | 19.31 / 95.84 | 100 | 59.03 / 78.10 | 19.39 / 95.80 | 100% | 60.99 / 76.49 | 21.88 / 94.96 | 100% | 62.43 / 75.44 | 23.05 / 94.69 | 100% | 67.37 / 73.08 | 22.93 / 95.01 |

Table 8: Robustness to Gaussian blur, JPEG compression, and adversarial training.

| Method | Gaussian blur | | JPEG compression | | Adversarial training | |
|---|---|---|---|---|---|---|
| | Near-OOD | Far-OOD | Near-OOD | Far-OOD | Near-OOD | Far-OOD |
| SCALE | 72.68 / 70.15 | 29.65 / 92.72 | 69.73 / 71.84 | 27.46 / 93.56 | 66.99 / 76.10 | 24.77 / 94.40 |
| OptFS | 71.84 / 69.66 | 30.27 / 92.24 | 69.43 / 71.12 | 28.34 / 92.95 | 70.11 / 73.51 | 28.32 / 93.42 |
| **AdaSCALE-A** | **67.53 / 75.18** | **28.11 / 93.31** | **61.24 / 77.59** | **25.87 / 94.06** | **60.21 / 80.68** | **21.13 / 95.35** |

**Robustness to real-world image corruptions.** To evaluate the practical robustness of our method, we assess its performance when input images are subjected to common real-world corruptions. We compare AdaSCALE against SCALE and OptFS on the ImageNet-1k dataset using a ResNet-50, applying Gaussian blur (kernel=5, sigma=(0.01, 0.5)) and JPEG compression (50%) to the input images. The results (FPR@95↓/AUROC↑), presented in Table 8 without any hyperparameter re-tuning, show that AdaSCALE's performance advantage is maintained even under these conditions.

**Robustness to adversarial training.** We evaluate AdaSCALE's performance on adversarially trained model ($\epsilon$=0.05) using the exact same hyperparameters as for the standard ResNet-50. The results (FPR@95↓ / AUROC↑) in Table 8 show that AdaSCALE outperforms baseline methods (SCALE and OptFS). This highlights the robustness of AdaSCALE across diverse training schemes.

**ID statistics.** With the rise of large models, where training data is often undisclosed or inaccessible, relying on full training ID datasets for OOD detection has become increasingly impractical. We rigorously assess AdaSCALE's effectiveness with limited data by conducting experiments on ImageNet-1k with ResNet-50 using $n_{val}$ ID samples to compute ID statistics, where $n_{val} \in \{10, 100, 1000, 5000\}$. The results in Table 9 confirm that even with substantially restricted access to ID data, AdaSCALE-A achieves state-of-the-art performance.

Table 9: AdaSCALE-A with restricted access to ID data using ResNet-50 network in FPR@95 ↓ / AUROC ↑ format.

| $n_{val}$ | **Near-OOD** | **Far-OOD** |
|---|---|---|
| 10 | 59.69 / 78.52 | 18.25 / 96.03 |
| 100 | 59.05 / 78.92 | **17.79** / 96.13 |
| 1000 | 58.99 / 78.95 | 17.86 / 96.13 |
| 5000 | **58.97 / 78.98** | 17.84 / **96.14** |

**Latency.** AdaSCALE incurs extra forward pass to compute perturbed activation $\mathbf{a}^\epsilon$. Also, top-k operations (time complexity: $\mathcal{O}(D \log D)$) are applied to $Q$ and $C_o$ to estimate OODness. Comparing variable vs. fixed

Table 10: Latency with fixed vs. variable percentile.

| | $D = 128$ | $D = 512$ | $D = 1024$ | $D = 2048$ | $D = 3024$ |
|---|---|---|---|---|---|
| Fixed percentile (SCALE/LTS) | $33\mu s$ | $40\mu s$ | $45\mu s$ | $48\mu s$ | $54\mu s$ |
| Variable percentile (AdaSCALE) | $152\mu s$ | $149\mu s$ | $155\mu s$ | $152\mu s$ | $164\mu s$ |
| Latency ratio (AdaSCALE / SCALE) | 4.66 | 3.76 | 3.42 | 3.14 | 3.02 |

percentiles for scaling in Table 10 over 10,000 trials, we observe that variable percentiles induce higher latency, though the latency ratio decreases with higher-dimensional activation spaces. While SCALE and OptFS have similar latency, AdaSCALE's is 2.91x higher with gradient attribution for trivial pixel perturbation, dropping to 1.56x with random pixel perturbation.

# 6 CONCLUSION

We propose **AdaSCALE**, a novel post-hoc OOD detection method that dynamically adjusts the scaling process based on a sample's estimated OODness. Leveraging the observation that OOD samples exhibit larger activation shifts under minor perturbations, AdaSCALE assigns stronger scaling to likely ID samples and weaker scaling to likely OOD samples, enhancing ID-OOD separability. AdaSCALE achieves state-of-the-art performance as well as generalization across architectures requiring negligibly few ID samples, making it highly practical for real-world deployment. It lays the foundation for future research to build upon the adaptive scaling mechanism.

# 7 BROADER IMPACTS

This work has positive impact in trustworthy deep learning by enabling detection of OOD samples. Reliable OOD detection can help prevent models from making overconfident predictions in high-stakes applications.

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

# Appendix

## A  NOTATIONS

Table 11 lists all the notations used in this paper.

Table 11: Table of Notations

| Notation | Meaning |
|---|---|
| $\mathcal{X}$ | Input space. |
| $\mathcal{Y}$ | Label space. |
| $C$ | Number of classes. |
| $C_{\text{in}}$ | Number of input channels. |
| $h$ | Classifier. |
| $\mathcal{P}_{\text{ID}}(x,y)$ | Underlying joint distribution of ID data. |
| $\mathcal{P}_{\text{OOD}}(x)$ | Distribution of OOD data. |
| $\mathcal{D}_{\text{ID}}$ | ID training dataset. |
| $N$ | Number of training samples. |
| $f_\theta$ | Feature extractor, parameterized by $\theta$. |
| $\mathcal{A}$ | Activation space. |
| $g_\mathcal{W}$ | Classifier (mapping activations to logits), parameterized by $\mathcal{W}$. |
| $\mathcal{Z}$ | Logit space. |
| $\mathbf{a}$ | Activation vector (output of $f_\theta(x)$). |
| $a_j$ | The $j$-th element of the activation vector $\mathbf{a}$. |
| $\mathbf{z}$ | Logit vector (output of $g_\mathcal{W}(\mathbf{a})$). |
| $\mathcal{L}$ | Loss function (e.g., cross-entropy). |
| $S(x)$ | OOD scoring function. |
| $\tau$ | Threshold for classifying an input as ID or OOD. |
| $x$ | Input image. |
| $x[c,h,w]$ | Channel value of input image $x$ at position $(c,h,w)$. |
| $H$ | Height of the input image. |
| $W$ | Width of the input image. |
| $AT(x,c,h,w)$ | Attribution function, assigning a score to each channel value of input $x$. |
| $o$ | Percent of channel values to perturb. |
| $R$ | Set of channel value indices with lowest absolute attribution scores. |
| $y_{\text{pred}}$ | Predicted class index. |
| $\varepsilon$ | Perturbation magnitude. |
| $x^\varepsilon$ | Perturbed input image. |
| $\mathbf{a}^\varepsilon$ | Activation vector of the perturbed input $x^\varepsilon$. |
| $\mathbf{a}^{\text{shift}}$ | Activation shift vector (absolute element-wise difference between $\mathbf{a}$ and $\mathbf{a}^\varepsilon$). |
| $k_1, k_2$ | Number of highest-magnitude activations considered for $Q$ and $C_o$, respectively. |
| $\text{argsort}(\mathbf{v})$ | Same as argsort($\mathbf{v}$, desc = True). Returns the indices that would sort the vector $\mathbf{v}$ in descending order. |
| $\max_k(\mathbf{v})$ | Returns the $k^{\text{th}}$ maximum element of vector $\mathbf{v}$. |
| $\mathbf{i}_1, \mathbf{i}_2$ | Index sets: $\mathbf{i}_1 = \text{argsort}(\mathbf{a}, \text{ desc} = \text{True})[:k_1]$, $\mathbf{i}_2 = \text{argsort}(\mathbf{a}, \text{ desc} = \text{True})[:k_2]$ |
| $Q$ | Sum of activation shifts for the top-$k_1$ activations. |
| $C_o$ | Correction term: sum of top-$k_2$ perturbed activations. |
| $\lambda$ | Weighting factor for $Q$ in the $Q'$ calculation. |
| $Q'$ | Estimated OOD likelihood. |
| $n_{\text{val}}$ | Number of ID validation samples. |
| $\bar{Q}_s$ | Set of $Q'$ values on the ID validation samples. |
| $F_{Q'}(Q')$ | Empirical cumulative distribution function (eCDF) of $Q'$ values. |
| $p_{\text{min}}, p_{\text{max}}$ | Minimum and maximum percentile thresholds. |
| $p_r$ | Raw ID likelihood from eCDF |
| $p$ | Adjusted percentile threshold |
| $P_p(\mathbf{a})$ | The $p$-th percentile value of all elements in $\mathbf{a}$ |
| $r$ | Scaling factor. |
| $\mathbf{a}_{\text{scaled}}$ | Scaled activation vector (AdaSCALE-A). |
| $\mathbf{z}_{\text{scaled}}$ | Scaled logit vector (AdaSCALE-L). |
| $\text{ReLU}(a_j)$ | Rectified Linear Unit activation function: $\text{ReLU}(a_j) = \max(0, a_j)$. |

# B  ALGORITHM

The algorithm for computing adaptive scaling factor $r$ is provided in Algorithm 1.

---

**Algorithm 1** Computing the Adaptive Scaling Factor

---

**Input:** Input sample $x$, perturbation magnitude $\varepsilon$, model $f_\theta$, hyperparameters $\lambda$, $k_1$, $k_2$, $p_{\min}$, $p_{\max}$, $\varepsilon$, $o$, precomputed empirical CDF $F_{Q'}$

**Output:** Scaling factor $r$

1:  // Extract features and compute activation shifts
2:  $\mathbf{a} \leftarrow f_\theta(x)$                               {Original activation}
3:  $\nabla_x z_c \leftarrow \frac{\partial g_W(f_\theta(x))_c}{\partial x}$    {Gradient for predicted class $c$}
4:  $R \leftarrow o\%$ of channel values with *lowest* $|\nabla_x z_c|$
5:  $x^\varepsilon \leftarrow x + \varepsilon \cdot \text{sign}(\nabla_x z_c) \cdot \mathbb{1}_R$ {Perturb selected regions}
6:  $\mathbf{a}^\varepsilon \leftarrow f_\theta(x^\varepsilon)$                        {Perturbed activation}
7:  $\mathbf{a}^{\text{shift}} \leftarrow |\mathbf{a}^\varepsilon - \mathbf{a}|$            {Compute activation shift}
8:  // Compute OOD likelihood estimate
9:  $\mathbf{i}_1 \leftarrow \text{argsort}(\mathbf{a}, \ \text{desc} = \text{True})[: k_1]$
10:  $Q \leftarrow \sum_{i \in \mathbf{i}_1} a_i^{\text{shift}}$           {Shift in top activations}
11:  $\mathbf{i}_2 \leftarrow \text{argsort}(\mathbf{a}, \ \text{desc} = \text{True})[: k_2]$
12:  $C_o \leftarrow \sum_{i \in \mathbf{i}_2} \text{ReLU}(a_i^\varepsilon)$        {Correction term}
13:  $Q' \leftarrow \lambda \cdot Q + C_o$        {OOD likelihood estimate}
14:  // Compute adaptive percentile
15:  $p_r \leftarrow (1 - F_{Q'}(Q'))$    {raw ID likelihood from eCDF}
16:  $p \leftarrow p_{\min} + p_r \cdot (p_{\max} - p_{\min})$    {Adjusted percentile}
17:  // Compute scaling factor
18:  $P_p(\mathbf{a}) \leftarrow$ the $p$-th percentile value of all elements in $\mathbf{a}$
19:  $r \leftarrow \sum_j a_j / \sum_{a_j > P_p(\mathbf{a})} a_j$      {Final scaling factor}
20:  **return** $r$

---

# C  ADDITIONAL OBSERVATION IN ACTIVATION SPACE.

Figure 5 shows perturbed activations $\mathbf{a}^\varepsilon$ are, on average, higher for ID samples than for OOD samples.

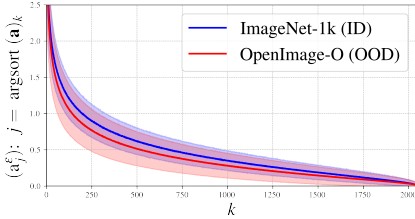

Figure 5: Perturbed activation magnitudes comparison between ID and OOD samples. ID samples consistently maintain higher average activation values in comparison to OOD samples.

# D  ADDITIONAL EXPERIMENTS AND ANALYSIS

## D.1  SUPERIORITY OF THE PROPOSED OODNESS METRIC ($Q'$)

To validate our choice of the OODness metric, we conducted an extensive ablation study comparing the performance of our proposed metric, $Q'$ against $Q$ alone. The results, summarized in Table 12, demonstrate that using $Q'$ provides a consistent and significant performance improvement over using $Q$ alone. This advantage holds across all eight diverse architectures.

Table 12: Comparison of OOD detection performance (FPR95↓ / AUROC↑) using the baseline OODness metric ($Q$) versus our proposed metric ($Q'$). The results show a clear advantage for $Q'$ across a wide range of model architectures.

| Category | Metric | ResNet-50 | ResNet-101 | RegNet-Y-16 | ResNeXt-50 | DenseNet-201 | EfficientNetV2-L | ViT-B-16 | Swin-B | Average |
|---|---|---|---|---|---|---|---|---|---|---|
| near-OOD | $Q$ | 59.65/77.82 | 56.67/81.65 | 60.63/86.00 | 63.99/79.34 | 68.65/78.03 | 73.18/80.17 | 82.61/69.31 | 75.85/72.26 | 68.90/78.07 |
| | $Q'$ | **58.98/78.98** | **57.96/81.68** | **47.91/89.18** | **64.14/79.96** | **61.28/79.66** | **53.78/86.94** | **71.87/73.14** | **73.41/74.48** | **61.17/80.50** |
| far-OOD | $Q$ | 19.82/95.72 | 20.32/95.50 | 31.53/93.94 | 26.87/94.32 | 35.55/92.19 | 66.20/84.60 | 77.52/76.39 | 63.20/80.78 | 42.63/89.18 |
| | $Q'$ | **17.84/96.14** | **18.51/95.95** | **21.37/95.84** | **22.08/95.24** | **28.01/93.23** | **37.61/91.48** | **47.63/86.83** | **47.81/87.14** | **30.11/92.73** |

For instance, if an OOD sample gets a very high $Q$ score, it will produce a very large scaling factor. This can amplify the logits so much that it washes out the subtle but important information contained within the values of the logits themselves. By adding the opposing term, we "tame" these potentially overconfident scaling factors. This ensures a healthier balance where both the feature-level instability (captured by $Q$) and the original logit distribution contribute to the final OOD score.

Illustrative Example: A simple toy example demonstrates why this balancing is critical.

Logits: Let $\text{logit}_{\text{ID}} = (1, 6, 2)$ and $\text{logit}_{\text{OOD}} = (1, 1, 1)$.

Case 1: Overconfident Scaling (using only $Q$)

Let's say this yields scaling factors $r_{\text{ID}} = 2$ and $r_{\text{OOD}} = 5$. Assume a threshold of 25, where energy > 25 is ID and energy < 25 is OOD (We deal with magnitude for simplicity).

$$\text{energy}_{\text{ID}} = \log(e^{1 \cdot 2^2} + e^{6 \cdot 2^2} + e^{2 \cdot 2^2}) = 24.0$$

$$\text{energy}_{\text{OOD}} = \log(e^{1 \cdot 5^2} + e^{1 \cdot 5^2} + e^{1 \cdot 5^2}) = 26.1$$

Here, $\text{energy}_{\text{OOD}} > \text{energy}_{\text{ID}}$, leading to an **OOD detection failure**.

Case 2: Balanced Scaling (using $Q'$ to temper the scaling)

This leads to less extreme scaling, e.g., $r_{\text{ID}} = 1$ and $r_{\text{OOD}} = 2$. Assume a threshold of 5, where energy > 5.5 is ID and energy < 5.5 is OOD (We deal with magnitude for simplicity).

$$\text{energy}_{\text{ID}} = \log(e^{1 \cdot 1^2} + e^{6 \cdot 1^2} + e^{2 \cdot 1^2}) = 6.0$$

$$\text{energy}_{\text{OOD}} = \log(e^{1 \cdot 2^2} + e^{1 \cdot 2^2} + e^{1 \cdot 2^2}) = 5.1$$

Here, $\text{energy}_{\text{ID}} > \text{energy}_{\text{OOD}}$, leading to a **successful OOD detection**.

## D.2  COMPARISON WITH ADDITIONAL BASELINES

Table 13: Comparison with distance-based OOD detection methods on ImageNet-1k using a ResNet-50 backbone. AdaSCALE-A significantly outperforms the distance-based baselines.

| Method | Near-OOD | | Far-OOD | |
|---|---|---|---|---|
| | FPR@95 ↓ | AUROC ↑ | FPR@95 ↓ | AUROC ↑ |
| LINe | 79.10 | 67.20 | 32.95 | 93.31 |
| NNGuide | 72.46 | 68.09 | 21.40 | 95.33 |
| MDS | 76.27 | 64.53 | 70.82 | 66.22 |
| **AdaSCALE-A** | **58.97** | **78.98** | **17.84** | **96.14** |

To situate AdaSCALE within the broader landscape of OOD detection techniques, we compare its performance against additional baselines. The experiments were conducted on the large-scale ImageNet-1k benchmark using a ResNet-50 network. As shown in Table 13, AdaSCALE demonstrates a significant performance advantage over all these methods.

**ATS vs AdaSCALE:** We provide a more in-depth comparison between ATS and AdaSCALE below:

- **Number of passes:** AdaSCALE uses two forward passes (on the original and perturbed input) to leverage an additional independent OOD detection signal ($Q'$) in its scaling mechanism. In contrast, ATS uses only one forward pass.
- **eCDF Usage:** ATS relies on multiple eCDFs, building one for each selected intermediate layer. AdaSCALE constructs a single eCDF from its final OODness score ($Q'$) in the last layer only. We hypothesize that the final layer is most effective for this, as deeper layers are responsible for semantic discrimination.
- **Role of eCDF:** In ATS, eCDFs are a core part of the inference pipeline, transforming raw activations into p-values that are aggregated to compute the final scaling factor. In AdaSCALE, the single eCDF plays a more indirect, regulatory role: it calibrates the OODness score to determine an adaptive percentile, which then informs the final scaling factor calculation.
- **Data Requirement for eCDF:** AdaSCALE requires only a very small number of ID samples (as few as 5-10). ATS requires access to the whole training dataset to reliably compute eCDFs for each intermediate layer.

### D.3 ANALYSIS ON ADVERSARIALLY TRAINED MODELS

We investigate the effectiveness of our approach on adversarially trained models to test its robustness. This analysis is twofold: first, we verify that the core assumption of our method holds, and second, we evaluate its OOD detection performance.

### D.3.1 ACTIVATION SENSITIVITY IN ADVERSARIALLY TRAINED NETWORKS

The effectiveness of our method relies on the activation shift ratio ($Q_{OOD}/Q_{ID}$) being greater than 1 for OOD samples compared to ID samples. We measured this ratio by taking the top-1% activation shifts on a ResNet-50 model adversarially trained on ImageNet-1k. The results in Table 14 confirm that the ratio remains consistently greater than 1 across various OOD datasets. This demonstrates that OOD samples still produce a larger activation shift than ID samples, even in adversarially robust models, validating our method's applicability.

Table 14: Activation shift ratio ($Q_{OOD}/Q_{ID}$) on a ResNet-50 model adversarially trained on ImageNet-1k. The ratio remaining greater than 1 confirms that the fundamental signal required for our method persists.

| **OOD Dataset** | $Q_{OOD}/Q_{ID}$ ($\epsilon = 0.01$) | $Q_{OOD}/Q_{ID}$ ($\epsilon = 0.05$) |
|---|---|---|
| SSB-hard | 1.78 | 1.73 |
| NINCO | 1.29 | 1.28 |
| ImageNet-O | 1.69 | 1.72 |
| OpenImage-O | 1.48 | 1.45 |
| iNaturalist | 1.14 | 1.14 |
| Textures | 1.33 | 1.18 |
| Places | 1.30 | 1.16 |

### D.3.2 PERFORMANCE OF ADASCALE ON ADVERSARIALLY TRAINED NETWORKS

We evaluate AdaSCALE's OOD detection performance on these adversarially trained models using the exact same hyperparameters as for the standard ResNet-50. The results in Table 15 show that AdaSCALE consistently outperforms baseline methods. This highlights the robustness of our

approach AdaSCALE, which remains superior even when applied to models with different training schemes without any specific hyperparameter tuning.

Table 15: OOD detection performance on adversarially trained ResNet-50 models. AdaSCALE consistently outperforms the baselines, demonstrating its robustness.

| Training | Method | Near-OOD | | Far-OOD | |
|---|---|---|---|---|---|
| | | FPR@95 ↓ | AUROC ↑ | FPR@95 ↓ | AUROC ↑ |
| $\epsilon = 0.01$ | SCALE | 68.54 | 75.43 | 25.95 | 94.40 |
| | OptFS | 70.36 | 72.97 | 28.60 | 93.24 |
| | **AdaSCALE-A** | **60.73** | **80.84** | **19.90** | **95.74** |
| $\epsilon = 0.05$ | SCALE | 66.99 | 76.10 | 24.77 | 94.40 |
| | OptFS | 70.11 | 73.51 | 28.32 | 93.42 |
| | **AdaSCALE-A** | **60.21** | **80.68** | **21.13** | **95.35** |

## D.4 ROBUSTNESS TO REAL-WORLD IMAGE CORRUPTIONS

To evaluate the practical robustness of our method, we assess its performance when input images are subjected to common real-world corruptions. We benchmark AdaSCALE against SCALE and OptFS on the ImageNet-1k dataset using a ResNet-50, applying Gaussian blur and JPEG compression to the input images. The results, presented without any hyperparameter re-tuning, show that AdaSCALE's performance advantage is maintained even under these challenging conditions.

Table 16: OOD detection performance with Gaussian blur (kernel size=5, sigma=(0.1, 2.0)).

| Method | Near-OOD | | Far-OOD | |
|---|---|---|---|---|
| | FPR@95 ↓ | AUROC ↑ | FPR@95 ↓ | AUROC ↑ |
| SCALE | 72.68 | 70.15 | 29.65 | 92.72 |
| OptFS | 71.84 | 69.66 | 30.27 | 92.24 |
| **AdaSCALE-A** | **67.53** | **75.18** | **28.11** | **93.31** |

Table 17: OOD detection performance with 50% JPEG compression.

| Method | Near-OOD | | Far-OOD | |
|---|---|---|---|---|
| | FPR@95 ↓ | AUROC ↑ | FPR@95 ↓ | AUROC ↑ |
| SCALE | 69.73 | 71.84 | 27.46 | 93.56 |
| OptFS | 69.43 | 71.12 | 28.34 | 92.95 |
| **AdaSCALE-A** | **61.24** | **77.59** | **25.87** | **94.06** |

# E SENSITIVITY STUDY

## E.1 SENSITIVITY STUDY OF $\varepsilon$

The sensitivity study of $\varepsilon$ presented at Table 18 suggests the optimal value of $\varepsilon$ to be around 0.5.

## E.2 SENSITIVITY STUDY OF $\lambda$

The sensitivity study of $\lambda$ presented at Table 18 suggests the optimal value of $\lambda$ to be around 10.

## E.3 SENSITIVITY STUDY OF $k_1$

The sensitivity study of $k_1$ presented at Table 20 suggests the optimal value of $k_1$ to be around 1%.

Table 18: Sensitivity study of $\varepsilon$ with ResNet-50 model on ImageNet-1k benchmark.

| $\varepsilon$ | Near-OOD | | Far-OOD | |
|---|---|---|---|---|
| | FPR@95 $\downarrow$ | AUROC $\uparrow$ | FPR@95 $\downarrow$ | AUROC $\uparrow$ |
| 0.1 | 63.76 | 77.50 | 19.26 | 95.85 |
| 0.5 | **58.97** | **78.98** | **17.84** | **96.14** |
| 1.0 | 61.60 | 76.96 | 19.31 | 95.84 |

Table 19: Sensitivity study of $\lambda$ with ResNet-50 model on ImageNet-1k benchmark.

| $\lambda$ | Near-OOD | | Far-OOD | |
|---|---|---|---|---|
| | FPR@95 $\downarrow$ | AUROC $\uparrow$ | FPR@95 $\downarrow$ | AUROC $\uparrow$ |
| 0.1 | 70.07 | 74.12 | 21.40 | 95.33 |
| 1 | 67.49 | 75.45 | 20.41 | 95.54 |
| 10 | **58.97** | **78.98** | **17.84** | **96.14** |
| 100 | 59.03 | 78.10 | 19.39 | 95.80 |

Table 20: Sensitivity study of $k_1$ with ResNet-50 model on ImageNet-1k benchmark.

| $k_1$ | Near-OOD | | Far-OOD | |
|---|---|---|---|---|
| | FPR@95 $\downarrow$ | AUROC $\uparrow$ | FPR@95 $\downarrow$ | AUROC $\uparrow$ |
| 1% | **58.97** | **78.98** | **17.84** | **96.14** |
| 5% | 60.36 | 77.74 | 19.95 | 95.63 |
| 10% | 60.40 | 77.89 | 19.72 | 95.63 |
| 50% | 60.34 | 77.25 | 20.90 | 95.25 |
| 100% | 60.99 | 76.49 | 21.88 | 94.96 |

## E.4 SENSITIVITY STUDY OF $k_2$

The sensitivity study of $k_2$ presented at Table 21 suggests the optimal value of $k_2$ to be around 5%.

Table 21: Sensitivity study of $k_2$ with ResNet-50 model on ImageNet-1k benchmark.

| $k_2$ | Near-OOD | | Far-OOD | |
|---|---|---|---|---|
| | FPR@95 $\downarrow$ | AUROC $\uparrow$ | FPR@95 $\downarrow$ | AUROC $\uparrow$ |
| 1% | 59.08 | 78.56 | 18.35 | 96.02 |
| 5% | **58.97** | **78.98** | **17.84** | **96.14** |
| 10% | 59.41 | 78.65 | 18.33 | 95.96 |
| 100% | 62.43 | 75.44 | 23.05 | 94.69 |

## E.5 SENSITIVITY STUDY OF $o$

We present the complete results of image perturbation study (FPR@95 $\downarrow$ / AUROC $\uparrow$) in Table 22.

Table 22: Image perturbation study with ResNet-50 model on ImageNet-1k benchmark.

| Pixel type | $o\%$ | OOD Detection | | FS-OOD Detection | |
| --- | --- | --- | --- | --- | --- |
| | | Near-OOD | Far-OOD | Near-OOD | Far-OOD |
| Random | 1% | 61.73 / 78.15 | 19.44 / 95.74 | 83.19 / 48.59 | 53.45 / 74.10 |
| | 5% | 59.97 / 78.67 | 18.14 / 96.06 | 81.92 / 49.19 | 52.35 / 74.84 |
| | 10% | 60.27 / 78.02 | 18.45 / 96.00 | 82.07 / 48.62 | 52.84 / 74.70 |
| | 50% | 62.81 / 76.27 | 19.95 / 95.70 | 83.40 / 46.55 | 54.94 / 73.42 |
| Trivial | 1% | 61.77 / 78.29 | 19.28 / 95.77 | 82.92 / 48.86 | 53.34 / 74.24 |
| | 5% | **58.97 / 78.98** | **17.84 / 96.14** | **81.52 / 49.35** | **52.33 / 74.89** |
| | 10% | 60.24 / 78.17 | 17.94 / 96.08 | 82.19 / 48.59 | 52.59 / 74.77 |
| | 50% | 66.43 / 74.10 | 21.58 / 95.29 | 85.26 / 44.86 | 56.56 / 72.82 |
| Salient | 1% | 69.43 / 75.13 | 22.62 / 95.17 | 85.59 / 48.32 | 53.76 / 75.23 |
| | 5% | 67.31 / 75.78 | 21.24 / 95.44 | 85.07 / 48.40 | 53.26 / 75.63 |
| | 10% | 65.65 / 76.20 | 20.39 / 95.62 | 84.36 / 48.33 | 53.17 / 75.55 |
| | 50% | 64.54 / 75.39 | 20.48 / 95.61 | 83.78 / 47.07 | 54.11 / 74.54 |
| All | 100% | 67.37 / 73.08 | 22.93 / 95.01 | 85.66 / 44.00 | 57.80 / 72.19 |

## E.6 SENSITIVITY STUDY OF $p_{\text{MIN}}$

We present sensitivity study (FPR@95 $\downarrow$ / AUROC $\uparrow$) of $p_{\min}$, setting $p_{\max}$ to 85 in the Table 23.

Table 23: Sensitivity study of $p_{\min}$ with ResNet-50 model on ImageNet-1k benchmark.

| $p_{\min}$ | 70 | 75 | 80 | 85 |
| --- | --- | --- | --- | --- |
| Near-OOD | 62.54 / 79.60 | 58.57 / 81.68 | **58.98 / 78.98** | 67.76 / 74.20 |
| Far-OOD | 21.62 / 95.19 | 19.86 / 95.62 | **17.84 / 96.14** | 21.44 / 95.39 |

## E.7 JOINT SENSITIVITY STUDY

We present joint sensitivity study in terms of near-OOD detection among three hyperparameters ($\lambda$, $p_{\max}, p_{\min}$) below:

Table 24: Joint sensitivity study of $k_1$ and $\lambda$ (setting $p_{\max} = 85\%$) with ResNet-50 model on ImageNet-1k benchmark.

| $k_1 \setminus \lambda$ | 0.1 | 1 | 10 | 100 |
| --- | --- | --- | --- | --- |
| 1% | 70.07 / 74.12 | 67.49 / 75.45 | **58.97 / 78.98** | 59.03 / 78.10 |
| 5% | 69.53 / 74.33 | 63.87 / 76.84 | 60.36 / 77.74 | 61.12 / 77.22 |
| 10% | 69.34 / 74.50 | 62.63 / 77.33 | 60.40 / 77.89 | 61.55 / 76.92 |
| 50% | 67.49 / 75.32 | 61.26 / 77.26 | 60.34 / 77.25 | 62.05 / 76.52 |
| 100% | 65.95 / 75.77 | 61.18 / 76.97 | 62.17 / 76.39 | 62.43 / 76.31 |

The results in Table 24 clearly indicate that the $\lambda$ hyperparameter exerts a more significant influence on the final OOD detection performance. Furthermore, a key relationship emerges: for optimal performance, a decrease in $\lambda$ must be compensated by an increase in $Q$ (and consequently, a higher $k_1$). This is explained by the formulation $Q' = \lambda \cdot Q + C_o$, where a smaller $\lambda$ necessitates a larger $Q$ value to ensure the term $\lambda \cdot Q$ dominates the regularizer $C_o$.

In Table 25, when $\lambda$ is set to 0.1 with $k_1 = 1\%$, $\lambda \cdot Q$ does not dominate over $C_o$ term. As a result, it leads to scaling based on the regularization / balancing term instead of "predetermined" actual OOD detection signal "activation shift at peak activations". Such value of $\lambda$ (e.g., $\lambda = 0.1$) invalidates the proposed hypothesis.

Table 25: Joint sensitivity study of $\lambda$ and $p_{\max}$ (setting $k_1 = 1\%$) with ResNet-50 model on ImageNet-1k benchmark.

| $\lambda \setminus p_{\max}$ | 85 | 90 | 95 |
|---|---|---|---|
| 0.1 | 70.07 / 74.12 | 87.45 / 66.06 | 91.81 / 56.09 |
| 1 | 67.49 / 75.45 | 84.31 / 69.32 | 89.41 / 60.11 |
| 10 | **58.97 / 78.98** | 69.47 / 79.68 | 78.77 / 76.26 |
| 100 | 59.03 / 78.10 | 63.19 / 79.67 | 71.43 / 78.51 |

From the results in Table 26, it can be observed that activation shift with all activations considered ($k_1 = 100\%$) still proves to be a useful OOD detection signal.

Furthermore, previous works (ASH, SCALE) observed that, particularly for ResNet-50 architecture, the performance drops after the percentile value of 85%. Similar observations can also be made in AdaSCALE from the results in Tables 25 and 26.

Table 26: Joint sensitivity study of $k_1$ and $p_{\max}$ (setting $\lambda = 10$) with ResNet-50 model on ImageNet-1k benchmark.

| $k_1 \setminus p_{\max}$ | 85 | 90 | 95 |
|---|---|---|---|
| 1% | **58.97 / 78.98** | 69.49 / 79.67 | 78.77 / 76.26 |
| 5% | 60.36 / 77.74 | 68.30 / 78.23 | 78.19 / 75.58 |
| 10% | 60.40 / 77.89 | 68.46 / 77.46 | 78.90 / 74.70 |
| 50% | 60.34 / 77.25 | 69.88 / 76.39 | 80.46 / 73.17 |
| 100% | 62.17 / 76.39 | 70.13 / 75.93 | 81.22 / 72.32 |

# F  ISH REGULARIZATION:

Apart from enhancing the prior postprocessor ASH (Djurisic et al., 2023), SCALE (Xu et al., 2015) introduces a training regularization to emphasize samples with more distinct ID characteristics. We assess the performance (FPR@95 ↓ / AUROC ↑) of each method in ResNet-50 and ResNet-101 model following this regularization in Table 27. The results indicate that AdaSCALE maintains a substantial advantage, surpassing the second-best method, SCALE, by **12.56%**/**5.82%** and **20.46%**/**1.21%** in FPR@95 / AUROC for near- and far-OOD detection in ResNet-50, respectively. Moreover, AdaS-CALE demonstrates superior performance beyond conventional OOD detection, with corresponding improvements of **4.10%/5.87%** and **9.70%/1.12%** in full-spectrum setting. Furthermore, ISH regularization further amplifies the performance gap between AdaSCALE-A and OptFS, enhancing the near-OOD detection improvement from 12.96% / 6.44% to **15.18%/14.83%**. These findings also generalize to ResNet-101 network.

Table 27: OOD detection results on ImageNet-1k benchmark with ISH (Xu et al., 2024) regularization.

| Method | OOD Detection | | FS-OOD Detection | |
| --- | --- | --- | --- | --- |
| | **Near-OOD** | **Far-OOD** | **Near-OOD** | **Far-OOD** |
| ResNet-50 | | | | |
| MSP | 74.07 / 62.16 | 51.13 / 84.64 | 87.52 / 40.36 | 74.41 / 61.52 |
| MLS | 74.38 / 66.43 | 41.57 / 88.90 | 88.89 / 39.49 | 71.53 / 61.69 |
| EBO | 74.68 / 66.46 | 41.85 / 88.83 | 89.05 / 39.18 | 71.77 / 61.11 |
| ReAct | 71.98 / 70.81 | 28.76 / 93.49 | 87.78 / 43.88 | 61.87 / 71.64 |
| ASH | 67.99 / 73.46 | 23.88 / 94.67 | 85.74 / 45.29 | 57.81 / 72.81 |
| SCALE | 65.68 / 76.41 | 20.77 / 95.62 | 84.31 / 48.40 | 54.48 / 74.79 |
| BFAct | 71.59 / 70.85 | 28.38 / 93.50 | 87.51 / 43.95 | 61.39 / 71.43 |
| LTS | 66.32 / 75.03 | 22.07 / 95.28 | 85.08 / 46.16 | 57.11 / 73.06 |
| OptFS | 67.71 / 73.03 | 24.65 / 94.18 | 85.38 / 45.91 | 57.09 / 72.95 |
| **AdaSCALE-A** | 57.43 / **80.86** | **16.52 / 96.46** | **80.85 / 51.24** | **51.57 / 75.63** |
| **AdaSCALE-L** | **56.83** / 80.81 | 17.62 / 96.22 | 80.97 / 50.59 | 53.43 / 74.62 |
| ResNet-101 | | | | |
| MSP | 71.39 / 68.31 | 51.00 / 84.81 | 85.70 / 45.72 | 73.69 / 62.79 |
| MLS | 72.32 / 71.94 | 41.04 / 88.99 | 87.39 / 44.88 | 69.94 / 63.23 |
| EBO | 72.78 / 71.92 | 41.45 / 88.87 | 87.64 / 44.66 | 70.23 / 62.65 |
| ReAct | 67.74 / 75.74 | 28.53 / 93.47 | 85.40 / 48.83 | 60.50 / 72.18 |
| ASH | 66.03 / 77.79 | 25.21 / 94.43 | 83.95 / 50.57 | 56.91 / 73.37 |
| SCALE | 64.30 / 78.98 | 23.09 / 94.95 | 83.14 / 51.05 | 55.88 / 73.27 |
| BFAct | 67.53 / 75.86 | 28.32 / 93.40 | 85.11 / 48.96 | 60.12 / 71.83 |
| LTS | 66.32 / 75.03 | 22.07 / 95.28 | 85.08 / 46.16 | 57.11 / 73.06 |
| OptFS | 67.71 / 73.03 | 24.65 / 94.18 | 85.38 / 45.91 | 57.09 / 72.95 |
| **AdaSCALE-A** | 54.66 / **83.52** | **16.81 / 96.32** | **78.52 / 55.19** | **49.92 / 76.20** |
| **AdaSCALE-L** | **53.91** / 83.49 | 17.55 / 96.15 | 78.63 / 54.60 | 51.47 / 75.47 |

# G  HYPERPARAMETERS

We present final hyperparameter values of AdaSCALE-A and AdaSCALE-L in Table 28 and Table 29.

Table 28: Hyperparameters $(p_{\min}, p_{\max})$ used for each dataset and network for AdaSCALE-A.

| Dataset | Network | $p_{\min}$ | $p_{\max}$ |
|---|---|---|---|
| CIFAR-10 | WideResNet-28-10 | 60 | 95 |
| | DenseNet-101 | 65 | 90 |
| CIFAR-100 | WideResNet-28-10 | 60 | 85 |
| | DenseNet-101 | 70 | 80 |
| ImageNet-1k | ResNet-50 | 80 | 85 |
| | ResNet-101 | 80 | 85 |
| | RegNet-Y-16 | 60 | 90 |
| | ResNeXt-50 | 80 | 85 |
| | DenseNet-201 | 90 | 95 |
| | EfficientNetV2-L | 60 | 99 |
| | Vit-B-16 | 60 | 85 |
| | Swin-B | 90 | 99 |

Table 29: Hyperparameters $(p_{\max}, p_{\min})$ used for each dataset and network for AdaSCALE-L.

| Dataset | Network | $p_{\min}$ | $p_{\max}$ |
|---|---|---|---|
| CIFAR-10 | WideResNet-28-10 | 60 | 85 |
| | DenseNet-101 | 70 | 85 |
| CIFAR-100 | WideResNet-28-10 | 60 | 80 |
| | DenseNet-101 | 65 | 75 |
| ImageNet-1k | ResNet-50 | 80 | 85 |
| | ResNet-101 | 70 | 80 |
| | RegNet-Y-16 | 60 | 85 |
| | ResNeXt-50 | 70 | 80 |
| | DenseNet-201 | 90 | 95 |
| | EfficientNetV2-L | 60 | 99 |
| | Vit-B-16 | 75 | 85 |
| | Swin-B | 90 | 99 |

# H CIFAR-RESULTS

## H.1 WRN-28-10

Table 30: Far-OOD detection results (FPR@95↓ / AUROC↑) on CIFAR-10 and CIFAR-100 benchmarks using the WRN-28-10 network, averaged over 3 trials. The overall average performance is reported. The best results are **bold**, and the second-best results are underlined.

| | *CIFAR-10 benchmark* | | | | |
|---|---|---|---|---|---|
| Method | **MNIST** | **SVHN** | **Textures** | **Places365** | **Average** |
| MSP | 17.02 / 94.61 | 21.71 / 92.96 | 60.50 / 88.06 | 42.27 / 90.04 | 35.38 / 91.42 |
| MLS | 13.01 / 96.76 | 30.35 / 93.06 | 76.12 / 86.65 | 52.56 / 90.46 | 43.01 / 91.73 |
| EBO | **12.93 / 96.93** | 30.35 / 93.12 | 76.15 / 86.68 | 52.57 / 90.56 | 43.00 / 91.82 |
| ReAct | 15.50 / 96.30 | 34.01 / 92.47 | 57.76 / 88.77 | 57.33 / 89.66 | 41.15 / 91.80 |
| ASH | 50.11 / 88.80 | 89.90 / 74.76 | 95.07 / 72.91 | 92.22 / 70.06 | 81.82 / 76.63 |
| SCALE | 13.24 / 96.70 | 32.21 / 92.88 | 75.76 / 86.77 | 55.81 / 90.09 | 44.26 / 91.61 |
| BFAct | 25.79 / 94.64 | 43.08 / 91.10 | **57.16 / 88.80** | 61.00 / 88.32 | 46.75 / 90.71 |
| LTS | 14.04 / 96.60 | 39.85 / 92.21 | 76.85 / 86.43 | 63.13 / 89.19 | 48.47 / 91.11 |
| OptFS | 25.68 / 94.83 | 51.58 / 89.86 | 62.14 / 88.07 | 80.19 / 84.05 | 54.90 / 89.20 |
| **AdaSCALE-A** | 14.93 / 96.02 | **17.84 / 95.14** | 64.96 / 88.31 | **34.57 / 92.31** | **33.08 / 92.95** |
| **AdaSCALE-L** | 15.58 / 95.98 | 18.41 / 95.10 | 62.87 / 88.67 | 37.59 / 91.97 | 33.61 / 92.93 |
| | *CIFAR-100 benchmark* | | | | |
| | **MNIST** | **SVHN** | **Textures** | **Places365** | **Average** |
| MSP | 49.79 / 78.72 | 56.76 / 80.70 | 64.49 / 76.86 | 56.66 / 79.96 | 56.92 / 79.06 |
| MLS | 46.57 / 81.43 | 53.08 / 83.37 | 64.59 / 77.65 | 59.70 / 79.82 | 55.99 / 80.57 |
| EBO | 46.41 / 81.99 | 52.92 / 83.77 | 64.58 / 77.61 | 59.76 / 79.60 | 55.92 / 80.74 |
| ReAct | 49.92 / 81.07 | 40.66 / 86.49 | 52.42 / 80.81 | 60.35 / 79.72 | 50.84 / 82.03 |
| ASH | 44.06 / 85.55 | 41.48 / 87.50 | 61.78 / 81.65 | 80.45 / 71.83 | 56.94 / 81.63 |
| SCALE | 40.65 / 84.68 | 48.56 / 85.56 | 58.45 / 80.81 | 60.51 / 79.90 | 52.04 / 82.74 |
| BFAct | 61.59 / 77.47 | 34.74 / 88.50 | **47.30 / 83.38** | 64.49 / 78.47 | 52.03 / 81.96 |
| LTS | **36.27 / 87.38** | 45.41 / 87.23 | 53.90 / 83.18 | 62.62 / 79.64 | 49.55 / 84.36 |
| OptFS | 57.61 / 79.47 | 37.04 / 86.43 | 53.02 / 80.43 | 70.44 / 76.77 | 54.53 / 80.78 |
| **AdaSCALE-A** | 45.18 / 81.69 | 36.79 / 89.20 | 55.93 / 81.93 | **56.48 / 81.55** | 48.59 / 83.59 |
| **AdaSCALE-L** | 42.13 / 83.58 | **32.44 / 91.02** | 50.87 / **84.14** | 57.83 / 81.51 | **45.82 / 85.06** |

Table 31: Near-OOD detection results (FPR@95↓ / AUROC↑) on CIFAR-10 and CIFAR-100 benchmarks using the WRN-28-10 network, averaged over 3 trials. The overall average performance is reported. The best results are **bold**, and the second-best results are underlined.

| | *CIFAR-10 benchmark* | | *CIFAR-100 benchmark* | | **Average** |
|---|---|---|---|---|---|
| Method | **CIFAR-100** | **TIN** | **CIFAR-10** | **TIN** | |
| MSP | 54.13 / 88.28 | 42.94 / 89.93 | **56.83** / 80.42 | 48.82 / 83.39 | 50.68 / 85.51 |
| MLS | 67.10 / 87.72 | 55.91 / 89.90 | 58.99 / **80.98** | 49.27 / 84.01 | 57.82 / 85.65 |
| EBO | 67.04 / 87.77 | 55.88 / 89.97 | 58.97 / 80.93 | 49.39 / 83.95 | 57.82 / 85.66 |
| ReAct | 65.96 / 87.76 | 51.44 / 90.33 | 69.17 / 79.10 | 51.56 / 83.77 | 59.53 / 85.24 |
| ASH | 91.33 / 70.72 | 90.77 / 73.28 | 85.25 / 69.96 | 78.31 / 74.55 | 86.42 / 72.13 |
| SCALE | 69.52 / 87.35 | 59.48 / 89.53 | 61.30 / 80.35 | 51.25 / 83.65 | 60.39 / 85.22 |
| BFAct | 66.31 / 86.94 | 57.12 / 89.69 | 78.90 / 74.98 | 59.04 / 82.25 | 65.34 / 83.47 |
| LTS | 74.28 / 86.38 | 66.01 / 88.56 | 64.17 / 79.57 | 54.24 / 83.09 | 64.68 / 84.40 |
| OptFS | 76.36 / 84.05 | 66.73 / 86.56 | 85.40 / 75.55 | 64.11 / 80.99 | 73.15 / 79.83 |
| **AdaSCALE-A** | **50.60 / 89.40** | **42.80 / 91.13** | 62.21 / 79.99 | **47.11 / 84.98** | 50.68 / **86.38** |
| **AdaSCALE-L** | 53.98 / 89.01 | 45.95 / 90.77 | 65.41 / 79.27 | 48.74 / 84.75 | 53.52 / 85.95 |

## H.2 DENSENET-101

Table 32: Far-OOD detection results (FPR@95↓ / AUROC↑) on CIFAR-10 and CIFAR-100 benchmarks using the DenseNet-101 network, averaged over 3 trials. The overall average performance is reported. The best results are **bold**, and the second-best results are underlined.

| Method | MNIST | SVHN | Textures | Places365 | Average |
|---|---|---|---|---|---|
| *CIFAR-10 benchmark* | | | | | |
| MSP | 17.91 / 94.22 | 32.04 / 90.38 | **46.80** / 87.53 | 37.59 / 89.24 | 33.59 / 90.34 |
| MLS | 10.02 / 97.58 | 31.25 / 92.59 | 64.43 / 85.58 | 39.19 / 90.74 | 36.22 / 91.62 |
| EBO | 9.74 / 97.76 | 31.23 / 92.69 | 64.46 / 85.48 | 39.17 / 90.81 | 36.15 / 91.68 |
| ReAct | 12.60 / 97.24 | 34.79 / 92.02 | 50.41 / 88.21 | **36.12** / **91.35** | 33.48 / 92.20 |
| ASH | 9.40 / **98.12** | 39.42 / 91.25 | 70.95 / 85.39 | 57.90 / 85.48 | 44.42 / 90.06 |
| SCALE | 9.04 / 97.88 | 26.99 / 93.54 | 61.52 / 86.77 | 39.57 / 90.76 | 34.28 / 92.24 |
| BFAct | 23.59 / 94.96 | 42.49 / 89.00 | 53.74 / 87.38 | 37.53 / 91.09 | 39.34 / 90.61 |
| LTS | **8.92** / 97.97 | 27.16 / 93.59 | 59.07 / 87.09 | 39.47 / 90.81 | 33.65 / 92.37 |
| OptFS | 9.74 / 97.88 | 41.20 / 90.71 | 51.35 / **88.48** | 59.47 / 86.03 | 40.44 / 90.77 |
| **AdaSCALE-A** | 12.42 / 96.85 | **25.04** / **94.05** | 58.28 / 87.35 | 36.77 / 91.20 | **33.13** / 92.36 |
| **AdaSCALE-L** | 10.92 / 97.44 | 26.43 / 93.87 | 58.59 / 87.19 | 37.03 / 91.25 | 33.24 / **92.44** |
| *CIFAR-100 benchmark* | | | | | |
| MSP | 65.65 / 72.43 | 63.81 / 76.52 | 75.34 / 72.19 | 61.36 / 77.16 | 66.54 / 74.57 |
| MLS | 58.69 / 78.55 | 57.12 / 79.43 | 79.05 / 72.68 | 62.72 / 78.38 | 64.39 / 77.26 |
| EBO | 58.58 / 78.98 | 56.76 / 79.19 | 79.09 / 72.44 | 62.86 / 78.08 | 64.32 / 77.17 |
| ReAct | 62.71 / 76.37 | 48.48 / 81.64 | **64.65** / 78.62 | **59.00** / **78.89** | 58.71 / 78.88 |
| ASH | **40.69** / **88.57** | 48.03 / **86.24** | 65.24 / **83.29** | 73.29 / 71.88 | **56.81** / **82.49** |
| SCALE | 56.92 / 79.53 | 53.81 / 80.96 | 76.10 / 74.47 | 62.49 / 78.51 | 62.33 / 78.37 |
| BFAct | 73.83 / 67.19 | 60.01 / 75.85 | 69.29 / 76.41 | 68.15 / 73.73 | 67.82 / 73.29 |
| LTS | 55.33 / 80.58 | 51.33 / 82.04 | 73.06 / 75.89 | 62.58 / 78.36 | 60.58 / 79.22 |
| OptFS | 64.24 / 75.24 | 59.81 / 76.46 | 66.15 / 77.50 | 73.47 / 69.76 | 65.92 / 74.74 |
| **AdaSCALE-A** | 62.51 / 74.96 | 46.29 / 84.31 | 71.40 / 76.59 | 61.70 / 78.86 | 60.47 / 78.68 |
| **AdaSCALE-L** | 61.33 / 75.73 | **43.97** / 85.30 | 69.31 / 77.71 | 61.97 / 78.69 | 59.15 / 79.36 |

Table 33: Near-OOD detection results (FPR@95↓ / AUROC↑) on CIFAR-10 and CIFAR-100 benchmarks using the DenseNet-101 network, averaged over 3 trials. The overall average performance is reported. The best results are **bold**, and the second-best results are underlined.

| Method | *CIFAR-10 benchmark* | | *CIFAR-100 benchmark* | | Average |
|---|---|---|---|---|---|
| | CIFAR-100 | TIN | CIFAR-10 | TIN | |
| MSP | **40.13** / 88.45 | **35.50** / 89.61 | **59.94** / 77.53 | 56.96 / 79.57 | **48.13** / 83.79 |
| MLS | 45.14 / 88.85 | 38.01 / 90.85 | 63.61 / **78.26** | 57.09 / 81.75 | 50.96 / 84.93 |
| EBO | 45.19 / 88.85 | 38.05 / 90.90 | 63.90 / 77.94 | 57.53 / 81.58 | 51.17 / 84.82 |
| ReAct | 44.34 / 89.19 | 37.10 / 91.08 | 70.77 / 75.06 | 61.30 / 80.40 | 53.38 / 83.93 |
| ASH | 67.78 / 82.68 | 62.54 / 85.18 | 81.65 / 65.84 | 78.66 / 70.01 | 72.66 / 75.93 |
| SCALE | 45.25 / 88.92 | 37.76 / 91.01 | 64.20 / 78.13 | 56.96 / 81.88 | 51.04 / 84.99 |
| BFAct | 52.06 / 87.70 | 44.09 / 89.89 | 79.31 / 67.39 | 71.79 / 74.18 | 61.81 / 79.79 |
| LTS | 44.89 / 89.03 | 37.67 / 91.10 | 64.66 / 77.87 | 56.83 / 81.87 | 51.01 / 84.97 |
| OptFS | 60.63 / 85.29 | 55.55 / 86.96 | 82.97 / 64.99 | 74.73 / 70.55 | 68.47 / 76.95 |
| **AdaSCALE-A** | 43.29 / 89.37 | 35.57 / 91.32 | 65.51 / 78.00 | **54.49** / 82.44 | 49.72 / **85.28** |
| **AdaSCALE-L** | 43.19 / **89.40** | 35.70 / **91.37** | 66.09 / 77.79 | 54.57 / **82.47** | 49.89 / 85.26 |

# I IMAGENET-1K RESULTS

## I.1 NEAR-OOD DETECTION

Table 34: Near-OOD detection results (FPR@95↓ / AUROC↑) on ImageNet-1k benchmark using ResNet-50 network. The best results are **bold**, and the second-best results are underlined.

| Method | SSB-Hard | NINCO | ImageNet-O | Average |
|---|---|---|---|---|
| MSP | 74.49 / 72.09 | 56.88 / 79.95 | 91.32 / 28.60 | 74.23 / 60.21 |
| MLS | 76.20 / 72.51 | 59.44 / 80.41 | 88.97 / 40.73 | 74.87 / 64.55 |
| EBO | 76.54 / 72.08 | 60.58 / 79.70 | 88.84 / 41.78 | 75.32 / 64.52 |
| REACT | 77.55 / 73.03 | 55.82 / 81.73 | 84.45 / 51.67 | 72.61 / 68.81 |
| ASH | 73.66 / 72.89 | 53.05 / 83.45 | 81.70 / 57.67 | 69.47 / 71.33 |
| SCALE | 67.72 / 77.35 | 51.80 / 85.37 | 83.77 / 59.89 | 67.76 / 74.20 |
| BFAct | 77.20 / 73.15 | 55.27 / 81.88 | 84.57 / 51.62 | 72.35 / 68.88 |
| LTS | 68.46 / 77.10 | 51.24 / 85.33 | 84.33 / 57.69 | 68.01 / 73.37 |
| OptFS | 78.32 / 71.01 | 52.09 / 82.51 | 78.56 / 59.40 | 69.66 / 70.97 |
| **AdaSCALE-A** | **57.96** / **81.68** | **44.92** / **87.15** | **74.06** / **68.12** | **58.98** / **78.98** |
| **AdaSCALE-L** | 58.68 / 81.42 | 45.01 / 87.11 | 75.83 / 67.33 | 59.84 / 78.62 |

Table 35: Near-OOD detection results (FPR@95↓ / AUROC↑) on ImageNet-1k benchmark using ResNet-101 network. The best results are **bold**, and the second-best results are underlined.

| Method | SSB-Hard | NINCO | ImageNet-O | Average |
|---|---|---|---|---|
| MSP | 73.20 / 72.57 | 55.27 / 80.61 | 87.42 / 48.57 | 71.96 / 67.25 |
| MLS | 74.68 / 74.37 | 55.65 / 82.29 | 85.81 / 57.89 | 72.05 / 71.51 |
| EBO | 74.96 / 74.12 | 56.33 / 81.79 | 85.66 / 58.72 | 72.32 / 71.54 |
| REACT | 75.96 / 74.43 | 52.58 / 83.27 | 75.67 / 67.31 | 68.07 / 75.00 |
| ASH | 72.48 / 74.23 | 49.41 / 84.62 | 73.84 / 70.98 | 65.24 / 76.61 |
| SCALE | 68.47 / 77.10 | 49.03 / 86.20 | 74.09 / 72.50 | 63.87 / 78.60 |
| BFAct | 75.48 / 74.74 | 52.23 / 83.37 | 76.16 / 67.37 | 67.96 / 75.16 |
| OptFS | 76.55 / 72.29 | 50.89 / 83.35 | 68.94 / 71.85 | 65.46 / 75.83 |
| **AdaSCALE-A** | **61.00** / 80.29 | **46.70** / **86.99** | 62.05 / 78.27 | 56.59 / 81.85 |
| **AdaSCALE-L** | 61.05 / **80.41** | 47.77 / 86.84 | **60.40** / **78.35** | **56.41** / **81.86** |

Table 36: Near-OOD detection results (FPR@95↓ / AUROC↑) on ImageNet-1k benchmark using RegNet-Y-16 network. The best results are **bold**, and the second-best results are underlined.

| Method | SSB-Hard | NINCO | ImageNet-O | Average |
|---|---|---|---|---|
| MSP | 65.35 / 78.28 | 48.48 / 86.85 | 72.82 / 77.09 | 62.22 / 80.74 |
| MLS | 62.48 / 84.83 | 42.76 / 91.56 | 83.60 / 77.58 | 62.94 / 84.66 |
| EBO | 62.10 / 85.28 | 42.49 / 91.67 | 83.82 / 77.33 | 62.80 / 84.76 |
| REACT | 73.02 / 73.17 | 59.81 / 80.91 | 79.37 / 72.02 | 70.73 / 75.37 |
| ASH | 80.58 / 67.70 | 77.23 / 71.42 | 89.71 / 64.30 | 82.51 / 67.81 |
| SCALE | 66.98 / 82.35 | 49.84 / 89.93 | 84.44 / 76.43 | 67.09 / 82.90 |
| BFAct | 79.40 / 64.39 | 73.98 / 70.35 | 82.76 / 63.54 | 78.72 / 66.09 |
| LTS | 69.52 / 79.78 | 55.38 / 87.71 | 84.55 / 74.78 | 69.82 / 80.75 |
| OptFS | 79.59 / 69.47 | 63.97 / 80.36 | 77.03 / 75.79 | 73.53 / 75.21 |
| **AdaSCALE-A** | **54.50** / **87.21** | **31.50** / **93.50** | **57.75** / **86.83** | **47.91** / **89.18** |
| **AdaSCALE-L** | 62.61 / 84.60 | 47.84 / 90.13 | 57.94 / 86.61 | 56.13 / 87.11 |

Table 37: Near-OOD detection results (FPR@95↓ / AUROC↑) on ImageNet-1k benchmark using ResNeXt-50 network. The best results are **bold**, and the second-best results are underlined.

| Method | SSB-Hard | NINCO | ImageNet-O | Average |
|---|---|---|---|---|
| MSP | 73.04 / 73.28 | 57.90 / 80.86 | 88.81 / 49.43 | 73.25 / 67.86 |
| MLS | 74.68 / 75.06 | 60.79 / 81.91 | 86.87 / 57.87 | 74.11 / 71.61 |
| EBO | 74.90 / 74.89 | 60.96 / 81.44 | 86.76 / 58.49 | 74.21 / 71.61 |
| REACT | 75.54 / 74.51 | 57.29 / 82.50 | 80.03 / 65.37 | 70.95 / 74.13 |
| ASH | 70.72 / 76.64 | 58.40 / 83.49 | 83.84 / 65.63 | 70.99 / 75.25 |
| SCALE | 67.77 / 79.73 | 56.87 / 85.39 | 87.15 / 63.48 | 70.60 / 76.20 |
| BFAct | 75.36 / 74.65 | 57.65 / 82.46 | 79.86 / 65.30 | 70.96 / 74.14 |
| LTS | 68.26 / 79.36 | 56.35 / 85.39 | 86.22 / 63.85 | 70.28 / 76.20 |
| OptFS | 75.62 / 73.82 | 57.07 / 82.37 | **75.13** / 68.33 | 69.27 / 74.84 |
| **AdaSCALE-A** | **61.03** / **81.86** | 50.80 / **86.54** | 80.57 / 71.48 | 64.13 / 79.96 |
| **AdaSCALE-L** | 61.57 / 81.11 | **48.78** / 86.40 | 75.88 / **73.02** | **62.08** / **80.18** |

Table 38: Near-OOD detection results (FPR@95↓ / AUROC↑) on ImageNet-1k benchmark using DenseNet-201 network. The best results are **bold**, and the second-best results are underlined.

| Method | SSB-Hard | NINCO | ImageNet-O | Average |
|---|---|---|---|---|
| MSP | 74.43 / 72.23 | 56.69 / 80.85 | 89.18 / 48.80 | 73.44 / 67.29 |
| MLS | 76.62 / 72.48 | 60.14 / 80.91 | 89.78 / 53.34 | 75.51 / 68.91 |
| EBO | 76.92 / 72.00 | 60.88 / 80.01 | 89.75 / 54.03 | 75.85 / 68.68 |
| ReAct | 78.62 / 70.93 | 57.51 / 81.19 | 73.78 / 68.83 | 69.97 / 73.65 |
| ASH | 78.80 / 68.71 | 63.84 / 79.45 | 80.07 / 68.19 | 74.24 / 72.12 |
| SCALE | 73.64 / 74.43 | 56.90 / 83.80 | 84.14 / 62.92 | 71.56 / 73.72 |
| BFAct | 81.57 / 67.52 | 65.10 / 77.38 | 66.93 / 72.93 | 71.20 / 72.61 |
| LTS | 73.46 / 74.36 | 57.54 / 83.79 | 82.87 / 65.52 | 71.29 / 74.56 |
| OptFS | 82.76 / 65.38 | 63.26 / 78.12 | 69.21 / 72.79 | 71.74 / 72.10 |
| **AdaSCALE-A** | **68.46** / **77.10** | **56.66** / **84.32** | 58.72 / 77.55 | **61.28** / 79.66 |
| **AdaSCALE-L** | 68.97 / 76.85 | 57.96 / 83.92 | **58.30** / **79.41** | 61.75 / **80.06** |

Table 39: Near-OOD detection results (FPR@95↓ / AUROC↑) on ImageNet-1k benchmark using EfficientNetV2-L network. The best results are **bold**, and the second-best results are underlined.

| Method | SSB-Hard | NINCO | ImageNet-O | Average |
|---|---|---|---|---|
| MSP | 81.28 / 75.03 | 57.97 / 86.70 | 78.26 / 80.53 | 72.51 / 80.76 |
| MLS | 84.74 / 73.50 | 72.88 / 84.83 | 86.71 / 79.32 | 81.44 / 79.22 |
| EBO | 85.27 / 71.58 | 75.81 / 82.07 | 87.49 / 77.81 | 82.86 / 77.15 |
| ReAct | 74.29 / 70.63 | 71.93 / 70.92 | 70.86 / 72.63 | 72.36 / 71.39 |
| ASH | 94.82 / 46.73 | 96.44 / 37.79 | 93.30 / 49.81 | 94.85 / 44.78 |
| SCALE | 90.16 / 57.07 | 89.93 / 59.69 | 89.03 / 63.60 | 89.70 / 60.12 |
| BFAct | 75.36 / 63.66 | 77.03 / 59.56 | 74.19 / 64.18 | 75.53 / 62.46 |
| LTS | 88.43 / 68.29 | 86.68 / 75.87 | 86.78 / 76.73 | 87.30 / 73.63 |
| OptFS | 74.68 / 73.83 | 70.24 / 76.18 | 71.94 / 75.86 | 72.29 / 75.29 |
| **AdaSCALE-A** | 60.84 / 83.48 | **47.45** / **89.47** | 53.04 / **87.87** | **53.78** / **86.94** |
| **AdaSCALE-L** | **53.56** / **85.00** | 58.55 / 84.75 | **52.72** / 87.58 | 54.95 / 85.77 |

Table 40: Near-OOD detection results (FPR@95↓ / AUROC↑) on
ImageNet-1k benchmark using ViT-B-16 network. The best results are
**bold**, and the second-best results are underlined.

| Method | SSB-Hard | NINCO | ImageNet-O | Average |
|---|---|---|---|---|
| MSP | 86.41 / **68.94** | 77.28 / 78.11 | 96.48 / 58.81 | 86.72 / 68.62 |
| MLS | 91.52 / 64.20 | 92.98 / 72.40 | 96.84 / 54.33 | 93.78 / 63.64 |
| EBO | 92.24 / 58.80 | 94.14 / 66.02 | 96.74 / 52.74 | 94.37 / 59.19 |
| ReAct | 90.46 / 63.10 | 78.50 / 75.43 | 90.94 / 66.53 | 86.63 / 68.35 |
| ASH | 93.50 / 53.90 | 95.37 / 52.51 | 94.47 / 53.19 | 94.45 / 53.20 |
| SCALE | 92.37 / 56.55 | 94.62 / 61.52 | 96.44 / 50.47 | 94.48 / 56.18 |
| BFAct | 89.81 / 64.16 | 71.37 / 78.06 | 85.09 / 69.75 | 82.09 / 70.66 |
| LTS | 91.42 / 64.35 | 82.63 / 75.48 | 92.42 / 62.46 | 88.83 / 67.43 |
| OptFS | 87.98 / 66.30 | 64.24 / 80.46 | 77.43 / 71.43 | 76.55 / 72.73 |
| **AdaSCALE-A** | **85.89** / 66.57 | 61.92 / **80.47** | **67.81** / 72.37 | **71.87** / 73.14 |
| **AdaSCALE-L** | 86.19 / 66.25 | **61.79** / 80.42 | 67.99 / **73.01** | 71.99 / **73.23** |

Table 41: Near-OOD detection results (FPR@95↓ / AUROC↑) on
ImageNet-1k benchmark using Swin-B network. The best results are **bold**,
and the second-best results are underlined.

| Method | SSB-Hard | NINCO | ImageNet-O | Average |
|---|---|---|---|---|
| MSP | 86.47 / 71.30 | 77.95 / 78.50 | 96.90 / 59.65 | 87.11 / 69.82 |
| MLS | 94.05 / 65.04 | 93.38 / 71.75 | 96.97 / 57.26 | 94.80 / 64.68 |
| EBO | 94.66 / 58.96 | 94.59 / 64.02 | 96.75 / 56.40 | 95.34 / 59.79 |
| ReAct | 89.19 / 68.70 | 68.54 / 80.16 | 90.20 / 70.93 | 82.64 / 73.26 |
| ASH | 97.15 / 45.47 | 96.64 / 47.36 | 95.32 / 49.92 | 96.37 / 47.58 |
| SCALE | 90.84 / 56.53 | 87.86 / 62.49 | 87.16 / 65.38 | 88.62 / 61.47 |
| BFAct | 84.86 / 69.41 | 61.30 / 81.10 | **69.27** / **75.34** | **71.81** / **75.28** |
| LTS | 90.36 / 64.51 | 81.02 / 74.23 | 88.44 / 62.92 | 86.61 / 67.22 |
| OptFS | 88.68 / 68.43 | 66.36 / 80.27 | 75.38 / 73.49 | 76.81 / 74.06 |
| **AdaSCALE-A** | **80.10** / **70.46** | 64.67 / 81.10 | 75.46 / 71.87 | 73.41 / 74.48 |
| **AdaSCALE-L** | 80.12 / 70.06 | **63.68** / **81.35** | 74.87 / 72.34 | 72.89 / 74.58 |

## I.2 FAR-OOD DETECTION

Table 42: Far-OOD detection results (FPR@95↓ / AUROC↑) on
ImageNet-1k benchmark using ResNet-50 network. The best results are
**bold**, and the second-best results are underlined.

| Method | iNaturalist | Textures | OpenImage-O | Places | Average |
|--------|-------------|----------|-------------|--------|---------|
| MSP | 43.34 / 88.41 | 60.87 / 82.43 | 50.13 / 84.86 | 58.26 / 80.55 | 53.15 / 84.06 |
| MLS | 30.61 / 91.17 | 46.17 / 88.39 | 37.88 / 89.17 | 55.62 / 84.05 | 42.57 / 88.19 |
| EBO | 31.30 / 90.63 | 45.77 / 88.70 | 38.09 / 89.06 | 55.73 / 83.97 | 42.72 / 88.09 |
| ReAct | 16.72 / 96.34 | 29.64 / 92.79 | 32.58 / 91.87 | 41.62 / 90.93 | 30.14 / 92.98 |
| ASH | 14.09 / 97.06 | 15.30 / 96.90 | 29.19 / 93.26 | 40.16 / 90.48 | 24.69 / 94.43 |
| SCALE | 9.50 / 98.02 | 11.90 / 97.63 | 28.18 / 93.95 | 36.18 / 91.96 | 21.44 / 95.39 |
| BFAct | 15.94 / 96.47 | 28.43 / 92.87 | 32.66 / 91.90 | 40.83 / 90.79 | 29.46 / 93.01 |
| LTS | 10.24 / 97.87 | 13.06 / 97.42 | 27.81 / 94.01 | 37.68 / 91.65 | 22.20 / 95.24 |
| OptFS | 15.88 / 96.65 | 16.60 / 96.10 | 29.94 / 92.53 | 40.24 / 90.20 | 25.66 / 93.87 |
| **AdaSCALE-A** | **7.61 / 98.31** | 10.57 / 97.88 | 20.67 / 95.62 | **32.60 / 92.74** | **17.86 / 96.14** |
| **AdaSCALE-L** | 7.78 / 98.29 | **10.33 / 97.92** | **20.61 / 95.62** | 32.97 / 92.63 | 17.92 / 96.12 |

Table 43: Far-OOD detection results (FPR@95↓ / AUROC↑) on
ImageNet-1k benchmark using ResNet-101 network. The best results are
**bold**, and the second-best results are underlined.

| Method | iNaturalist | Textures | OpenImage-O | Places | Average |
|--------|-------------|----------|-------------|--------|---------|
| MSP | 48.30 / 86.27 | 59.00 / 83.60 | 49.36 / 84.82 | 58.84 / 80.56 | 53.87 / 83.81 |
| MLS | 41.11 / 88.83 | 43.59 / 89.85 | 38.13 / 89.25 | 52.74 / 85.28 | 43.89 / 88.30 |
| EBO | 41.65 / 88.30 | 43.66 / 90.14 | 38.48 / 89.12 | 53.42 / 85.37 | 44.30 / 88.23 |
| ReAct | 19.86 / 95.66 | 26.94 / 93.78 | 30.18 / 92.54 | 42.58 / 90.41 | 29.89 / 93.10 |
| ASH | 19.90 / 95.68 | 13.94 / 97.32 | 27.76 / 93.63 | 43.11 / 89.59 | 26.18 / 94.06 |
| SCALE | 13.90 / 97.05 | 9.34 / 98.04 | 25.91 / 94.47 | 40.99 / 90.64 | 22.54 / 95.05 |
| BFAct | 19.60 / 95.69 | 25.79 / 93.79 | 30.18 / 92.55 | 42.14 / 90.13 | 29.43 / 93.04 |
| LTS | 15.07 / 96.83 | 10.33 / 97.89 | 25.51 / 94.52 | 41.40 / 90.53 | 23.07 / 94.94 |
| OptFS | 19.11 / 95.70 | 16.53 / 96.35 | 28.76 / 92.94 | 43.47 / 89.22 | 26.97 / 93.55 |
| **AdaSCALE-A** | **10.74 / 97.64** | **8.90 / 98.21** | 18.75 / 96.03 | **35.66 / 91.92** | **18.51 / 95.95** |
| **AdaSCALE-L** | 11.71 / 97.36 | 10.44 / 97.93 | **17.87 / 96.18** | 36.57 / 91.55 | 19.15 / 95.76 |

Table 44: Far-OOD detection results (FPR@95↓ / AUROC↑) on
ImageNet-1k benchmark using RegNet-Y-16 network. The best results are
**bold**, and the second-best results are underlined.

| Method | iNaturalist | Textures | OpenImage-O | Places | Average |
|---|---|---|---|---|---|
| MSP | 28.13 / 94.67 | 44.73 / 88.48 | 36.27 / 91.96 | 52.51 / 85.21 | 40.41 / 90.08 |
| MLS | 9.10 / 98.05 | 39.74 / 92.82 | 25.71 / 95.70 | 57.14 / 88.22 | 32.92 / 93.70 |
| EBO | 7.72 / 98.29 | 38.18 / 93.02 | 25.94 / 95.83 | 58.04 / 88.13 | 32.47 / 93.82 |
| ReAct | 21.24 / 94.14 | 41.20 / 87.25 | 43.46 / 89.20 | 74.92 / 74.10 | 45.20 / 86.17 |
| ASH | 48.89 / 87.39 | 45.75 / 88.79 | 70.98 / 82.52 | 72.99 / 77.06 | 59.65 / 83.94 |
| SCALE | 11.13 / 97.88 | 28.29 / 95.31 | 33.59 / 94.87 | 55.62 / 88.59 | 32.16 / 94.16 |
| BFAct | 37.88 / 86.24 | 54.87 / 77.64 | 62.53 / 79.59 | 79.46 / 65.39 | 58.69 / 77.22 |
| LTS | 14.29 / 97.52 | 25.21 / 95.72 | 43.38 / 93.53 | 57.08 / 87.51 | 34.99 / 93.57 |
| OptFS | 28.95 / 93.68 | 39.99 / 90.13 | 44.96 / 89.85 | 75.59 / 73.24 | 47.37 / 86.73 |
| **AdaSCALE-A** | **4.34** / **99.09** | 26.06 / 95.21 | **13.09** / **97.57** | **41.98** / **91.48** | 21.37 / 95.84 |
| **AdaSCALE-L** | 4.41 / 99.02 | **13.50** / **97.61** | 18.56 / 96.92 | 43.93 / 91.22 | **20.10** / **96.19** |

Table 45: Far-OOD detection results (FPR@95↓ / AUROC↑) on
ImageNet-1k benchmark using ResNeXt-50 network. The best results are
**bold**, and the second-best results are underlined.

| Method | iNaturalist | Textures | OpenImage-O | Places | Average |
|---|---|---|---|---|---|
| MSP | 43.56 / 88.04 | 62.23 / 82.13 | 48.06 / 85.65 | 58.42 / 81.02 | 53.07 / 84.21 |
| MLS | 32.96 / 90.93 | 51.58 / 87.39 | 37.33 / 89.80 | 57.76 / 83.77 | 44.91 / 87.97 |
| EBO | 33.42 / 90.54 | 51.73 / 87.56 | 37.79 / 89.72 | 57.56 / 83.62 | 45.12 / 87.86 |
| ReAct | 17.64 / 95.95 | 32.86 / 91.67 | 29.82 / 92.37 | 39.92 / 90.76 | 30.06 / 92.69 |
| ASH | 17.90 / 96.22 | 23.74 / 95.18 | 30.83 / 93.13 | 44.21 / 89.35 | 29.17 / 93.47 |
| SCALE | 15.66 / 96.75 | 27.75 / 94.94 | 31.43 / 93.41 | 47.62 / 89.08 | 30.62 / 93.54 |
| BFAct | 17.40 / 95.91 | 32.00 / 91.83 | 29.53 / 92.38 | 39.89 / 90.57 | 29.71 / 92.67 |
| LTS | 16.29 / 96.63 | 26.64 / 95.07 | 30.50 / 93.50 | 48.04 / 88.78 | 30.37 / 93.49 |
| OptFS | 17.20 / 96.12 | 23.11 / 94.69 | 29.59 / 92.75 | 40.24 / 90.05 | 27.54 / 93.40 |
| **AdaSCALE-A** | **10.02** / **97.80** | **17.99** / **96.38** | 22.93 / 95.17 | **37.38** / **91.62** | **22.08** / **95.24** |
| **AdaSCALE-L** | 11.28 / 97.45 | 18.46 / 96.20 | **21.23** / **95.35** | 37.68 / 91.03 | 22.16 / 95.01 |

Table 46: Far-OOD detection results (FPR@95↓ / AUROC↑) on
ImageNet-1k benchmark using DenseNet-201 network. The best results are
**bold**, and the second-best results are underlined.

| Method | iNaturalist | Textures | OpenImage-O | Places | Average |
|---|---|---|---|---|---|
| MSP | 42.02 / 89.84 | 62.33 / 81.56 | 50.31 / 85.19 | 59.74 / 81.14 | 53.60 / 84.43 |
| MLS | 31.99 / 92.11 | 57.75 / 85.56 | 42.70 / 88.28 | 61.30 / 83.82 | 48.43 / 87.44 |
| EBO | 33.12 / 91.46 | 57.47 / 85.55 | 43.75 / 87.91 | 61.46 / 83.67 | 48.95 / 87.15 |
| ReAct | 19.41 / 95.64 | 23.86 / 94.63 | 32.54 / 91.83 | **47.06** / 88.52 | 30.72 / 92.65 |
| ASH | 21.57 / 95.47 | 21.42 / 95.56 | 41.23 / 90.19 | 49.80 / 87.45 | 33.50 / 92.17 |
| SCALE | 18.13 / **96.29** | 27.22 / 94.52 | 34.52 / 92.15 | 52.82 / 87.83 | 33.17 / 92.70 |
| BFAct | 20.64 / 95.42 | 21.70 / 95.17 | 39.76 / 89.97 | 47.72 / **88.61** | 32.45 / 92.29 |
| LTS | 15.68 / 96.71 | 22.49 / 95.81 | 34.27 / 92.37 | 51.23 / 88.26 | 30.92 / 93.29 |
| OptFS | 25.81 / 93.92 | 21.75 / 95.01 | 38.45 / 89.67 | 51.66 / 85.54 | 34.42 / 91.04 |
| **AdaSCALE-A** | **17.30** / 96.03 | 19.42 / 96.23 | **23.12** / 94.68 | 52.20 / 85.98 | 28.01 / **93.23** |
| **AdaSCALE-L** | 17.97 / 95.87 | **16.87** / **96.69** | 23.64 / **94.69** | 53.50 / 85.46 | **28.00** / 93.18 |

Table 47: Far-OOD detection results (FPR@95↓ / AUROC↑) on
ImageNet-1k benchmark using EfficientNetV2-L network. The best results
are **bold**, and the second-best results are underlined.

| Method | iNaturalist | Textures | OpenImage-O | Places | Average |
|---|---|---|---|---|---|
| MSP | 25.14 / 95.12 | 74.42 / 84.20 | 40.64 / 91.74 | 78.74 / 80.61 | 54.74 / 87.92 |
| MLS | 35.28 / 94.13 | 86.65 / 80.26 | 62.11 / 90.26 | 90.53 / 74.56 | 68.64 / 84.80 |
| EBO | 49.84 / 91.21 | 87.72 / 75.77 | 68.77 / 87.66 | 91.60 / 69.89 | 74.48 / 81.13 |
| ReAct | 46.44 / 80.96 | 54.56 / 77.17 | 60.79 / 78.20 | 78.39 / 64.99 | 60.05 / 75.33 |
| ASH | 96.26 / 37.76 | 95.40 / 50.98 | 97.52 / 43.19 | 97.07 / 34.34 | 96.56 / 41.57 |
| SCALE | 87.08 / 67.69 | 86.22 / 67.44 | 91.05 / 67.21 | 94.18 / 47.99 | 89.63 / 62.58 |
| BFAct | 57.31 / 69.11 | 63.43 / 67.70 | 69.30 / 67.49 | 76.86 / 58.52 | 66.72 / 65.70 |
| LTS | 79.05 / 84.72 | 86.89 / 75.39 | 88.00 / 81.53 | 93.45 / 63.56 | 86.85 / 76.30 |
| OptFS | 38.62 / 89.80 | 45.77 / 86.94 | 53.77 / 85.49 | 76.31 / 72.23 | 53.62 / 83.62 |
| **AdaSCALE-A** | **18.51** / **96.67** | 42.07 / 90.56 | **31.00** / **94.44** | 58.87 / **84.26** | **37.61** / **91.48** |
| **AdaSCALE-L** | 26.58 / 95.02 | **32.81** / **92.38** | 39.19 / 92.31 | **56.66** / 82.33 | 38.81 / 90.51 |

Table 48: Far-OOD detection results (FPR@95↓ / AUROC↑) on
ImageNet-1k benchmark using Vit-B-16 network. The best results are **bold**,
and the second-best results are underlined.

| Method | iNaturalist | Textures | OpenImage-O | Places | Average |
|---|---|---|---|---|---|
| MSP | 42.40 / 88.19 | 56.46 / 85.06 | 56.19 / 84.86 | 70.59 / 80.38 | 56.41 / 84.62 |
| MLS | 72.98 / 85.29 | 78.93 / 83.74 | 85.78 / 81.60 | 89.88 / 75.05 | 81.89 / 81.42 |
| EBO | 83.56 / 79.30 | 83.66 / 81.17 | 88.82 / 76.48 | 91.77 / 68.42 | 86.95 / 76.34 |
| ReAct | 48.22 / 86.11 | 55.87 / 86.66 | 57.68 / 84.29 | 75.48 / 77.52 | 59.31 / 83.65 |
| ASH | 97.02 / 50.62 | 98.50 / 48.53 | 94.79 / 55.51 | 93.60 / 53.97 | 95.98 / 52.16 |
| SCALE | 86.60 / 73.94 | 84.70 / 79.00 | 89.48 / 72.72 | 92.67 / 63.60 | 88.36 / 72.32 |
| BFAct | 40.56 / 87.96 | 48.65 / 88.31 | 48.24 / 86.59 | 68.86 / 80.21 | 51.58 / 85.77 |
| LTS | 50.42 / 88.92 | 61.70 / 86.53 | 69.26 / 83.45 | 76.07 / 78.82 | 64.37 / 84.43 |
| OptFS | **34.39** / **89.99** | **46.41** / **88.48** | **42.20** / **88.23** | 61.44 / **82.69** | **46.11** / **87.35** |
| **AdaSCALE-A** | 36.38 / 89.60 | 51.13 / 87.16 | 43.02 / 88.07 | **59.97** / 82.48 | 47.63 / 86.83 |
| **AdaSCALE-L** | 35.16 / 89.84 | 50.91 / 87.37 | 43.01 / 88.13 | 60.05 / 82.55 | 47.28 / 86.97 |

Table 49: Far-OOD detection results (FPR@95↓ / AUROC↑) on
ImageNet-1k benchmark using Swin-B network. The best results are **bold**,
and the second-best results are underlined.

| Method | iNaturalist | Textures | OpenImage-O | Places | Average |
|---|---|---|---|---|---|
| MSP | 55.63 / 86.47 | 79.28 / 80.12 | 81.22 / 81.72 | 77.41 / 79.78 | 73.39 / 82.02 |
| MLS | 93.46 / 78.87 | 94.60 / 74.73 | 97.61 / 70.72 | 94.97 / 69.17 | 95.16 / 73.37 |
| EBO | 95.11 / 67.72 | 95.36 / 69.69 | 97.97 / 60.19 | 95.87 / 58.35 | 96.08 / 63.99 |
| ReAct | 40.77 / 88.60 | 62.26 / 85.54 | 58.19 / 85.76 | 74.21 / 79.16 | 58.86 / 84.77 |
| ASH | 98.59 / 42.18 | 98.55 / 43.37 | 98.23 / 43.28 | 97.57 / 43.98 | 98.23 / 43.20 |
| SCALE | 87.83 / 62.98 | 87.71 / 69.63 | 88.75 / 66.63 | 82.08 / 67.82 | 86.59 / 66.77 |
| BFAct | **25.76** / 91.42 | **45.73** / **87.34** | **32.13** / **91.02** | **52.33** / **84.08** | **38.99** / **88.47** |
| LTS | 57.92 / 86.10 | 77.66 / 78.02 | 73.20 / 80.16 | 82.69 / 72.71 | 72.86 / 79.25 |
| OptFS | 31.94 / 90.56 | 50.27 / 86.91 | 36.50 / 90.18 | 58.38 / 83.51 | 44.27 / 87.79 |
| **AdaSCALE-A** | 32.82 / 90.73 | 61.82 / 85.34 | 38.58 / 89.78 | 58.02 / 82.71 | 47.81 / 87.14 |
| **AdaSCALE-L** | 30.95 / **91.69** | 60.17 / 86.30 | 37.52 / 90.08 | 56.32 / 83.82 | 46.24 / 87.97 |

## I.3 FULL-SPECTRUM NEAR-OOD DETECTION

Table 50: Near-FSOOD detection results (FPR@95↓ / AUROC↑) on ImageNet-1k benchmark using ResNet-50 network. The best results are **bold**, and the second-best results are underlined.

| Method | SSB-Hard | NINCO | ImageNet-O | Average |
|---|---|---|---|---|
| MSP | 88.17 / 47.34 | 78.15 / 54.73 | 96.29 / 13.81 | 87.54 / 38.63 |
| MLS | 90.04 / 43.32 | 82.06 / 50.23 | 95.59 / 18.94 | 89.23 / 37.50 |
| EBO | 90.19 / 42.62 | 82.64 / 49.01 | 95.54 / 19.57 | 89.46 / 37.07 |
| ReAct | 90.65 / 45.19 | 80.05 / 53.37 | 93.62 / 26.15 | 88.10 / 41.57 |
| ASH | 88.82 / 44.08 | 78.35 / 54.54 | 92.48 / 30.49 | 86.55 / 43.04 |
| SCALE | 85.85 / 48.10 | 77.54 / 57.01 | 93.26 / 32.58 | 85.55 / 45.90 |
| BFAct | 90.43 / 45.29 | 79.62 / 53.50 | 93.62 / 26.20 | 87.89 / 41.66 |
| LTS | 86.37 / 47.43 | 77.54 / 56.40 | 93.61 / 30.57 | 85.84 / 44.80 |
| OptFS | 90.78 / 44.01 | 77.24 / 54.91 | 90.91 / 32.26 | 86.31 / 43.73 |
| **AdaSCALE-A** | **81.30 / 51.88** | **74.13 / 58.55** | **89.15 / 37.62** | **81.52 / 49.35** |
| **AdaSCALE-L** | 81.85 / 51.38 | 74.42 / 58.23 | 90.07 / 36.91 | 82.11 / 48.84 |

Table 51: Near-FSOOD detection results (FPR@95↓ / AUROC↑) on ImageNet-1k benchmark using ResNet-101 network. The best results are **bold**, and the second-best results are underlined.

| Method | SSB-Hard | NINCO | ImageNet-O | Average |
|---|---|---|---|---|
| MSP | 87.09 / 49.18 | 76.45 / 56.92 | 94.24 / 28.33 | 85.93 / 44.81 |
| MLS | 88.90 / 46.45 | 79.19 / 53.62 | 93.94 / 31.76 | 87.34 / 43.94 |
| EBO | 89.02 / 45.99 | 79.60 / 52.66 | 93.87 / 32.40 | 87.50 / 43.68 |
| ReAct | 89.50 / 47.79 | 77.22 / 56.02 | 89.37 / 39.62 | 85.36 / 47.81 |
| ASH | 87.84 / 46.39 | 75.36 / 56.72 | 88.48 / 42.80 | 83.90 / 48.64 |
| SCALE | 85.81 / 48.94 | 75.33 / 58.79 | 88.49 / 44.45 | 83.21 / 50.73 |
| BFAct | 89.19 / 48.07 | 76.94 / 56.02 | 89.54 / 39.65 | 85.22 / 47.91 |
| LTS | 86.02 / 48.72 | 74.89 / 58.41 | 89.17 / 43.02 | 83.36 / 50.05 |
| OptFS | 89.63 / 46.23 | 75.52 / 56.71 | 85.82 / 44.21 | 83.65 / 49.05 |
| **AdaSCALE-A** | **82.33 / 51.47** | **74.38 / 59.17** | 82.89 / 48.49 | **79.87 / 53.04** |
| **AdaSCALE-L** | 82.53 / 51.31 | 75.22 / 58.62 | **82.19 / 48.03** | 79.98 / 52.66 |

Table 52: Near-FSOOD detection results (FPR@95↓ / AUROC↑) on ImageNet-1k benchmark using RegNet-Y-16 network. The best results are **bold**, and the second-best results are underlined.

| Method | SSB-Hard | NINCO | ImageNet-O | Average |
|---|---|---|---|---|
| MSP | 83.74 / 57.23 | 72.32 / 67.69 | 87.81 / 56.61 | 81.29 / 60.51 |
| MLS | 82.91 / 60.89 | 71.22 / 70.86 | 93.27 / 55.21 | 82.46 / 62.32 |
| EBO | 82.77 / **61.63** | 71.17 / 71.23 | 93.39 / 55.02 | 82.44 / 62.63 |
| ReAct | 87.74 / 55.64 | 80.22 / 65.24 | 91.11 / 55.13 | 86.36 / 58.67 |
| ASH | 87.26 / 57.45 | 84.81 / 61.59 | 93.78 / 54.76 | 88.61 / 57.93 |
| SCALE | 83.23 / 61.02 | 72.48 / **71.33** | 92.82 / 57.16 | 82.84 / 63.17 |
| BFAct | 90.46 / 54.73 | 87.03 / 61.98 | 92.37 / 54.13 | 89.96 / 56.95 |
| LTS | 83.53 / 60.77 | 74.23 / 70.84 | 92.30 / 57.71 | 83.35 / 63.11 |
| OptFS | 90.33 / 51.78 | 80.82 / 63.86 | 88.86 / 59.45 | 86.67 / 58.36 |
| **AdaSCALE-A** | **81.68** / 60.46 | **68.05** / 70.95 | 83.25 / 60.91 | **77.66** / **64.11** |
| **AdaSCALE-L** | 84.30 / 59.42 | 76.30 / 68.49 | **81.86** / **63.12** | 80.82 / 63.68 |

Table 53: Near-FSOOD detection results (FPR@95↓ / AUROC↑) on ImageNet-1k benchmark using ResNeXt-50 network. The best results are **bold**, and the second-best results are underlined.

| Method | SSB-Hard | NINCO | ImageNet-O | Average |
|---|---|---|---|---|
| MSP | 86.95 / 49.77 | 78.27 / 57.25 | 94.79 / 28.95 | 86.67 / 45.32 |
| MLS | 88.56 / 47.94 | 81.58 / 54.31 | 94.26 / 32.40 | 88.13 / 44.88 |
| EBO | 88.68 / 47.69 | 81.67 / 53.51 | 94.21 / 32.99 | 88.19 / 44.73 |
| ReAct | 89.45 / 47.44 | 80.37 / 55.29 | 91.40 / 37.58 | 87.07 / 46.77 |
| ASH | 86.26 / 49.73 | 79.62 / 57.17 | 92.63 / 39.67 | 86.17 / 48.86 |
| SCALE | 84.64 / **52.60** | 78.75 / 58.90 | 94.20 / 37.99 | 85.86 / 49.83 |
| BFAct | 89.22 / 47.70 | 80.39 / 55.34 | 91.24 / 37.66 | 86.95 / 46.90 |
| LTS | 85.03 / 51.87 | 78.65 / 58.48 | 93.77 / 37.96 | 85.82 / 49.43 |
| OptFS | 89.63 / 46.23 | 75.52 / 56.71 | 85.82 / 44.21 | 83.65 / 49.05 |
| **AdaSCALE-A** | **82.33** / 51.47 | **74.38** / **59.17** | **82.89** / **48.49** | **79.87** / **53.04** |
| **AdaSCALE-L** | 82.70 / 52.14 | 75.85 / 58.05 | 89.55 / 43.81 | 82.70 / 51.33 |

Table 54: Near-FSOOD detection results (FPR@95↓ / AUROC↑) on ImageNet-1k benchmark using DenseNet-201 network. The best results are **bold**, and the second-best results are underlined.

| Method | SSB-Hard | NINCO | ImageNet-O | Average |
|---|---|---|---|---|
| MSP | 87.27 / 49.71 | **76.81** / 58.30 | 95.00 / 29.36 | 86.36 / 45.79 |
| MLS | 89.24 / 47.21 | 80.48 / 55.21 | 95.61 / 30.74 | 88.44 / 44.39 |
| EBO | 89.39 / 46.79 | 80.94 / 54.13 | 95.60 / 31.44 | 88.65 / 44.12 |
| ReAct | 90.76 / 45.54 | 79.54 / 55.57 | 88.40 / 42.63 | 86.23 / 47.91 |
| ASH | 89.75 / 47.08 | 81.35 / 58.13 | 90.47 / 46.81 | 87.19 / 50.67 |
| SCALE | 87.55 / **49.53** | 78.35 / 59.54 | 92.79 / 39.22 | 86.23 / 49.43 |
| BFAct | 92.33 / 44.88 | 83.90 / 54.83 | 84.88 / 49.50 | 87.04 / 49.74 |
| LTS | 87.30 / 50.02 | 78.41 / **60.35** | 92.12 / 42.25 | 85.94 / 50.88 |
| OptFS | 92.32 / 43.61 | 81.45 / 56.09 | 85.02 / **50.48** | 86.26 / 50.06 |
| **AdaSCALE-A** | **86.22** / 48.41 | 80.25 / 56.38 | 81.35 / 47.45 | **82.60** / 50.75 |
| **AdaSCALE-L** | 86.43 / 48.54 | 80.83 / 56.30 | **81.00** / 50.00 | 82.75 / **51.61** |

Table 55: Near-FSOOD detection results (FPR@95↓ / AUROC↑) on ImageNet-1k benchmark using EfficientNetV2-L network. The best results are **bold**, and the second-best results are underlined.

| Method | SSB-Hard | NINCO | ImageNet-O | Average |
|---|---|---|---|---|
| MSP | 83.74 / 57.23 | 72.32 / 67.69 | 87.81 / 56.61 | 81.29 / 60.51 |
| MLS | 82.91 / 60.89 | 71.22 / 70.86 | 93.27 / 55.21 | 82.46 / 62.32 |
| EBO | 82.77 / **61.63** | 71.17 / 71.23 | 93.39 / 55.02 | 82.44 / 62.63 |
| ReAct | 87.74 / 55.64 | 80.22 / 65.24 | 91.11 / 55.13 | 86.36 / 58.67 |
| ASH | 87.26 / 57.45 | 84.81 / 61.59 | 93.78 / 54.76 | 88.61 / 57.93 |
| SCALE | 83.23 / 61.02 | 72.48 / 71.33 | 92.82 / 57.16 | 82.84 / 63.17 |
| BFAct | 90.46 / 54.73 | 87.03 / 61.98 | 92.37 / 54.13 | 89.96 / 56.95 |
| LTS | 83.53 / 60.77 | 74.23 / 70.84 | 92.30 / 57.71 | 83.35 / 63.11 |
| OptFS | 90.33 / 51.78 | 80.82 / 63.86 | 88.86 / 59.45 | 86.67 / 58.36 |
| **AdaSCALE-A** | **81.68** / 60.46 | **68.05** / 70.95 | 83.25 / 60.91 | **77.66** / 64.11 |
| **AdaSCALE-L** | 84.30 / 59.42 | 76.30 / 68.49 | **81.86** / **63.12** | 80.82 / 63.68 |

Table 56: Near-FSOOD detection results (FPR@95↓ / AUROC↑) on
ImageNet-1k benchmark using Vit-B-16 network. The best results are **bold**,
and the second-best results are underlined.

| Method | SSB-Hard | NINCO | ImageNet-O | Average |
|---|---|---|---|---|
| MSP | 92.28 / 47.57 | 87.44 / **56.23** | 98.02 / 39.33 | 92.58 / 47.71 |
| MLS | 94.11 / 44.88 | 95.17 / 52.44 | 98.00 / 37.77 | 95.76 / 45.03 |
| EBO | 94.47 / 42.06 | 95.86 / 48.45 | 97.89 / 38.03 | 96.07 / 42.85 |
| ReAct | 94.95 / 41.84 | 88.65 / 52.48 | 95.21 / 44.64 | 92.94 / 46.32 |
| ASH | **88.95** / **56.47** | 91.09 / 55.11 | 90.00 / **55.78** | 90.01 / 55.79 |
| SCALE | 94.52 / 41.21 | 96.30 / 45.78 | 97.76 / 37.09 | 96.19 / 41.36 |
| BFAct | 94.99 / 41.44 | 85.62 / 53.35 | 92.62 / 45.64 | 91.07 / 46.81 |
| LTS | 95.33 / 43.36 | 90.52 / 53.14 | 95.90 / 41.30 | 93.91 / 45.93 |
| OptFS | 94.19 / 43.01 | 81.66 / 55.60 | 88.90 / 46.04 | 88.25 / **48.22** |
| **AdaSCALE-A** | 93.30 / 42.49 | 81.00 / 54.72 | **84.12** / 45.71 | **86.14** / 47.64 |
| **AdaSCALE-L** | 93.52 / 41.83 | **80.94** / 54.16 | 84.26 / 45.82 | 86.24 / 47.27 |

Table 57: Near-FSOOD detection results (FPR@95↓ / AUROC↑) on
ImageNet-1k benchmark using Swin-B network. The best results are **bold**,
and the second-best results are underlined.

| Method | SSB-Hard | NINCO | ImageNet-O | Average |
|---|---|---|---|---|
| MSP | 91.55 / **53.29** | 86.73 / **60.62** | 97.85 / 42.90 | 92.04 / 52.27 |
| MLS | 94.49 / 50.01 | 93.94 / 56.40 | 97.11 / 43.76 | 95.18 / 50.06 |
| EBO | 94.66 / 47.41 | 94.58 / 52.04 | 96.76 / 45.84 | 95.33 / 48.43 |
| ReAct | 94.04 / 47.83 | 82.85 / 58.52 | 94.60 / 50.41 | 90.50 / 52.25 |
| ASH | 91.77 / 50.35 | 90.91 / 52.09 | 88.80 / 54.50 | 90.49 / 52.31 |
| SCALE | 93.39 / 47.38 | 91.25 / 53.00 | 90.80 / **56.10** | 91.81 / 52.16 |
| BFAct | 92.65 / 48.33 | **79.61** / 59.66 | **84.26** / 53.17 | **85.51** / **53.72** |
| LTS | 94.26 / 48.70 | 88.32 / 57.60 | 93.04 / 46.33 | 91.87 / 50.88 |
| OptFS | 94.06 / 47.77 | 81.91 / 59.14 | 86.95 / 51.34 | 87.64 / 52.75 |
| **AdaSCALE-A** | 90.29 / 46.84 | 81.54 / 57.44 | 87.74 / 47.55 | 86.52 / 50.61 |
| **AdaSCALE-L** | **90.18** / 46.63 | 80.77 / 57.85 | 87.27 / 47.90 | 86.07 / 50.79 |

### I.4 FULL-SPECTRUM FAR-OOD DETECTION

Table 58: Far-FSOOD detection results (FPR@95↓ / AUROC↑) on
ImageNet-1k benchmark using ResNet-50 network. The best results are
**bold**, and the second-best results are underlined.

| Method | iNaturalist | Textures | OpenImage-O | Places | Average |
|---|---|---|---|---|---|
| MSP | 69.31 / 65.65 | 80.57 / 59.22 | 73.94 / 60.74 | 79.02 / 56.04 | 75.71 / 60.41 |
| MLS | 64.71 / 63.30 | 74.69 / 61.67 | 69.73 / 60.60 | 80.04 / 55.08 | 72.29 / 60.16 |
| EBO | 65.30 / 61.43 | 74.48 / 61.87 | 69.92 / 59.93 | 80.12 / 54.48 | 72.45 / 59.42 |
| ReAct | 51.90 / 75.79 | 63.55 / 69.22 | 65.64 / 67.36 | 71.81 / 67.01 | 63.22 / 69.84 |
| ASH | 49.21 / 76.93 | 50.54 / 77.64 | 63.04 / 69.03 | 70.62 / 64.79 | 58.35 / 72.10 |
| SCALE | 43.34 / 79.23 | 46.60 / 79.58 | 62.26 / 70.54 | 67.94 / 67.05 | 55.04 / 74.10 |
| BFAct | 51.01 / 75.85 | 62.45 / 69.03 | 65.52 / 67.21 | 71.17 / 66.45 | 62.54 / 69.63 |
| LTS | 45.12 / 78.72 | 48.77 / 79.13 | 62.43 / 70.17 | 69.31 / 66.20 | 56.41 / 73.55 |
| OptFS | 49.39 / 77.14 | 50.25 / 75.59 | 62.46 / 68.87 | 69.84 / 65.65 | 57.99 / 71.81 |
| **AdaSCALE-A** | **41.24** / 79.53 | 45.55 / 79.64 | **56.43** / **72.85** | **66.13** / **67.54** | **52.33** / 74.89 |
| **AdaSCALE-L** | 41.75 / **79.63** | 45.67 / **79.96** | 56.80 / 72.82 | 66.69 / 67.28 | 52.73 / **74.92** |

Table 59: Far-FSOOD detection results (FPR@95↓ / AUROC↑) on
ImageNet-1k benchmark using ResNet-101 network. The best results are
**bold**, and the second-best results are underlined.

| Method | iNaturalist | Textures | OpenImage-O | Places | Average |
|---|---|---|---|---|---|
| MSP | 71.78 / 64.51 | 78.82 / 62.20 | 72.52 / 62.27 | 78.71 / 57.55 | 75.46 / 61.63 |
| MLS | 70.55 / 62.07 | 72.11 / 65.46 | 68.68 / 62.27 | 77.60 / 58.03 | 72.23 / 61.96 |
| EBO | 70.94 / 60.64 | 72.18 / 65.71 | 68.96 / 61.59 | 77.97 / 57.75 | 72.51 / 61.42 |
| ReAct | 53.59 / 75.05 | 59.92 / 72.13 | 62.45 / 69.16 | 71.14 / 66.87 | 61.78 / 70.80 |
| ASH | 53.84 / 74.07 | 47.67 / 79.09 | 60.66 / 70.12 | 71.45 / 64.40 | 58.40 / 71.92 |
| SCALE | 47.82 / 76.79 | 42.04 / 80.95 | 59.26 / 71.76 | 70.32 / 65.86 | 54.86 / 73.84 |
| BFAct | 53.23 / 74.86 | 58.84 / 71.86 | 62.33 / 68.97 | 70.67 / 66.17 | 61.27 / 70.47 |
| LTS | 49.64 / 76.17 | 43.87 / 80.60 | 59.32 / 71.46 | 70.85 / 65.35 | 55.92 / 73.39 |
| OptFS | 51.52 / 75.52 | 48.88 / 76.96 | 59.99 / 70.28 | 70.86 / 65.15 | 57.81 / 71.98 |
| **AdaSCALE-A** | **44.77** / **77.51** | **42.17** / **80.73** | 53.58 / **73.90** | **67.14** / **66.84** | **51.91** / **74.75** |
| **AdaSCALE-L** | 46.52 / 76.39 | 44.87 / 79.90 | **53.32** / 73.81 | 68.16 / 65.88 | 53.22 / 73.99 |

Table 60: Far-FSOOD detection results (FPR@95↓ / AUROC↑) on
ImageNet-1k benchmark using RegNet-Y-16 network. The best results are
**bold**, and the second-best results are underlined.

| Method | iNaturalist | Textures | OpenImage-O | Places | Average |
|---|---|---|---|---|---|
| MSP | 53.98 / 80.49 | 69.36 / 70.97 | 62.02 / 75.95 | 75.48 / 65.76 | 65.21 / 73.29 |
| MLS | 39.99 / 85.09 | 69.12 / 73.89 | 58.30 / 80.08 | 79.98 / 66.96 | 61.85 / 76.51 |
| EBO | 38.42 / 86.11 | 68.23 / 74.16 | 58.77 / 80.93 | 80.56 / 67.08 | 61.49 / 77.07 |
| ReAct | 43.95 / 85.15 | 66.24 / 73.34 | 68.23 / 77.80 | 88.79 / 57.47 | 66.80 / 73.44 |
| ASH | 61.03 / 79.08 | 58.16 / 80.78 | 80.03 / 74.57 | 81.60 / 67.74 | 70.21 / 75.54 |
| SCALE | 39.13 / 86.34 | 56.30 / 79.93 | 60.66 / 81.10 | 76.25 / 70.04 | 58.09 / 79.35 |
| BFAct | 52.17 / 82.68 | 71.55 / 70.74 | 78.37 / 74.00 | 90.50 / 56.09 | 73.15 / 70.88 |
| LTS | 38.98 / 86.90 | 50.10 / 82.26 | 65.29 / 81.17 | 75.41 / **70.91** | 57.44 / 80.31 |
| OptFS | 52.61 / 82.40 | 62.47 / 76.57 | 66.68 / 76.79 | 88.01 / 56.14 | 67.44 / 72.98 |
| **AdaSCALE-A** | 34.25 / 87.68 | 63.81 / 76.12 | **50.37** / 82.13 | 74.81 / 68.37 | 55.81 / 78.58 |
| **AdaSCALE-L** | **32.72** / **88.82** | **47.93** / **82.84** | 53.84 / **83.09** | **73.90** / 70.37 | **52.10** / **81.28** |

Table 61: Far-FSOOD detection results (FPR@95↓ / AUROC↑) on
ImageNet-1k benchmark using ResNeXt-50 network. The best results are
**bold**, and the second-best results are underlined.

| Method | iNaturalist | Textures | OpenImage-O | Places | Average |
|---|---|---|---|---|---|
| MSP | 68.90 / 66.67 | 80.91 / 60.21 | 71.91 / 63.25 | 78.58 / 57.84 | 75.07 / 62.00 |
| MLS | 64.59 / 65.49 | 76.51 / 62.51 | 67.70 / 63.96 | 79.89 / 56.90 | 72.17 / 62.21 |
| EBO | 64.95 / 64.23 | 76.59 / 62.61 | 68.00 / 63.52 | 79.80 / 56.39 | 72.34 / 61.69 |
| ReAct | 51.97 / 76.08 | 65.34 / 68.27 | 63.05 / 68.94 | 70.13 / **67.58** | 62.62 / 70.22 |
| ASH | 53.84 / 74.07 | 47.67 / 79.09 | 60.66 / 70.12 | 71.45 / 64.40 | 58.40 / 71.92 |
| SCALE | 49.27 / 76.82 | 60.07 / 75.14 | 62.91 / 70.69 | 73.38 / 64.22 | 61.41 / 71.72 |
| BFAct | 53.23 / 74.86 | 58.84 / 71.86 | 62.33 / 68.97 | 70.67 / 66.17 | 61.27 / 70.47 |
| LTS | 49.64 / 76.17 | 48.77 / 79.13 | 62.43 / 70.17 | 70.85 / 65.35 | 56.41 / 73.55 |
| OptFS | 51.52 / 75.77 | 50.25 / 75.59 | 62.46 / 68.87 | 69.84 / 65.65 | 57.99 / 71.81 |
| **AdaSCALE-A** | **43.82** / **78.47** | **45.55** / **79.64** | 56.43 / **72.85** | 66.13 / 67.54 | **52.33** / **74.89** |
| **AdaSCALE-L** | 46.23 / 76.67 | 54.29 / 75.80 | 56.84 / 72.44 | 69.13 / 64.99 | 56.62 / 72.47 |

Table 62: Far-FSOOD detection results (FPR@95↓ / AUROC↑) on
ImageNet-1k benchmark using DenseNet-201 network. The best results are
**bold**, and the second-best results are underlined.

| Method | iNaturalist | Textures | OpenImage-O | Places | Average |
|---|---|---|---|---|---|
| MSP | 68.90 / 66.67 | 80.91 / 60.21 | 71.91 / 63.25 | 78.58 / 57.84 | 75.07 / 62.00 |
| MLS | 64.59 / 65.49 | 76.51 / 62.51 | 67.70 / 63.96 | 79.89 / 56.90 | 72.17 / 62.21 |
| EBO | 64.95 / 64.23 | 76.59 / 62.61 | 68.00 / 63.52 | 79.80 / 56.39 | 72.34 / 61.69 |
| ReAct | 51.97 / 76.08 | 65.34 / 68.27 | 63.05 / 68.94 | 70.13 / **67.58** | 62.62 / 70.22 |
| ASH | 53.84 / 74.07 | **47.67** / 79.09 | 60.66 / 70.12 | 71.45 / 64.40 | 58.40 / 71.92 |
| SCALE | **49.27** / **76.82** | 60.07 / 75.14 | 62.91 / 70.69 | 73.38 / 64.22 | 61.41 / 71.72 |
| BFAct | 53.23 / 74.86 | 58.84 / 71.86 | 62.33 / 68.97 | 70.67 / 66.17 | 61.27 / 70.47 |
| LTS | 49.64 / 76.17 | 48.77 / **79.13** | 62.43 / 70.17 | 70.85 / 65.35 | **56.41** / **73.55** |
| OptFS | 51.52 / 75.77 | 50.25 / 75.59 | 62.46 / 68.87 | **69.84** / 65.65 | 57.99 / 71.81 |
| **AdaSCALE-A** | 52.55 / 73.62 | 54.70 / 76.19 | **58.11** / 71.31 | 77.77 / 58.83 | 60.78 / 69.99 |
| **AdaSCALE-L** | 53.04 / 73.62 | 51.83 / 78.00 | 58.30 / **71.92** | 78.40 / 58.47 | 60.39 / 70.50 |

Table 63: Far-FSOOD detection results (FPR@95↓ / AUROC↑) on
ImageNet-1k benchmark using EfficientNetV2-L network. The best results
are **bold**, and the second-best results are underlined.

| Method | iNaturalist | Textures | OpenImage-O | Places | Average |
|---|---|---|---|---|---|
| MSP | 53.98 / 80.49 | 69.36 / 70.97 | 62.02 / 75.95 | 75.48 / 65.76 | 65.21 / 73.29 |
| MLS | 39.99 / 85.09 | 69.12 / 73.89 | 58.30 / 80.08 | 79.98 / 66.96 | 61.85 / 76.51 |
| EBO | **38.42** / 86.11 | 68.23 / 74.16 | 58.77 / 80.93 | 80.56 / 67.08 | 61.49 / 77.07 |
| ReAct | 43.95 / 85.15 | 66.24 / 73.34 | 68.23 / 77.80 | 88.79 / 57.47 | 66.80 / 73.44 |
| ASH | 61.03 / 79.08 | 58.16 / 80.78 | 80.03 / 74.57 | 81.60 / 67.74 | 70.21 / 75.54 |
| SCALE | 39.13 / 86.34 | 56.30 / 79.93 | 60.66 / 81.10 | 76.25 / 70.04 | 58.09 / 79.35 |
| BFAct | 52.17 / 82.68 | 71.55 / 70.74 | 78.37 / 74.00 | 90.50 / 56.09 | 73.15 / 70.88 |
| LTS | 38.98 / 86.90 | 50.10 / 82.26 | 65.29 / 81.17 | 75.41 / **70.91** | 57.44 / 80.31 |
| OptFS | 52.61 / 82.40 | 62.47 / 76.57 | 66.68 / 76.79 | 88.01 / 56.14 | 67.44 / 72.98 |
| **AdaSCALE-A** | 40.76 / **88.62** | 63.95 / 77.69 | **53.96** / **84.71** | 77.17 / 69.22 | 58.96 / 80.06 |
| **AdaSCALE-L** | 43.28 / 87.91 | 50.23 / **83.05** | 57.11 / 84.45 | **73.87** / 70.29 | **56.12** / **81.43** |

Table 64: Far-FSOOD detection results (FPR@95↓ / AUROC↑) on ImageNet-1k benchmark using ViT-B-16 network. The best results are **bold**, and the second-best results are underlined.

| Method | iNaturalist | Textures | OpenImage-O | Places | Average |
|---|---|---|---|---|---|
| MSP | 66.12 / **67.29** | 75.49 / 64.02 | 75.32 / 63.46 | 83.84 / **58.77** | 75.19 / 63.39 |
| MLS | 82.34 / 64.58 | 85.83 / 63.52 | 90.12 / 61.11 | 92.95 / 55.08 | 87.81 / 61.07 |
| EBO | 87.94 / 59.51 | 88.03 / 62.20 | 91.84 / 57.54 | 94.14 / 50.68 | 90.49 / 57.48 |
| ReAct | 70.31 / 62.56 | 75.49 / 64.90 | 76.64 / 61.30 | 87.02 / 54.77 | 77.37 / 60.88 |
| ASH | 93.23 / 53.23 | 95.19 / 51.16 | 90.38 / 57.95 | 89.04 / 56.56 | 91.96 / 54.72 |
| SCALE | 90.13 / 55.74 | 88.71 / 61.09 | 92.30 / 55.21 | 94.75 / 47.62 | 91.47 / 54.92 |
| BFAct | 66.39 / 62.89 | 71.84 / 65.55 | 71.56 / 62.19 | 84.23 / 55.84 | 73.51 / 61.62 |
| LTS | 71.01 / 67.15 | 78.15 / 65.01 | 82.72 / 60.98 | 86.73 / 56.56 | 79.65 / 62.43 |
| OptFS | **62.89** / 66.41 | **70.96** / **65.68** | **68.25** / **64.30** | 80.02 / 58.53 | **70.53** / **63.73** |
| **AdaSCALE-A** | 65.49 / 65.27 | 74.75 / 63.69 | 69.79 / 63.49 | **79.93** / 57.47 | 72.49 / 62.48 |
| **AdaSCALE-L** | 64.72 / 64.84 | 74.72 / 63.49 | 69.92 / 62.86 | 79.99 / 57.04 | 72.34 / 62.06 |

Table 65: Far-FSOOD detection results (FPR@95↓ / AUROC↑) on ImageNet-1k benchmark using Swin-B network. The best results are **bold**, and the second-best results are underlined.

| Method | iNaturalist | Textures | OpenImage-O | Places | Average |
|---|---|---|---|---|---|
| MSP | 73.78 / 70.69 | 87.48 / 63.62 | 88.58 / 65.05 | 86.43 / 62.22 | 84.07 / 65.39 |
| MLS | 94.01 / 64.59 | 94.98 / 61.02 | 97.72 / 57.01 | 95.29 / 54.55 | 95.50 / 59.29 |
| EBO | 95.11 / 55.73 | 95.35 / 58.70 | 97.99 / 49.71 | 95.87 / 47.30 | 96.08 / 52.86 |
| ReAct | 66.17 / 68.53 | 79.27 / 66.85 | 76.92 / 65.82 | 85.99 / 57.89 | 77.09 / 64.77 |
| ASH | 94.55 / 47.18 | 94.43 / 48.22 | 93.77 / 48.28 | 92.55 / 48.99 | 93.82 / 48.17 |
| SCALE | 91.23 / 53.32 | 91.15 / 60.76 | 91.87 / 57.44 | 87.05 / 58.13 | 90.32 / 57.41 |
| BFAct | **54.53** / **73.59** | **69.73** / **68.65** | **59.76** / **74.29** | **74.09** / **64.30** | **64.53** / **70.21** |
| LTS | 72.97 / 70.44 | 86.21 / 62.18 | 83.29 / 63.87 | 89.39 / 56.32 | 82.96 / 63.20 |
| OptFS | 58.91 / 71.84 | 72.14 / 68.00 | 62.45 / 72.16 | 77.16 / 63.35 | 67.66 / 68.83 |
| **AdaSCALE-A** | 61.33 / 68.58 | 79.90 / 64.03 | 65.40 / 67.52 | 77.68 / 59.21 | 71.08 / 64.83 |
| **AdaSCALE-L** | 60.15 / 70.58 | 78.72 / 65.23 | 64.83 / 68.17 | 76.43 / 60.64 | 70.03 / 66.15 |

## J    VISUALIZATIONS OF SCORE SEPARATION

We present comparative visualizations of the energy score separation between SCALE and AdaS-CALE using the RegNet-Y-16 architecture in terms of challenging near-OOD datasets below. As illustrated in Figure 6, AdaSCALE demonstrates a distinctively larger separation between ID and OOD score distributions compared to SCALE.

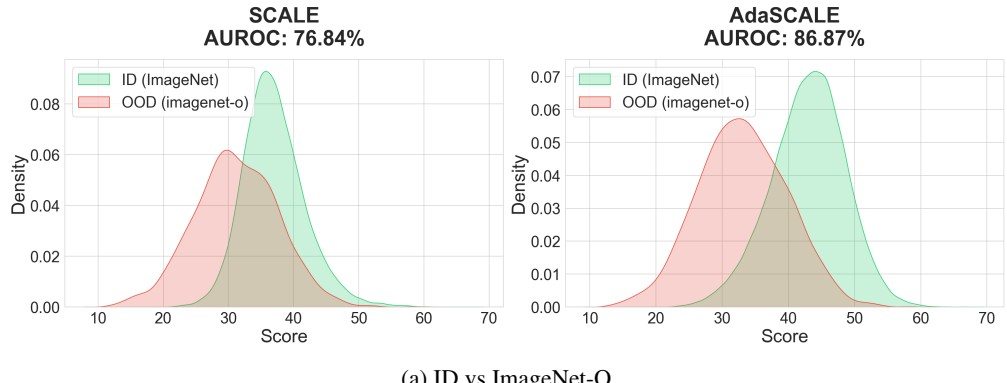

(a) ID vs ImageNet-O

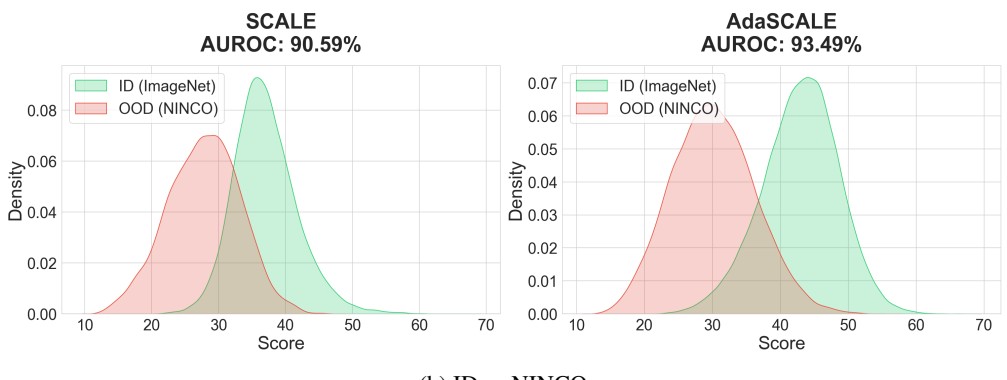

(b) ID vs NINCO

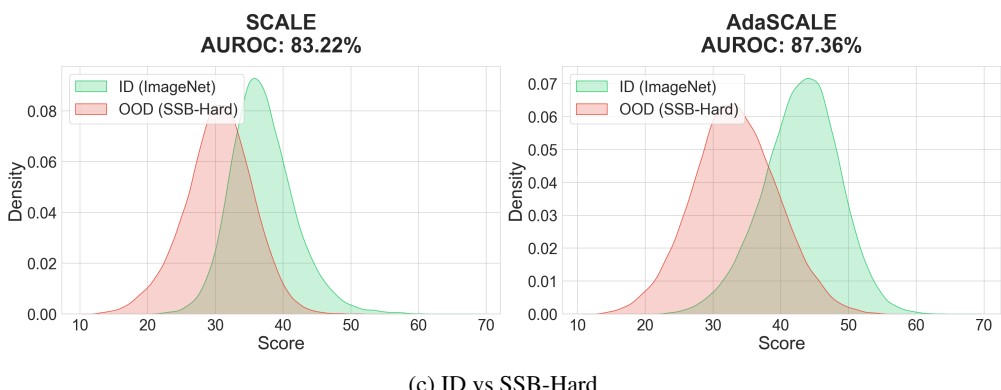

(c) ID vs SSB-Hard

Figure 6: Comparative visualizations of score separation. AdaSCALE consistently shows reduced overlap between ID and OOD distributions compared to SCALE.

