# OpenReview forum: "AdaSCALE: Adaptive Scaling for OOD detection"
_ICLR.cc/2026/Conference — Submitted to ICLR 2026_

### Official Review · Reviewer_upif · 2025-10-26

**Soundness:** 3
**Presentation:** 3
**Contribution:** 2
**Rating:** 4
**Confidence:** 4

**Summary:**

This paper proposes AdaSCALE, an adaptive scaling procedure
that dynamically adjusts the percentile threshold based on a sample’s estimated
OODness. This estimation leverages a key observation: OOD samples exhibit
significantly more pronounced activation shifts at high-magnitude activations under
minor perturbation compared to ID samples. AdaSCALE achieves state-of-the-art OOD detection
performance, outperforming the latest rival OptFS by 14.94% in near-OOD and
21.67% in far-OOD datasets in average FPR@95 metric on the ImageNet-1k
benchmark across eight diverse architectures.

**Strengths:**

1.	Clear motivation and novel observation: OOD samples exhibit greater sensitivity of high-magnitude activations to minor perturbations, providing an intuitive and effective signal for designing an adaptive mechanism.
2.	Overall, AdaSCALE  is  a simple yet effective method.
3.	The experiments  quite extensive with significant results: across 10 architectures and 3 datasets by tuning mere one hyperparameter for a given setup.

**Weaknesses:**

1. AdaSCALE shares a similar design  with ATS (Krumpl et al., 2024), as both methods adaptively adjust scaling parameters during post-processing to enhance ID and OOD separability. It would be better to further clarify the differences in design rationale and implementation mechanisms between the two approaches and provide a more in-depth comparative analysis.

2. The experiments employ gradient-based perturbations, which significantly increase computational overhead and compromise practicality. It remains unclear how applicable this approach would be in real-time systems such as autonomous driving.

3. Although the paper claims that hyperparameters are transferable, p_min and p_max still require manual tuning for each model and dataset.

4. The paper lacks  visual examples (e.g., t-SNE plots or histograms of energy scores) illustrating the improved separability between ID) and OOD samples.

5. The core hypothesis of the paper that OOD samples exhibit greater instability in high-magnitude activations under minor perturbations is currently supported only by intuitive reasoning, lacking theoretical justification or references to prior work.

6. The implementation details are mostly complete, but the perturbation setup needs more clarification.

**Questions:**

Please refer to the Weaknesses. The overall experimental evaluation is very comprehensive; however, it would be valuable  to further address above questions to strengthen the paper’s clarity and I will take the rebuttal into consideration when revising my final score.

---

> ### Author Response · Authors · 2025-11-18
>
> We thank Reviewer upif for the constructive feedback and for recognizing our work's **"clear motivation,"** **"novel observation,"** and **"extensive experiments with significant results."** We are glad the reviewer found AdaSCALE to be a **"simple yet effective method."**
>
> We have addressed all concerns below.
>
> ### 1. Comparison with ATS (Krumpl et al., 2024)
>
> **TLDR: AdaSCALE is fundamentally different from ATS in its core mechanism. AdaSCALE uses two forward passes to leverage a new OOD signal and a single eCDF on the final layer, whereas ATS uses one pass and relies on multiple eCDFs from intermediate layers to directly compute a scaling factor.**
>
> While we mention ATS in our related work, we appreciate the opportunity to provide a more in-depth comparison.
>
> *   **Number of passes:** AdaSCALE uses two forward passes (on the original and perturbed input) to leverage an additional independent OOD detection signal ($Q’$) in its scaling mechanism. In contrast, ATS uses only one forward pass.
> *   **eCDF Usage:** ATS relies on multiple empirical cumulative distribution functions (eCDFs), building one for **each selected intermediate layer**. AdaSCALE constructs a single eCDF from its final OODness score ($Q’$) in the **last layer only**. We hypothesize that the final layer is most effective for this, as deeper layers are responsible for semantic discrimination.
> *   **Role of eCDF:** In ATS, eCDFs are a core part of the inference pipeline, transforming raw activations into p-values that are aggregated to compute the final scaling factor. In AdaSCALE, the single eCDF plays a more indirect, regulatory role: it calibrates the OODness score to determine an *adaptive percentile*, which then informs the final scaling factor calculation.
> *   **Data Requirement for eCDF:** AdaSCALE requires only a very small number of ID samples (as few as 5-10). ATS requires access to the whole training dataset to reliably compute eCDFs for each intermediate layer.
>
> **We will include this detailed comparison in the Appendix. Thanks for this feedback.**
>
> ### 2. Computational Overhead of Gradient-Based Perturbations
>
> **TLDR: The gradient is not necessary. Random pixel perturbation achieves nearly identical performance, making the method practical with only a 1.56x latency increase compared to SCALE/OptFS, which is justified by the significant performance gains.**
>
> As we state in the paper (Lines 229-233):
> > Remark: Is gradient attribution necessary for perturbation? While we employ gradient-based attribution for principled pixel selection for perturbation, as we show later in Section E.5, it is important to note that **even random selection empirically performs similarly**, whereas selecting salient pixels degrades performance.
>
> As shown in Table 22 (Sec E.5), while gradient-based perturbation performs best, random pixel perturbation almost matches its performance. This removes the need for a backward pass, making the method highly practical.
>
> ### 3. Hyperparameter Tuning of p_min and p_max
>
> **TLDR: Tuning only one hyperparameter ($p_\text{max}$) is sufficient, as the performance difference compared to tuning both is trivial. This aligns with the standard practice of our direct predecessors (SCALE/LTS/ASH).**
>
> We agree that minimizing manual tuning is important. Our experiments show that fixing $p_\text{min}$ and tuning only $p_\text{max}$ results in a minimal performance difference, confirming that AdaSCALE generalizes well with just one tunable hyperparameter.
>
> | Hyperparameters tuned | **ImageNet-1k Near-OOD** | **ImageNet-1k Far-OOD** | **CIFAR-100 Near-OOD** | **CIFAR-100 Far-OOD** |
> | :--- | :---: | :---: | :---: | :---: |
> | Tune ($p_\text{min}$, $p_\text{max}$) | **61.17** / **80.50** | **30.11** / **92.73** | **57.33** / **81.35** | **54.53** / **81.14** |
> | Tune ($p_\text{max}$ only) | 62.29 / 79.72 | 32.72 / 91.82 | 58.00 / 81.14 | 56.47 / 80.99 |
>
> This practice is consistent with prior works like ASH, SCALE, and LTS, which also tune a single percentile hyperparameter.
>
> ### 4. Lack of Visual Examples
>
> Thank you for this suggestion. We will add visualizations of the energy score distributions to the appendix to clearly illustrate the improved separability between ID and OOD samples.

---

> > ### Author Response · Authors · 2025-11-18
> >
> > ### 5. Clarification on Perturbation
> >
> > We reiterate that gradient attribution is not necessary for selecting pixels to perturb; random selection performs similarly.
> > For the sake of complete clarity, we provide the PyTorch code for our perturbation function below:
> >
> > ```python
> > @torch.no_grad()
> > def perturb(self, data, grad):
> >     batch_size, channels, height, width = data.shape
> >     n_pixels = int(channels * height * width * self.o)
> >     abs_grad = abs(grad).view(batch_size, channels * height * width)
> >     # Select pixels with the SMALLEST gradients for perturbation
> >     _, topk_indices = torch.topk(abs_grad, n_pixels, dim=1, largest=False)
> >     mask = torch.zeros_like(abs_grad, dtype=torch.uint8)
> >     mask.scatter_(1, topk_indices, 1)
> >     mask = mask.view(batch_size, channels, height, width)
> >     # Add noise to the selected pixels
> >     data_ood = data + grad.sign() * mask * 0.5
> >     return data_ood
> > ```
> >
> > ---
> > We hope these clarifications address the reviewer's concerns, and we would be grateful if the reviewer would consider raising their score.

---

> > > ### Author Response · Authors · 2025-11-21
> > >
> > > ### On Theoretical Justification
> > >
> > > We agree that a deeper theoretical justification would be a valuable contribution and an important direction for future research. While our work does not offer a formal proof, we believe the core phenomenon is highly intuitive and follows directly from well-established observations in the field. We further attempt to explain our intuition below:
> > >
> > > *   **ID Stability:** Since a model is trained on the ID distribution, we expect it to be robust to minor, trivial perturbations of ID inputs. It has learned the underlying data manifold, so small changes should not drastically alter the high-level semantic features, leading to a stable activation pattern.
> > > *   **OOD Instability:** The model has no explicit knowledge of OOD data. From its perspective, a minor perturbation applied to an OOD input can create what it perceives as a *completely different* OOD input. We can borrow from the well-known observation from ReAct that "OOD inputs have abnormally high activations."
> > > *   **The Shift in High Activations:** Because the original OOD input and the trivially perturbed OOD input are seen as two distinct, unfamiliar samples by the model, both will likely trigger "abnormally high activations." However, the probability that these high activations occur in the exact same set of neurons is low. This may result in a significant shifts of the highest-magnitude activations, which our method effectively measures.
> > >
> > > We are more than happy to offer further clarifications. We are grateful for the reviewer for encouraging us to further clarify our intuition behind the activation shift at high-magnitude activations.

---

> > > > ### Author Response · Authors · 2025-11-21
> > > >
> > > > We think it would be helpful to further emphasize our contributions.
> > > >
> > > > As Reviewer Vxks notes, our work **“moves scaling-based OOD detection from heuristic to a more principled footing”** and provides **“a significant conceptual advancement”** by **"discovering an important phenomenon"**. To achieve such significant empirical performance, our work makes the following novel contributions:
> > > >
> > > > *   Our work makes a **proper successful fusion** of three independent OOD detection signals: activation pattern, logits (indirectly or directly) and activation shift at top-k activations ($Q$). Here, identifying activation shift at top-k activations as a possible **independent** OOD detection signal is a novel contribution in itself. We further show that naive usage of $Q$ can lead to relying on overconfident scaling factors and use a correction term to tame the overconfident scaling factors so as to preserve the valuable information contained in logits. (We respectfully direct Reviewer upif to **Sec D.1**. for more detail.)
> > > > *   Furthermore, injecting this identified independent OOD detection signal through a dynamic percentile hyperparameter is a non-trivial addition, though in hindsight it may look like a straightforward addition.
> > > > *   Overall, our work introduces the novel paradigm of adaptive scaling.

---

### Official Review · Reviewer_yGZh · 2025-10-27

**Soundness:** 2
**Presentation:** 3
**Contribution:** 3
**Rating:** 6
**Confidence:** 3

**Summary:**

This paper presents a novel adaptive scaling procedure for OOD detection that dynamically adjusts the detection threshold based on the likelihood of OODness for individual samples. Through extensive experiments, the superiority of the method against several baselines is established.

**Strengths:**

The paper is generally easy to follow, and most of the intuition for the proposed methodology and the different components of the algorithm are empirically justified.

The authors provide good figures to better convey the empirical results and justifications

The authors provide a good set of experiments and ablation studies that examine different properties of the algorithm and different sensitivities.

**Weaknesses:**

Despite the easy flow of the text, some parts are abstract and require some more detailed domain knowledge, which could have been better established. This mainly pertains to Fig 1, the introduction, and sec 4.1.

Some parts of the paper seem to be stating contradicting requirements and observations. This could also be my lack of familiarity with the different terminology and their differences:
- In the preliminaries, it is mentioned that higher scores are for OOD samples. Yet the paragraph at 84 talks about giving lower scores to OODs. Same comment regarding 4.1.
- 284: "Fig 3 illustrates Q'/Q' > Q/Q", whereas the figure shows the opposite of this. Also, in Fig 3 caption.

There are some baselines that are missing. Based on my own experiments, it is critical that the authors **must** include a comparison with LINe [1] for my final decision. Other baselines can be found in [2]

Although I understand the regime of the experimental setup and its comparison with similar post-hoc methods, a comparison with methods that assume access to OOD data during training could also be useful. It is not necessary for the methodology of this paper to beat such methods, but a comparison with such work could be useful.


[1] Ahn YH, Park GM, Kim ST. Line: Out-of-distribution detection by leveraging important neurons. In2023 IEEE/CVF Conference on Computer Vision and Pattern Recognition (CVPR) 2023 Jun 17 (pp. 19852-19862). IEEE.

[2] https://github.com/Jingkang50/OpenOOD

**Questions:**

If the absolute perturbation sum over activations is indeed such a good measure of the likelihood of OODness, why not use this metric itself as the scoring function? Why do you first utilize this to acquire OODness and then utilize that information for another scoring function?

Also see weaknesses.

---

> ### Author Response · Authors · 2025-11-18
>
> We thank Reviewer yGZh for their positive feedback, noting that the paper is **“easy to follow,”** with **“good figures”** and a **“good set of experiments and ablation studies.”**
>
> We believe the weaknesses identified are primarily due to minor typos and areas that can be clarified. We have addressed all concerns below.
>
> ### 1. On "Lower vs. Higher" Energy Scores
>
> You are correct to point out the potential for confusion. Because the energy score is computed with a negative sign ( $-\log \sum_{i=1}^C e^{(\mathbf{z}^{\text{scaled}}_i)}$ ), our intention was to say we apply “smaller scaling factors for samples with high OODness to yield **lower-magnitude** energy scores” and “larger scaling factors for probable ID samples to yield **higher-magnitude** energy scores.” We appreciate you flagging this and have corrected the wording in the paper for clarity.
>
> ### 2. On the Typo in Figure 3 Caption
>
> Thank you for pointing this out. This is indeed a typo in the text. We have corrected the text to state $Q^{\prime}_\text{OOD}/Q^{\prime}_\text{ID} < Q_\text{OOD}/Q_\text{ID}$, which accurately reflects the relationship demonstrated in Figure 3.
>
> ### 3. On Missing Baselines
>
> We agree that a comparison with LINe is valuable. We have run the experiments, and the results below (FPR@95↓ / AUROC↑) show that AdaSCALE consistently and significantly outperforms LINe across all tested architectures.
>
> | **Category** | **Metric** | **ResNet-50** | **ResNet-101** | **RegNet-Y-16** | **ResNeXt-50** | **DenseNet-201** | **EfficientNetV2-L** | **ViT-B-16** | **Swin-B** | **Average** |
> | :--- | :--- | :--- | :--- | :--- | :--- | :--- | :--- | :--- | :--- | :--- |
> | near-OOD | LINe | 79.10/67.20 | 78.63/71.46 | 86.56/63.61 | 78.17/71.47 | 87.14/63.37 | 95.01/50.13 | 93.01/57.97 | 93.80/57.91 | 86.43/62.89 |
> | | AdaSCALE-A | **58.98/78.98** | **57.96/81.68** | **47.91/89.18** | **64.14/79.96** | **61.28/79.66** | **53.78/86.94** | **71.87/73.14** | **73.41/74.48** | **61.17/80.50** |
> | far-OOD | LINe | 32.95/93.31 | 36.95/92.46 | 64.11/82.13 | 38.72/91.99 | 58.87/84.59 | 96.97/49.09 | 91.05/61.32 | 92.93/68.02 | 64.07/77.86 |
> | | AdaSCALE-A | **17.84/96.14** | **18.51/95.95** | **21.37/95.84** | **22.08/95.24** | **28.01/93.23** | **37.61/91.48** | **47.63/86.83** | **47.81/87.14** | **30.11/92.73** |
>
> While LINe shows acceptable performance in some far-OOD settings, AdaSCALE is clearly superior. We see this as a relevant addition in experimental comparison. We also will revise the paper to include it in the related works section. We will also add other baselines mentioned in OpenOOD to the Appendix.
>
> Regarding methods that use OOD training data, we have limited the scope of our study to post-hoc methods. As noted by the OpenOOD v1.5 [1], post-hoc methods often surpass training-based methods in large-scale settings like ImageNet-1k, which is the focus of our work.
>
> [1]  "Openood v1. 5: Enhanced benchmark for out-of-distribution detection." arXiv preprint arXiv:2306.09301 (2023).

---

> > ### Author Response · Authors · 2025-11-18
> >
> > ### 4. Question: Why Not Use the Activation Shift (Q) Directly as the Score? / Design rationale
> >
> > **TDLR: As mentioned in Line 425 - 426, estimated OODness alone — without adaptive scaling — does not result in strong performance. AdaSCALE works very well because it allows the seamless fusion of three independent and effective OOD detection signals: activation pattern, logit (directly or indirectly), and $Q’$ (which we introduce through _adaptive percentile_ instead of using fixed percentile).**
> >
> > **The Goal: Maximizing ID/OOD Score Separation**. The ultimate objective is to maximize the separation between the final energy scores of ID and OOD samples. This is most effectively achieved by applying a **stronger scaling factor** to _ID-likely_ samples and a **weaker scaling factor** to _OOD-likely_ samples.
> >
> > Fixed Scaling (for ex. LTS): In LTS, the logits are scaled with scaling factor $r$ as: $\mathbf{z}^{\text{scaled}} = \mathbf{z} \cdot r^{2}$  energy score is $E = -\log \sum_{i=1}^C e^{(\mathbf{z}^{scaled}_i)}$. This already fuses two independent OOD signals: (1) activation pattern, via $r$, and (2) the logits $\mathbf{z}$.
> >
> > **The Mechanism: The Need of Dynamic Percentile ( $p$ )**. Recall the static scaling formula, $r = sum(a) / sum(a > P_p(a))$. To achieve a larger scaling factor $r$, the denominator must be smaller. This happens when the percentile threshold $p$ is set to a higher value, as more high-magnitude activations are excluded from the sum.
> >
> > - To achieve **stronger scaling for ID-likely** (our goal), they should be assigned a **higher percentile** $p$.
> > - To achieve **weaker scaling for OOD-likely**, they should be assigned a **lower percentile** $p$.
> >
> > **The Inevitable Conclusion: The Need for Dynamic Percentiles**. The reasoning above leads directly to the conclusion that a static, fixed percentile $p$ for all samples is inherently suboptimal. The percentile should ideally be dynamic and adaptive, conditioned on whether a sample is likely to be ID or OOD. This is the foundational motivation for moving beyond prior work like SCALE and ASH.
> >
> > **The Solution: Designing an OOD Likelihood Heuristic ( $Q'$ )**. To enable this dynamic adjustment, we need a reliable signal to estimate a sample's OODness before scaling. This is precisely the purpose of our proposed metric $Q'$. As detailed in Section 4.1, $Q'$ is designed based on the key observation that OOD samples exhibit pronounced activation shifts under minor perturbations. Therefore:
> >
> > - A **high $Q'$** value indicates a high OODness.
> > - A **low $Q'$** value indicates a high IDness.
> >
> > **Synthesizing the Components into AdaSCALE**. We now have all the logical pieces:
> >
> > **Goal**: High $p$ for ID-likely samples, low $p$ for OOD-likely samples.
> >
> > **Tool**: $Q'$ as an estimate of OODness.
> >
> > We design the adaptive percentile calculation (Equation 9) to directly implement this inverse relationship: $p = p_\text{min} + (1 - F_{Q'}(Q')) * (p_\text{max} - p_\text{min})$
> >
> > Here, the $(1 - F_Q'(Q'))$ term is critical. When a sample has a high $Q'$ (is OOD-likely), its $F_Q'(Q')$ is high, making $(1 - F_Q'(Q'))$ low. This pushes $p$ towards $p_\text{min}$. Conversely, for a low $Q'$ (ID-likely) sample, $p$ is pushed towards $p_\text{max}$.
> >
> > In essence, **the design of AdaSCALE was a principled effort to inject another independent source of OOD signal** (activation shift at the expense of extra forward pass) directly into the scaling mechanism itself. While prior methods use a static mechanism to amplify an existing signal, AdaSCALE makes the amplification mechanism itself intelligent and adaptive to the nature of each sample.
> >
> > ---
> > We thank the reviewer again for their constructive feedback and sincerely hope these clarifications will be considered in the final evaluation.

---

> ### Author Response · Authors · 2025-11-22
>
> We have updated the Appendix by including comparison with LINe, MDS and NNGuide. We request the reviewer to let us know if comparison with any other specific method is needed. Also, we have included LINe in the related works section.

---

### Official Review · Reviewer_Vxks · 2025-10-30

**Soundness:** 3
**Presentation:** 3
**Contribution:** 3
**Rating:** 8
**Confidence:** 3

**Summary:**

This paper proposes AdaSCALE, a post-hoc OOD detection method that adaptively adjusts activation scaling based on the estimated OODness of each test sample. The method is built on an empirical observation: OOD samples exhibit larger shifts in top-k activations under small input perturbations, while ID samples remain stable. Leveraging this phenomenon, the authors compute a perturbation-induced activation shift statistic, convert it into a normalized OODness score using an empirical CDF with only a few ID samples, and dynamically determine the percentile threshold for scaling. Extensive experiments across CIFAR and ImageNet-1k, eight architectures, FSOOD settings, corrupted inputs, and adversarially trained networks demonstrate consistent SOTA performance and strong generalization. The method is parameter-efficient, requires minimal ID samples, and preserves ID accuracy.

**Strengths:**

- The paper provides a compelling mechanistic explanation for why activation scaling works for OOD detection: semantic activations are stable for ID samples but unstable for OOD samples under perturbations. This moves scaling-based OOD detection from heuristic to a more principled footing. I consider this a significant conceptual advancement.

-Prior scaling methods use a static percentile threshold; adapting it per-sample based on estimated OODness is a meaningful and novel extension.

- Demonstrates consistent SOTA performance across: CIFAR & ImageNet-1k and 8 architectures

- only need a very small number of ID samples (as few as 10) for calibration and preserves accuracy,.

- No model retraining or gradients beyond attribution; easy to integrate into existing systems.

**Weaknesses:**

- Although trivial/random perturbation works, the selection of perturbation magnitude and pixel selection lacks theoretical grounding, and different perturbation strategies may influence results.

- Requires an extra forward pass + top-k operations, resulting in ~2–4× latency vs. fixed-scaling baselines, which may be limiting in latency-critical systems.

- The work discovers an important phenomenon, but lacks a deeper theoretical unification of scaling and perturbation in relation to existing methods. Such a theory would substantially enhance the impact.

**Questions:**

See Weakness

---

> ### Author Response · Authors · 2025-11-21
>
> We are sincerely grateful to Reviewer Vxks for their strong support and insightful feedback. We are particularly encouraged that the reviewer found our work to be a **“significant conceptual advancement”** that moves scaling-based methods **“from heuristic to a more principled footing.”**
>
> We appreciate the points raised in the review and offer the following clarifications.
>
> ### 1. On Theoretical Grounding and Unification
>
> We agree that a deeper theoretical unification would be a valuable contribution and an important direction for future research. While our work does not offer a formal proof, we believe the core phenomenon is highly intuitive and follows directly from well-established observations in the field. We further attempt to explain our intuition below:
>
> *   **ID Stability:** Since a model is trained on the ID distribution, we expect it to be robust to minor, trivial perturbations of ID inputs. It has learned the underlying data manifold, so small changes should not drastically alter the high-level semantic features, leading to a stable activation pattern.
> *   **OOD Instability:** The model has no explicit knowledge of OOD data. From its perspective, a minor perturbation applied to an OOD input can create what it perceives as a *completely different* OOD input. We can borrow from the well-known observation from ReAct that "OOD inputs have abnormally high activations."
> *   **The Shift in High Activations:** Because the original OOD input and the trivially perturbed OOD input are seen as two distinct, unfamiliar samples by the model, both will likely trigger "abnormally high activations." However, the probability that these high activations occur in the exact same set of neurons is low. This may result in a significant shifts of the highest-magnitude activations, which our method effectively measures.
>
> ### 2. On Computational Latency
>
> This is a fair point. The additional forward pass (or say additional input in same batch) with top-k operations does introduce latency compared to fixed-scaling baselines. We view this as a direct trade-off between computational cost and detection performance. Given that AdaSCALE achieves consistent state-of-the-art results across a wide range of benchmarks and architectures, we believe the performance gain justifies the additional overhead for many applications.
>
> That said, designing a more efficient adaptive scaling mechanism is an excellent and challenging direction for future work.
>
> We thank the reviewer once again for the positive assessment and valuable suggestions. We are particularly thankful for encouraging us to further articulate our intuition behind the activation shift in high-magnitude activations.

---

### Official Review · Reviewer_GRZb · 2025-10-31

**Soundness:** 2
**Presentation:** 3
**Contribution:** 2
**Rating:** 2
**Confidence:** 3

**Summary:**

The paper addresses out-of-distribution (OOD) detection in deep networks and identifies a limitation of recent post-hoc methods: they use sample-specific activation scaling with a fixed percentile threshold, which is suboptimal for separating in-distribution (ID) vs OOD samples. As models scale up, a static threshold can’t account for the varying “OOD-ness” of each sample.
AdaSCALE is proposed as an adaptive scaling procedure that adjusts the scaling percentile per sample based on an estimated OOD likelihood. The core insight is that OOD inputs exhibit larger shifts in their top activations under small perturbations compared to ID inputs.
By quantifying this activation shift (via a gradient-based input perturbation and measuring changes in high-magnitude activations), the method assigns a higher percentile (stronger scaling) to inputs likely ID and a lower percentile (weaker scaling) to those likely OOD. This yields more separable energy scores for detection, effectively reducing OOD confidence for OOD samples while preserving ID confidence.

**Strengths:**

- The method demonstrates substantial performance gains. AdaSCALE achieves state-of-the-art OOD detection results on challenging benchmarks (ImageNet-1k), significantly outperforming prior post-hoc methods. For example, it dramatically lowers false positive rates (FPR@95) compared to strong baselines (e.g., vs. OptFS and SCALE) in both near-OOD and far-OOD settings.
- The paper shows robust generalization on several different network architectures (ResNets, EfficientNet-V2, ViT, etc.), where AdaSCALE outperforms or matches previous methods on each, addressing a known shortcoming of some prior methods that were tied to specific models.
- The paper is well-structured and clear. It provides a detailed algorithm, conceptual diagrams, and ablations. The authors situate AdaSCALE in context by comparing to a wide range of baselines (MSP, ODIN, ReAct, ASH, SCALE, LTS, OptFS, etc.), and perform comprehensive experiments (including metrics like AUROC/FPR, near-vs-far OOD, and even a “full-spectrum” test incorporating covariate shifts. The writing is generally easy to follow, with a solid explanation of the method and the intuition behind it.

**Weaknesses:**

- While the adaptive scaling idea is effective, the contribution can be seen as a relatively incremental extension of existing methods. AdaSCALE builds directly on the activation scaling paradigm introduced by ASH/SCALE/LTS. The key insight (that OOD causes unstable high activations under perturbation) extends the known observation from ReAct that OOD samples have high activation magnitudes. Thus, the method’s novelty, though useful, feels a bit additive: it combines perturbation-based OOD scoring with the existing scaling framework.
- Certain design elements in AdaSCALE appear somewhat heuristic and might lack theoretical justification. For instance, the OOD likelihood score is a composite of the top-k activation shift and a “correction” term for ID bias, weighted by a parameter $\lambda$. This was introduced to counter the high variance in OOD shift metrics, which indicates that the raw signal can be noisy or overestimate OOD likelihood without tuning. The need to hand-craft this combination (including hyperparameters $k_1$, $k_2$, $\lambda$) suggests the solution required empirically balancing factors, rather than deriving a unified criterion. The current design feels overcomplicated and may benefit from simplifications focusing on the most essential theoretical insights.
- AdaSCALE introduces additional hyperparameters (e.g., percentile range $p_{min}$/$p_{max}$, top-k sizes, perturbation size, etc.) and a more complex inference procedure. This raises two concerns:
  - The paper mentions using an automatic search (via OpenOOD) to set these, but it’s unclear how sensitive the results are to these settings. In Table 7, it is shown that AdaSCALE is somewhat insensitive to single-hyperparameter variations with ResNet-50 on ImageNet-1k. However, there is no study involving multiple-hyperparameter variations (e.g., two hyperparameters varying at the same time; considering that AdaSCALE has a lot of hyperparameters) or study on other architectures and datasets.
  - Moreover, it is unclear whether the automatic search (via OpenOOD) utilizes an OOD validation set. If it does, then the current evaluation setting may not be fair to methods that do not require any OOD validation sets. Another related concern is how sensitive AdaSCALE is to any mismatch between the OOD validation set and the actual OOD test set.
- Unlike some post-hoc methods that require no extra data, AdaSCALE does rely on a small set of ID validation samples to compute the empirical CDF for its adaptive threshold. The paper emphasizes this is a minimal requirement (only 10 samples used), but it seems that Table 9 only shows the result from a single experiment (per setting). The impact of using only a handful of samples to estimate the distribution of the OOD score isn’t deeply explored. One might wonder if this calibration could be unstable or biased if those few samples aren’t perfectly representative. This is a minor weakness given how small the requirement is, but it’s worth noting as a practical consideration.
- There is a typo in the formula for the proposed adaptive percentile threshold: a right parenthesis is missing.

**Questions:**

- The adaptive scaling relies on the heuristic that OOD samples’ top activations are unstable under small perturbations. Are there types of OOD inputs or shifts where this heuristic might fail or be less effective? For example, if an OOD sample is very near the ID distribution (or if the model’s features are insensitive to the chosen perturbation), would AdaSCALE risk mis-classifying it as ID? Conversely, could certain ID samples that are borderline or noisy exhibit large activation shifts and be mistaken for OOD?
- OptFS was designed for cross-architecture generalization using a piecewise constant scaling function. AdaSCALE empirically outperforms OptFS across architectures, but could the authors elaborate on *why* adaptivity gives a *significant* edge here? From Figure 2, the signal doesn’t appear to be very clear (the standard deviation regions largely overlap between ID and OOD).

---

> ### Author Response · Authors · 2025-11-18
>
> We appreciate Reviewer GRZb’s detailed feedback and positive comments that our paper is **“well-structured and clear,”** demonstrates **“substantial performance gains,”** and shows **“robust generalization on several different network architectures.”**
>
> We believe the primary concerns raised are due to misunderstandings regarding the method's novelty and design, as well as the stability of the hyperparameter tuning. We have addressed all concerns below and provided new experimental results to support our claims.
>
> ### 1. On Incremental Novelty
>
> **TLDR: Our work introduces a new paradigm of adaptive scaling by successfully fusing three independent OOD detection signals, a non-trivial conceptual advance that leads to state-of-the-art performance.**
>
> While we agree that AdaSCALE builds upon the successful foundation of prior activation-scaling methods (ASH, SCALE, LTS), we respectfully argue that our extension has non-trivial impact. As Reviewer Vxks notes, our work **“moves scaling-based OOD detection from heuristic to a more principled footing”** and provides **“a significant conceptual advancement”** by **"discovering an important phenomenon"**. To achieve such significant empirical performance, our work makes the following novel contributions:
>
> *   Our work makes a **proper successful fusion** of three independent OOD detection signals: activation pattern, logits (indirectly or directly) and activation shift at top-k activations ($Q$). Here, identifying activation shift at top-k activations as a possible **independent** OOD detection signal is a novel contribution in itself. We further show that naive usage of $Q$ can lead to relying on overconfident scaling factors and use a correction term to tame the overconfident scaling factors so as to preserve the valuable information contained in logits. (We respectfully direct Reviewer GRZb to **Sec D.1**. for more detail.)
> *   Furthermore, injecting this identified independent OOD detection signal through a dynamic percentile hyperparameter is a non-trivial addition, though in hindsight it may look like a straightforward addition.
> *   Overall, our work introduces the novel paradigm of adaptive scaling.
>
> ### 2. On the Design of AdaSCALE
>
> **TLDR: Since no signal is perfect (unless it is an oracle signal), a promising strategy is the proper fusion of independent OOD detection signals. Our design thoughtfully balances these signals to enhance detection and is robust enough to leverage even "noisy" signals effectively.**
>
> Before going into AdaSCALE, let's analyze the scaled energy score LTS uses. Taking the example of LTS:
>
> First, the logits are scaled with scaling factor $r$ as:
>
> $\mathbf{z}^{\text{scaled}} = \mathbf{z} \cdot r^{2}$
>
> And then the energy score is computed as:
>
> Energy score = $-\log \sum_{i=1}^C e^{(\mathbf{z}^{\text{scaled}}_i)}$
>
> Here, in the final energy score, there is the contribution of both the activation pattern (through $r$) and logit information (through $\mathbf{z}$). **Neither the activation pattern nor the logit information is an oracle here**. This scoring works well because of the fusion of two sources of independent and effective OOD detection signals. What AdaSCALE does is the **fusion of these sources with another independent OOD detection signal** by the clever introduction of a dynamic percentile.
>
> For instance, let’s suppose $Q’$ = $Q$ and there is no correction factor at all. There would still be the contribution of logit information in the scaled energy score, albeit dominated by $Q$. Seeking only $Q$ for adaptive scaling is, in some way, moving towards searching for an oracle OOD detection signal while moving away from the OOD detection signal contained in logit information. In fact, we believe the framework of AdaSCALE allowing even a “**noisy signal**” for OOD detection enhancement should be seen as a strength rather than a weakness. A noisy signal is okay as long as $Q^{\prime}_{OOD} / Q^{\prime}_{ID}$ is greater than 1, which is a requirement of AdaSCALE (Please see Figure 3). Note: Although we show the activation shift at all indices in Figure 2, we only use the top-1% shift (which is most discriminative).
>
> We think the usage of a phrase such as “correction term” led to some misunderstandings. We will expand more on our design rationale later.

---

> ### Author Response · Authors · 2025-11-18
>
> ### 3. On Hyperparameter Sensitivity Studies
>
> **TLDR: We have conducted new, extensive hyperparameter studies for ViT-B-16, including a multi-hyperparameter sweep, which demonstrates that AdaSCALE is not overly sensitive. We also provide clear intuition for setting each hyperparameter.**
>
> The sensitivity study of the hyperparameters for ViT-B-16 is presented below (FPR@95↓ / AUROC↑):
>
> | $\epsilon$ | Near-OOD | Far-OOD |
> |:---|:---|:---|
> | 0.1 | 72.71 / 73.06 | 46.60 / 87.22 |
> | 0.5 | 71.87 / 73.14 | 47.63 / 86.83 |
> | 1.0 | 72.92 / 73.06 | 49.80 / 86.49 |
>
> | $\lambda$ | Near-OOD | Far-OOD |
> |:---|:---|:---|
> | 1 | 73.92 / 72.22 | 47.15 / 87.15 |
> | 10 | 71.87 / 73.14 | 47.63 / 86.83 |
> | 20 | 72.94 / 73.22 | 51.96 / 85.93 |
> | 30 | 73.72 / 73.06 | 54.55 / 84.16 |
>
> | $k_1$ (%) | Near-OOD | Far-OOD |
> |:---|:---|:---|
> | 1 | 71.87 / 73.14 | 47.63 / 86.83 |
> | 2 | 72.47 / 73.37 | 51.60 / 85.96 |
> | 3 | 73.20 / 73.10 | 54.79 / 85.12 |
>
> | $k_2$ (%) | Near-OOD | Far-OOD |
> |:---|:---|:---|
> | 5 | 71.87 / 73.14 | 47.63 / 86.83 |
> | 10 | 72.06 / 73.26 | 47.55 / 87.03 |
> | 20 | 72.58 / 72.87 | 47.89 / 86.96 |
>
> | $o$ (%) | Near-OOD | Far-OOD |
> |:---|:---|:---|
> | 2 | 72.28 / 73.65 | 47.70 / 87.02 |
> | 5 | 71.87 / 73.14 | 47.63 / 86.83 |
> | 10 | 72.34 / 73.30 | 48.86 / 86.69 |
>
> | $p_\text{max}$ | Near-OOD | Far-OOD |
> |:---|:---|:---|
> | 75 | 76.87 / 71.78 | 52.97 / 86.07 |
> | 80 | 72.60 / 73.24 | 48.57 / 86.78 |
> | 85 | 71.87 / 73.14 | 47.63 / 86.83 |
> | 90 | 72.40 / 73.65 | 48.80 / 86.64 |
> | 95 | 72.35 / 73.63 | 48.73 / 86.69 |
>
> The results show AdaSCALE is not overly sensitive to any single hyperparameter.
>
> Taking time/resource constraint into consideration, we present the joint sensitivity study of 4 hyperparameters in the ViT-B-16 network below with ($k_1=1%$, and $k_2=5%$) **in terms of validation AUROC** (the metric used to choose hyperparameters):
>
> | $\epsilon$ | $\lambda$ | $o$ (%) | $p_\text{max}$ | val AUROC |
> |:---|:---|:---|:---|:---|
> | 0.1 | 1 | 2 | 80 | 87.13 |
> | | | | 85 | 87.55 |
> | | | | 90 | 87.74 |
> | | | 5 | 80 | 87.11 |
> | | | | 85 | 87.53 |
> | | | | 90 | 87.73 |
> | | | 10 | 80 | 87.07 |
> | | | | 85 | 87.47 |
> | | | | 90 | 87.65 |
> | | 10 | 2 | 80 | 87.46 |
> | | | | 85 | 87.74 |
> | | | | 90 | 87.78 |
> | | | 5 | 80 | 87.43 |
> | | | | 85 | 87.71 |
> | | | | 90 | 87.77 |
> | | | 10 | 80 | 87.41 |
> | | | | 85 | 87.72 |
> | | | | 90 | 87.79 |
> | | 20 | 2 | 80 | 87.59 |
> | | | | 85 | 87.77 |
> | | | | 90 | 87.67 |
> | | | 5 | 80 | 87.54 |
> | | | | 85 | 87.73 |
> | | | | 90 | 87.63 |
> | | | 10 | 80 | 87.48 |
> | | | | 85 | 87.68 |
> | | | | 90 | 87.62 |
> | 0.5 | 1 | 2 | 80 | 87.08 |
> | | | | 85 | 87.47 |
> | | | | 90 | 87.65 |
> | | | 5 | 80 | 86.94|
> | | | | 85 | 87.29 |
> | | | | 90 | 87.41 |
> | | | 10 | 80 | 86.65 |
> | | | | 85 | 86.87 |
> | | | | 90 | 86.88 |
> | | 10 | 2 | 80 | 87.16 |
> | | | | 85 | 87.47 |
> | | | | 90 | 87.54 |
> | | | 5 | 80 | 86.92 |
> | | | | 85 | 87.25 |
> | | | | 90 | 87.35 |
> | | | 10 | 80 | 86.76 |
> | | | | 85 | 87.15 |
> | | | | 90 | 87.32 |
> | | 20 | 2 | 80 | 86.86 |
> | | | | 85 | 86.98 |
> | | | | 90 | 86.86 |
> | | | 5 | 80 | 86.40 |
> | | | | 85 | 86.58 |
> | | | | 90 | 86.53 |
> | | | 10 | 80 | 86.09|
> | | | | 85 | 86.43 |
> | | | | 90 | 86.55 |
> | 1.0 | 1 | 2 | 80 | 87.00 |
> | | | | 85 | 87.34 |
> | | | | 90 | 87.45 |
> | | | 5 | 80 | 86.72|
> | | | | 85 | 86.90 |
> | | | | 90 | 86.87 |
> | | | 10 | 80 | 85.96|
> | | | | 85 | 85.81 |
> | | | | 90 | 85.42 |
> | | 10 | 2 | 80 | 86.73 |
> | | | | 85 | 87.09 |
> | | | | 90 | 87.22 |
> | | | 5 | 80 | 86.31 |
> | | | | 85 | 86.71 |
> | | | | 90 | 86.93 |
> | | | 10 | 80 | 85.97 |
> | | | | 85 | 86.42 |
> | | | | 90 | 86.69 |
> | | 20 | 2 | 80 | 86.00 |
> | | | | 85 | 86.23|
> | | | | 90 | 86.23|
> | | | 5 | 80 | 85.27 |
> | | | | 85 | 85.60 |
> | | | | 90 | 85.75 |
> | | | 10 | 80 | 84.69 |
> | | | | 85 | 85.09 |
> | | | | 90 | 85.35 |

---

> ### Author Response · Authors · 2025-11-18
>
> **Comments on hyperparameters:**
>
> *   **$\epsilon$**: Since the mean of RGB statistics is (0.485, 0.456, 0.406), a magnitude of 0.5 is close to those individual means and hence sufficient enough to destroy the pixel information to cause disturbance to peak activations (which is the goal).
> *   **$o$**: $o$ deals with the extent of perturbation to cause a disturbance in peak activations. We just need to make a trivial perturbation. $o$ can be simply adjusted to 5%.
> *   **$\lambda$**: The value of $\lambda$ should be high enough for $Q$ to dominate $C_o$ in forming $Q’$. **As we want to fuse multiple independent OOD detection signals**, $Q$ captures the activation shift while $C_o$ provides relative importance to logit information. If the contribution from $C_o$ is greater than the contribution from $Q$, it leads to scaling OOD strongly and ID weakly (which is exactly opposite to our goal). $\lambda$ can be set to 10 in all cases.
> *   **$k_1$**: Inspired by ReAct, we observe only a few peak activations (e.g., 1%) in OOD show high fluctuations in activations under minor perturbation. $k_1$ can be simply set to 1%.
> *   **$k_2$**: Since $C_o$ is a regularizer to prevent overconfident scaling, it is dependent on $Q$ and thereby $k_1$. Empirically, we find $k_2=5%$ to work well across all architectures to prevent the overconfident scaling factor from overshadowing the OOD detection signal contained in logits.
> *   **$p_\text{min}$**: It can be simply set to a lower limit of percentile tuning (~60), inspired by ASH/SCALE.
> *   **$p_\text{max}$**: Tuning $p_\text{max}$ is exactly like tuning the percentile ($p$) hyperparameter in prior works like ASH/SCALE.
>
> ### 4. On OOD Validation Set and Comparison to OptFS
>
> **TLDR: We use a standard and fair setup from OpenOOD v1.5. While OptFS prioritizes zero-tuning generalization, AdaSCALE achieves superior performance by allowing for minimal tuning on a single hyperparameter, a valid trade-off for post-hoc methods.**
>
> We use the standard setup of OpenOOD v1.5 which utilizes the validation split (~1k images) of OpenImage-O. We believe there is a significant variation in the OOD test set as it contains other OOD datasets such as: NINCO, SSB-Hard, iNaturalist, and Textures. Additionally, we also include ImageNet-O and Places OOD datasets for extensive evaluation.
>
> The dataset-specific results can be found in the Appendix (Sec I). Even when ignoring results on the OpenImage-O test-split, AdaSCALE still shows overwhelming empirical superiority. However, yes, we agree that the line of work OptFS adopts can simply set one hyperparameter value across all cases for near-optimal performance. And, another line of work AdaSCALE adopts (SCALE/ASH/LTS) needs to tune one hyperparameter value for its optimal performance. But, it can also be argued that the latter line of work’s efficacy doesn’t depend on access to a large number of training samples.
>
> We sincerely believe the contribution of AdaSCALE should be properly situated with respect to its direct predecessors such as SCALE/LTS/ASH. However, we acknowledge the pioneering contribution of OptFS in terms of cross-architecture generalization.
>
> ### 5. On the Stability of Using Few ID Validation Samples
>
> **TLDR: New experiments on ResNet-101 and multi-trial runs with just 5 samples on ResNet-50 confirm that performance drop is trivial.**
>
> While we presented results for the ResNet-50 in Table 9, to directly address the concern of stability and potential bias from a single experiment, we have conducted the same analysis on the **ResNet-101 architecture**. The results are presented below:
>
> | $n_\text{val}$ | Near-OOD (FPR@95↓ / AUROC↑) | Far-OOD (FPR@95↓ / AUROC↑) |
> | :--- | :---: | :---: |
> | 10 | 59.59 / 81.68 | 19.40 / 95.78 |
> | 100 | 57.15 / 81.85 | 18.58 / 95.94 |
> | 1000 | 56.75 / 81.80 | 18.53 / 95.93 |
> | 5000 | 57.96 / 81.68 | 18.51 / 95.95 |
>
> As the table shows, the results are consistent. The performance with just **10 validation samples** is nearly identical to the performance achieved with 5000 samples.
>
> We further present the results with the ResNet-50 architecture across 3 trials using **only 5 samples** below:
>
> | $n_\text{val}$ | Near-OOD (FPR@95↓ / AUROC↑) | Far-OOD (FPR@95↓ / AUROC↑) |
> | :--- | :---: | :---: |
> | 5 | 60.17 ± 1.01 / 78.83 ± 0.58 | 18.47 ± 0.27 /  96.01 ± 0.06 |
> | 5000 | 57.96 / 81.68 | 18.51 / 95.95 |
>
> The results show that there is no such significant performance drop in using just 5 samples across 3 trials.
>
> ### 6. Typo in Formula
>
> Thank you, we will correct it.

---

> > ### Author Response · Authors · 2025-11-18
> >
> > ### 7. Overall explanation of our design rationale:
> >
> > **TDLR: As mentioned in Line 425 - 426, estimated OODness alone — without adaptive scaling — does not result in strong performance. AdaSCALE works very well because it allows the seamless fusion of three independent and effective OOD detection signals: activation pattern, logit (directly or indirectly), and $Q’$ (which we introduce through _adaptive percentile_ instead of using fixed percentile).**
> >
> > **The Goal: Maximizing ID/OOD Score Separation**. The ultimate objective is to maximize the separation between the final energy scores of ID and OOD samples. This is most effectively achieved by applying a **stronger scaling factor** to _ID-likely_ samples and a **weaker scaling factor** to _OOD-likely_ samples.
> >
> > Fixed Scaling (for ex. LTS): In LTS, the logits are scaled with scaling factor $r$ as: $\mathbf{z}^{\text{scaled}} = \mathbf{z} \cdot r^{2}$  energy score is $E = -\log \sum_{i=1}^C e^{(\mathbf{z}^{scaled}_i)}$. This already fuses two independent OOD signals: (1) activation pattern, via $r$, and (2) the logits $\mathbf{z}$.
> >
> > **The Mechanism: The Need of Dynamic Percentile ( $p$ )**. Recall the static scaling formula, $r = sum(a) / sum(a > P_p(a))$. To achieve a larger scaling factor $r$, the denominator must be smaller. This happens when the percentile threshold $p$ is set to a higher value, as more high-magnitude activations are excluded from the sum.
> >
> > - To achieve **stronger scaling for ID-likely** (our goal), they should be assigned a **higher percentile** $p$.
> > - To achieve **weaker scaling for OOD-likely**, they should be assigned a **lower percentile** $p$.
> >
> > **The Inevitable Conclusion: The Need for Dynamic Percentiles**. The reasoning above leads directly to the conclusion that a static, fixed percentile $p$ for all samples is inherently suboptimal. The percentile should ideally be dynamic and adaptive, conditioned on whether a sample is likely to be ID or OOD. This is the foundational motivation for moving beyond prior work like SCALE and ASH.
> >
> > **The Solution: Designing an OOD Likelihood Heuristic ( $Q'$ )**. To enable this dynamic adjustment, we need a reliable signal to estimate a sample's OODness before scaling. This is precisely the purpose of our proposed metric $Q'$. As detailed in Section 4.1, $Q'$ is designed based on the key observation that OOD samples exhibit pronounced activation shifts under minor perturbations. Therefore:
> >
> > - A **high $Q'$** value indicates a high OODness.
> > - A **low $Q'$** value indicates a high IDness.
> >
> > **Synthesizing the Components into AdaSCALE**. We now have all the logical pieces:
> >
> > **Goal**: High $p$ for ID-likely samples, low $p$ for OOD-likely samples.
> >
> > **Tool**: $Q'$ as an estimate of OODness.
> >
> > We design the adaptive percentile calculation (Equation 9) to directly implement this inverse relationship: $p = p_\text{min} + (1 - F_{Q'}(Q')) * (p_\text{max} - p_\text{min})$
> >
> > Here, the $(1 - F_Q'(Q'))$ term is critical. When a sample has a high $Q'$ (is OOD-likely), its $F_Q'(Q')$ is high, making $(1 - F_Q'(Q'))$ low. This pushes $p$ towards $p_\text{min}$. Conversely, for a low $Q'$ (ID-likely) sample, $p$ is pushed towards $p_\text{max}$.
> >
> > In essence, **the design of AdaSCALE was a principled effort to inject another independent source of OOD signal** (activation shift at the expense of extra forward pass) directly into the scaling mechanism itself. While prior methods use a static mechanism to amplify an existing signal, AdaSCALE makes the amplification mechanism itself intelligent and adaptive to the nature of each sample.

---

> ### Author Response · Authors · 2025-11-18
>
> ## Question 1.
>
> This is a **great question** which helps us to clarify the core misunderstanding. Let’s reframe it in this way: “Would _activation pattern_ risk misclassifying OOD as ID or vice versa?” Yes. “Would _logit information_ risk misclassifying OOD as ID or vice versa?” Yes. “Would _activation shift at high-magnitude activations_ risk misclassifying OOD as ID or vice versa?” Yes. Indeed, there is no oracle signal which is perfect on its own. However, if all of these sources of OOD detection signal are somewhat independent and there is a smart fusion of these signals, they can work together to provide robust final signal that can be used for OOD detection.
>
> ## Question 2.
> Indeed, this is also very **insightful question** which we believe to have somewhat answered just above. The notable work OptFS uses _modified logit_ obtained with activation reshaping (note: **interval specific scaling factor**). However, what if, for a particular sample, _original logit_ contained correct OOD detection signal and activation reshaping was not reliable for it? Specifically for near-OOD samples, activations can have similar pattern to that of ID distribution and activation reshaping may lead to suboptimality instead. For example, see Sec. I for dataset-specific results for OptFS on the SSB‑Hard dataset. This is not to say one OOD detection signal should be prioritized more than other in this case. We hypothesize that there needs to be proper fusion of independent OOD detection signals where if one signal fails, another can compensate leading to correct OOD detection. Furthermore, AdaSCALE uses extra independent OOD detection signals “activation shifts at top-k activations” while OptFS does not. The comparison between OptFS and AdaSCALE **-L** makes the difference even more clearer.
>
> ---
> We hope these detailed responses have resolved the reviewer’s concerns. We appreciate Reviewer GRZb's insightful questions which, we believe, has led to resolving the confusions/misunderstandings.

---

> ### Author Response · Authors · 2025-11-21
>
> ### On Theoretical Justification
>
> We agree that a deeper theoretical justification would be a valuable contribution and an important direction for future research. While our work does not offer a formal proof, we believe the core phenomenon is highly intuitive and follows directly from well-established observations in the field. We further attempt to explain our intuition below:
>
> *   **ID Stability:** Since a model is trained on the ID distribution, we expect it to be robust to minor, trivial perturbations of ID inputs. It has learned the underlying data manifold, so small changes should not drastically alter the high-level semantic features, leading to a stable activation pattern.
> *   **OOD Instability:** The model has no explicit knowledge of OOD data. From its perspective, a minor perturbation applied to an OOD input can create what it perceives as a *completely different* OOD input. We can borrow from the well-known observation from ReAct that "OOD inputs have abnormally high activations."
> *   **The Shift in High Activations:** Because the original OOD input and the trivially perturbed OOD input are seen as two distinct, unfamiliar samples by the model, both will likely trigger "abnormally high activations." However, the probability that these high activations occur in the exact same set of neurons is **relatively** low. This may result in a significant shifts of the highest-magnitude activations, which our method effectively measures.
>
> We are grateful to Reviewer GRZb for their thoughtful feedback. Their comments were instrumental in helping us clarify the core intuition behind the activation shift phenomenon, and we believe the paper is much stronger as a result. We respectfully request that our detailed clarifications be considered in the final assessment.

---

### Official Review · Reviewer_UieH · 2025-11-06

**Soundness:** 2
**Presentation:** 2
**Contribution:** 2
**Rating:** 4
**Confidence:** 4

**Summary:**

This paper proposes AdaSCALE, a post-hoc method for out-of-distribution (OOD) detection that adaptively adjusts the activation scaling percentile based on the estimated OOD likelihood of each sample. The method is motivated by the observation that OOD inputs exhibit stronger top-k activation shifts under small perturbations than ID inputs. The authors design a mechanism to estimate sample OODness via activation differences and dynamically modulate the scaling strength. Extensive experiments on ImageNet-1k, CIFAR-10/100, and various architectures demonstrate consistent gains over prior post-hoc methods (e.g., SCALE, LTS, OptFS), achieving state-of-the-art performance.

**Strengths:**

- Motivation and observation are clear and intuitive. The link between activation stability under perturbation and OOD likelihood is conceptually appealing.

- Strong empirical performance across datasets and architectures shows that the adaptive scaling strategy generalizes better than fixed-threshold methods.

- Method simplicity.

**Weaknesses:**

- Incremental novelty. While the adaptive percentile idea is reasonable, it extends existing activation scaling works (e.g., ASH, SCALE, LTS) rather than introducing a fundamentally new principle.

- Limited theoretical grounding. The paper claims that activation shift reflects OODness, but lacks a formal analysis or statistical justification. The perturbation mechanism and threshold mapping are mostly heuristic.  It may disentangle this effect from other confounders (e.g., gradient norm, layer saturation, or input magnitude)

- Clarity issues. Some equations and algorithmic steps (e.g., computation of OODness score and eCDF calibration) are dense and could benefit from clearer notation or pseudo-code.

- Minor reproducibility concern. The reliance on gradient-based perturbations may introduce stochasticity; details of implementation choices (e.g., ε magnitude, percentile selection ranges) could be more transparent.

**Questions:**

- How sensitive is AdaSCALE to the choice of perturbation direction and strength (ε)? Could adversarial directions yield different results?

- Maybe the authors could provide more intuition or quantitative evidence that the activation shift magnitude correlates with the sample’s epistemic uncertainty rather than simply gradient magnitude?

---

> ### Author Response · Authors · 2025-11-18
>
> We thank Reviewer UieH for the valuable feedback and for recognizing our work's **"clear and intuitive"** motivation, **"strong empirical performance,"** and **"method simplicity."**
>
> We believe the concerns raised are primarily areas that we can clarify. We have addressed all points below.
>
> ### 1. On Incremental Novelty
>
> **TLDR: Our work introduces a new paradigm of adaptive scaling by successfully fusing three independent OOD detection signals, a non-trivial conceptual advance that leads to state-of-the-art performance.**
>
> While we agree that AdaSCALE builds upon the successful foundation of prior activation-scaling methods (ASH, SCALE, LTS), we respectfully argue that our extension has non-trivial impact. As Reviewer Vxks notes, our work **“moves scaling-based OOD detection from heuristic to a more principled footing”** and provides **“a significant conceptual advancement”** by **"discovering an important phenomenon"**. To achieve such significant empirical performance, our work makes the following novel contributions:
>
> *   Our work makes a **proper successful fusion** of three independent OOD detection signals: activation pattern, logits (indirectly or directly) and activation shift at top-k activations ($Q$). Here, identifying activation shift at top-k activations as a possible **independent** OOD detection signal is a novel contribution in itself. We further show that naive usage of $Q$ can lead to relying on overconfident scaling factors and use a correction term to tame the overconfident scaling factors so as to preserve the valuable information contained in logits. (We respectfully direct Reviewer UieH to **Sec D.1**. for more details.)
> *   Furthermore, injecting this identified independent OOD detection signal through a dynamic percentile hyperparameter is a non-trivial addition, though in hindsight it may look like a straightforward addition.
> *   Overall, our work introduces the novel paradigm of adaptive scaling.
>
> ### 2. On Gradient magnitude / directions / reproducibility
>
> **TLDR: Our method is highly reproducible, as the gradient is not even necessary; random perturbation works just as well.**
>
> Regarding reproducibility and the perturbation mechanism, we want to clarify a key point that also addresses the question about perturbation direction. As we state in Lines 229-233:
>
> > Remark: Is gradient attribution necessary for perturbation? While we employ gradient-based attribution for principled pixel selection for perturbation, as we show later in Section E.5, it is important to note that **even random selection empirically performs similarly**, whereas selecting salient pixels degrades performance.
>
> Since the gradient is not required, any concerns about stochasticity or the specific direction of the gradient are moot. The core idea is to introduce a minor, trivial perturbation to observe the activation stability. Perturbing a small percentage of pixels randomly is sufficient, which also reduces computational cost.
>
> To address the question about the perturbation strength (ε), we provide the following sensitivity studies from our paper:
>
> **Table: Sensitivity study of `+ε` with ResNet-50 on ImageNet-1k**
> | | **Near-OOD** | | **Far-OOD** | |
> | :--- | :---: | :---: | :---: | :---: |
> | **ε** | **FPR@95 ↓** | **AUROC ↑** | **FPR@95 ↓** | **AUROC ↑** |
> | 0.1 | 63.76 | 77.50 | 19.26 | 95.85 |
> | **0.5** | **58.97** | **78.98** | **17.84** | **96.14** |
> | 1.0 | 61.60 | 76.96 | 19.31 | 95.84 |
>
> **Table: Sensitivity study of `-ε` with ResNet-50 on ImageNet-1k**
> | | **Near-OOD** | | **Far-OOD** | |
> | :--- | :---: | :---: | :---: | :---: |
> | **ε** | **FPR@95 ↓** | **AUROC ↑** | **FPR@95 ↓** | **AUROC ↑** |
> | 0.1 | 62.01 | 77.82 | 18.73 | 95.95 |
> | **0.5** | **58.98** | **78.99** | **17.88** | **96.13** |
> | 1.0 | 60.43 | 78.42 | 18.54 | 95.99 |
>
> The performance is stable across different strengths, with an optimal value around 0.5. We hypothesize this is because the mean of the normalized RGB statistics is (0.485, 0.456, 0.406), and a perturbation magnitude of 0.5 is sufficient to disrupt pixel information and cause a measurable disturbance in the peak activations.

---

> > ### Author Response · Authors · 2025-11-18
> >
> > ### 3. Role of _activation shift magnitude_ / Overall design rationale of AdaSCALE:
> >
> > **TDLR: As mentioned in Line 425 - 426, estimated OODness alone (or _activation shift magnitude_) — without adaptive scaling — does not result in strong performance. AdaSCALE works very well because it allows the seamless fusion of three independent and effective OOD detection signals: activation pattern, logit (directly or indirectly), and $Q’$ (which we introduce through _adaptive percentile_ instead of using fixed percentile).**
> >
> > **The Goal: Maximizing ID/OOD Score Separation**. The ultimate objective is to maximize the separation between the final energy scores of ID and OOD samples. This is most effectively achieved by applying a **stronger scaling factor** to _ID-likely_ samples and a **weaker scaling factor** to _OOD-likely_ samples.
> >
> > Fixed Scaling (for ex. LTS): In LTS, the logits are scaled with scaling factor $r$ as: $\mathbf{z}^{\text{scaled}} = \mathbf{z} \cdot r^{2}$  energy score is $E = -\log \sum_{i=1}^C e^{(\mathbf{z}^{scaled}_i)}$. This already fuses two independent OOD signals: (1) activation pattern, via $r$, and (2) the logits $\mathbf{z}$.
> >
> > **The Mechanism: The Need of Dynamic Percentile ( $p$ )**. Recall the static scaling formula, $r = sum(a) / sum(a > P_p(a))$. To achieve a larger scaling factor $r$, the denominator must be smaller. This happens when the percentile threshold $p$ is set to a higher value, as more high-magnitude activations are excluded from the sum.
> >
> > - To achieve **stronger scaling for ID-likely** (our goal), they should be assigned a **higher percentile** $p$.
> > - To achieve **weaker scaling for OOD-likely**, they should be assigned a **lower percentile** $p$.
> >
> > **The Inevitable Conclusion: The Need for Dynamic Percentiles**. The reasoning above leads directly to the conclusion that a static, fixed percentile $p$ for all samples is inherently suboptimal. The percentile should ideally be dynamic and adaptive, conditioned on whether a sample is likely to be ID or OOD. This is the foundational motivation for moving beyond prior work like SCALE and ASH.
> >
> > **The Solution: Designing an OOD Likelihood Heuristic ( $Q'$ )**. To enable this dynamic adjustment, we need a reliable signal to estimate a sample's OODness before scaling. This is precisely the purpose of our proposed metric $Q'$. As detailed in Section 4.1, $Q'$ is designed based on the key observation that OOD samples exhibit pronounced activation shifts under minor perturbations. Therefore:
> >
> > - A **high $Q'$** value indicates a high OODness.
> > - A **low $Q'$** value indicates a high IDness.
> >
> > **Synthesizing the Components into AdaSCALE**. We now have all the logical pieces:
> >
> > **Goal**: High $p$ for ID-likely samples, low $p$ for OOD-likely samples.
> >
> > **Tool**: $Q'$ as an estimate of OODness.
> >
> > We design the adaptive percentile calculation (Equation 9) to directly implement this inverse relationship: $p = p_\text{min} + (1 - F_{Q'}(Q')) * (p_\text{max} - p_\text{min})$
> >
> > Here, the $(1 - F_Q'(Q'))$ term is critical. When a sample has a high $Q'$ (is OOD-likely), its $F_Q'(Q')$ is high, making $(1 - F_Q'(Q'))$ low. This pushes $p$ towards $p_\text{min}$. Conversely, for a low $Q'$ (ID-likely) sample, $p$ is pushed towards $p_\text{max}$.
> >
> > In essence, **the design of AdaSCALE was a principled effort to inject another independent source of OOD signal** (activation shift at the expense of extra forward pass) directly into the scaling mechanism itself. While prior methods use a static mechanism to amplify an existing signal, AdaSCALE makes the amplification mechanism itself intelligent and adaptive to the nature of each sample.
> >
> > ---
> >
> > We hope these clarifications and detailed results address the concerns. We thank the reviewer for the opportunity to make clarifications. We hope these clarifications will be considered in the final rating. We are happy to answer any further questions.

---

> > > ### Author Response · Authors · 2025-11-21
> > >
> > > ### On Theoretical Justification
> > >
> > > We agree that a deeper theoretical justification would be a valuable contribution and an important direction for future research. While our work does not offer a formal proof, we believe the core phenomenon is highly intuitive and follows directly from well-established observations in the field. We further attempt to explain our intuition below:
> > >
> > > *   **ID Stability:** Since a model is trained on the ID distribution, we expect it to be robust to minor, trivial perturbations of ID inputs. It has learned the underlying data manifold, so small changes should not drastically alter the high-level semantic features, leading to a stable activation pattern.
> > > *   **OOD Instability:** The model has no explicit knowledge of OOD data. From its perspective, a minor perturbation applied to an OOD input can create what it perceives as a *completely different* OOD input. We can borrow from the well-known observation from ReAct that "OOD inputs have abnormally high activations."
> > > *   **The Shift in High Activations:** Because the original OOD input and the trivially perturbed OOD input are seen as two distinct, unfamiliar samples by the model, both will likely trigger "abnormally high activations." However, the probability that these high activations occur in the exact same set of neurons is low. This may result in a significant shifts of the highest-magnitude activations, which our method effectively measures.
> > >
> > > We are grateful to Reviewer UieH for encouraging us to provide detailed intuition behind the method. We respectfully request that our detailed clarifications be considered in the final assessment. Please let us know if you have any outstanding concerns so we can resolve them.

---

### Official Review · Reviewer_Ev7Q · 2025-11-09

**Soundness:** 3
**Presentation:** 2
**Contribution:** 3
**Rating:** 4
**Confidence:** 4

**Summary:**

The paper introduces a post-hoc OoD detection method, AdaSCALE. The basis of the proposal is that OoD inputs exhibit higher activation instability for minor perturbations compared to ID samples. AdaSCALE proposes an adaptive scaling mechanism and computes a per sample OODness score Q'.

**Strengths:**

- the phenomenon according to which ID inputs activate stable features while OoD samples cause unstable high-magnitude activations under small perturbations is well highlighted and illustrated (Figure 2)

- the score mapping strategy is straightforward and effective. The authors rely on an empirical CDF built using a small ID validation set. This results in a simple and non-parametric way of translating the scores into percentile ranges.

- extensive comparison with other relevant approaches (e.g. ReAct, ASH, SCALE, LTS, OptFS) and strong overall performance, including some great results (e.g. FPR@95 of 61.17 on imagenet-1k vs 71.91 for OptFS)

**Weaknesses:**

- despite the generalization claims, the presented method seems to be systematically over-tuned for each test setup. The authors state that a fixed set of hyperparams may be used, and that by tuning only $p_{max}$ "near-optimal performance can be achieved", see Table 5. However, the main results (Tables 2 and 3) are obtained following a humongous level of finetuning (and implicitly computational cost), apparent in Tables 28 and 29. Presenting Tables 2/3 and claiming generalization is grossly misleading. I see two ethical ways out : either the authors claim a good generalization across all setups and finetuning goes to appendix, or the generalization claim is dropped and Table 5 goes to Appendix.

- the OODNess metric is artificially complex; the introduction of the correction term + 2 additional parameters is justified by the "high variance" of Q alone. However, Q has a FPR@95 of 59.43 vs 58.97 for the full metric.In my opinion, this shows that Q works quite well, and that the additional complexity and cost implied by the use of Q' are hard to justify.

- I also am very doubtful about the positive impact of the proposed perturbation mechanism; the "trivial" method performs a backward pass and is 2.91x heavier, while the random perturbation is only 1.56x slower. The benefit for the doubled cost is minimal (Table 22). This trade-off seems poor and is poorly described; the work should probably propose the random selection by default and introduce the other mechanism as a costlier alternative. Overall, the 3x slowdown (compared to SCALE) is not even once mentioned in the abstract / intro / conclusion; in my opinion this is a significant omission and a failure in transparency about the method's trade-offs.

**Questions:**

Please see the three points raised in the Weaknesses section.

**Details Of Ethics Concerns:**

everything OK

---

> ### Author Response · Authors · 2025-11-15
> **Clarifications regarding the three misunderstandings.**
>
> We thank Reviewer Ev7Q for the constructive feedback and for recognizing that our phenomenon is well highlighted, our score mapping is straightforward and effective, and our comparisons are extensive with strong overall performance. The main concerns seem to stem from misunderstandings about our hyperparameter tuning and design motivations; we clarify these below.
>
> ---
>
> # **1. "systematically over-tuned" and "humongous level of finetuning".**
>
> **TLDR: Our main results are not over-tuned. The non-percentile hyperparameters were fixed after tuning on a single model (ResNet-50), a standard practice. The "humongous finetuning" mentioned refers only to tuning two percentile hyperparameters, and the performance difference from tuning only one is trivial.**
>
> We would like to clarify our hyperparameter tuning protocol. The core, non-percentile hyperparameters of AdaSCALE ($\lambda$, $\epsilon$, $o$, $k_1$, $k_2$) were determined **only once** using the de-facto standard model for ImageNet-1k evaluation, ResNet-50. These values were then **fixed** for all other architectures and experiments.
>
> The only difference between our main results (Tables 1 & 2) and the generalization results (Table 5) is the number of percentile hyperparameters tuned, which is a common practice in related works (e.g., ASH, SCALE, LTS).
> *   **Main Tables (1, 2):** We tuned two percentile values, $p_\text{min}$ and $p_\text{max}$, for each setup to show the best possible performance of AdaSCALE.
> *   **Generalization Table (5):** We tuned only one, $p_\text{max}$, to directly compare with prior works that tune a single hyperparameter.
>
> As the table below demonstrates, the performance difference between these two protocols is minimal, which **contradicts** the claim of **"grossly misleading"** and requirement of **"humongous"** tuning for its main results.
>
> | **Method** | **ImageNet-1k Near-OOD** | **ImageNet-1k Far-OOD** | **CIFAR-100 Near-OOD** | **CIFAR-100 Far-OOD** |
> | :--- | :--- | :--- | :--- | :--- |
> | Tune ($p_\text{min}$, $p_\text{max}$) / Tables 1, 2 | **61.17** / **80.50** | **30.11** / **92.73** | **57.33** / **81.35** | **54.53** / **81.14** |
> | Tune ($p_\text{max}$ only) / Table 5 | 62.29 / 79.72 | 32.72 / 91.82 | 58.00 / 81.14 | 56.47 / 80.99 |
>
> The difference is indeed trivial, confirming that AdaSCALE generalizes well with just one hyperparameter tuning.
>
> ---
>
> # **2. "OODNess metric is artificially complex".**
>
> **TLDR: The complexity of our OODness metric (Q') is necessary and justified. It consistently provides significant performance gains over the simpler metric (Q) across eight diverse model architectures by preventing overconfident scaling factors from dominating the final score.**
>
> As we state in the main paper (Lines 268-269), the formulation of Q' is designed to solve a specific problem: overconfident scaling factors in OOD samples that can mask the underlying logit information. This correction term ensures a healthier balance where both feature-level instability (captured by Q) and the original logit distribution contribute to the final OOD score.
>
> To validate this, we direct the reviewer to our extensive ablation study in **Sec D.1**, which shows that Q' offers a consistent and significant performance improvement over Q alone across all eight tested architectures:
>
> | **Category** | **Metric** | **Average (FPR@95/AUROC)** |
> | :--- | :--- | :--- |
> | near-OOD | *Q* | 68.90/78.07 |
> | | *Q'* | **61.17/80.50** |
> | far-OOD | *Q* | 42.63/89.18 |
> | | *Q'* | **30.11/92.73** |
>
> The data clearly shows that the additional complexity is not artificial but rather a well-motivated and empirically validated design choice that leads to superior performance. **See Illustrative Example from Sec D.1.**
>
> ### Further justification on the design of $Q’$
> Scaled energy scores perform well by combining two complementary signals: feature-space activation patterns and logits. We inject third, independent signal, activation shift, by replacing the fixed percentile with an adaptive percentile.
>
> # **3. On the concern about the perturbation mechanism's trade-offs.**
>
> As mentioned in Line 214 - 215, our work is based on the hypothesis that “the positions of such high activations in OOD samples are relatively unstable under **minor perturbations** compared to ID samples.“
>
> **The gradient-based method was the most principled approach to test this** by perturbing a minimal and targeted set of trivial pixels. However, as the reviewer correctly points out, our own results in Table 22 show that random pixel selection performs similarly well, with a much lower computational overhead (1.56x slower vs. 2.91x). We agree this is a crucial finding. We view this as a strength of AdaSCALE: the core idea is robust even to a much simpler and more efficient perturbation strategy.
>
> ---
>
> We will include relative latency of AdaSCALE in the introduction. We hope these clarifications have addressed the reviewer's concerns. We are happy to answer any further questions.

---

> > ### Author Response · Authors · 2025-11-22
> >
> > We think it would be helpful to further emphasize our contributions.
> >
> > As Reviewer Vxks notes, our work **“moves scaling-based OOD detection from heuristic to a more principled footing”** and provides **“a significant conceptual advancement”** by **"discovering an important phenomenon"**. To achieve such significant empirical performance, our work makes the following novel contributions:
> >
> > *   Our work makes a **proper successful fusion** of three independent OOD detection signals: activation pattern, logits (indirectly or directly) and activation shift at top-k activations ($Q$). Here, identifying activation shift at top-k activations as a possible **independent** OOD detection signal is a novel contribution in itself. We further show that naive usage of $Q$ can lead to relying on overconfident scaling factors and use a correction term to tame the overconfident scaling factors so as to preserve the valuable information contained in logits. (We respectfully direct Reviewer Ev7Q to **Sec D.1**. for more detail.)
> > *   Furthermore, injecting this identified independent OOD detection signal through a dynamic percentile hyperparameter is a non-trivial addition, though in hindsight it may look like a straightforward addition.
> > *   Overall, our work introduces the novel paradigm of adaptive scaling.

---

### Meta-Review · Area_Chair_dyGP · 2026-01-08

**Summary:**

This paper tackles the problem of post-hoc OoD detection.  It proposes a per sample OODness score Q' and an adaptive thresholding thresholding strategy based on per-sample perturbations.

The reviewers appreciate the effectiveness of the method and its good performance.

They raised concerns about
- extent of tuning required for each test setup
- additional compute requirements
- theoretical grounding
- novelty of the methodology

**Reviewer Concerns:**

The majority of the clarification questions were well addressed by the rebuttal.  The manuscript needs to be transparent about the extra compute and extra number of passes the methodology requires compared to other post-hoc approaches.

The main concern which remains unaddressed by the work is the novelty and lack of theoretical grounding. Given that the methodology builds heavily on the previous approaches for OOD detection, one would expect to have strong theoretical insight for the proposed approach, rather than only intuitive outcomes.

**Reviewer Scores:**

Ev7Q: 4 - I don't think the reviewer would have changed their score
UieHJ: 4 - I don't think the reviewers would have changed their score, as the listened weaknesses on incremental novelty and limited theoretical grounding are not addressed in the rebuttal
GRZb: 2 - this reviewer may have increased their score from 2 to 4.  The main criticism is similar to previous, on incremental contribution and limited theoretical grounding
Vxks: 8 - this reviewer has a strong opinion of the paper already, unlikely to further increase the score
yGZh: 6 - I don't think this reviewer would change their score.  Most concerns are clarification based
upif: 4 - The reviews may increase their score

---

### Decision · Program_Chairs · 2026-01-26

Reject